# In-situ measurements of NH₃: instrument performance and applicability

Marsailidh M. Twigg[1], Augustinus J.C. Berkhout[2], Nicholas Cowan[1], Sabine Crunaire[3], Enrico Dammers[4], Volker Ebert[10],Vincent Gaudion[3], Marty Haaima[2], Christoph Häni[6], Lewis John[7], Matthew R. Jones[1], Bjorn Kamps[8], John Kentisbeer[1], Thomas Kupper[6], Sarah R. Leeson[1], Daiana Leuenberger[9], Nils O. B. Lüttschwager[10], Ulla Makkonen[11], Nicholas A. Martin[12], David Missler[13], Duncan Mounsor[7], Albrecht Neftel[14], Chad Nelson[15], Eiko Nemitz[1], Rutger Oudwater[15], Celine Pascale[9], Jean-Eudes Petit[5,13], Andrea Pogany[10], Nathalie Redon[3], Jörg Sintermann[16], Amy Stephens[1], Mark A. Sutton[1], Yuk S. Tang[1], Rens Zijlmans[8‡], Christine F. Braban[1] and Bernhard Niederhauser[9].

[1]UK Centre for Ecology & Hydrology, Bush Estate, Penicuik, EH26 0QB, UK.
[2]National Institute for Public Health and the Environment (RIVM), Antonie van Leeuwenhoeklaan 9, 3721 MA Bilthoven, the Netherlands
[3]IMT Nord Europe, Univ. Lille, CERI EE, 59000 Lille, France
[4]TNO, Department of Climate, Air and Sustainability, Utrecht, the Netherlands
[5]Laboratoire des Sciences du Climat et de l'Environnement, CEA / Orme des Merisiers; 91191 Gif-sur-Yvette, France
[6]Bern University of Applied Sciences, School of Agricultural, Forest and Food Sciences , Laenggasse 85, 3052 Zollikofen, Switzerland
[7]Enviro Technology Services Ltd, UK
[8]LSE monitors BV, the Netherlands
[9]Federal Institute of Metrology (METAS), Lindenweg 50, 3003 Bern-Wabern, Switzerland
[10]Physikalisch-Technische Bundesanstalt, Bundesallee 100, 38116 Braunschweig, Germany
[11]Finnish Meteorological Institute, Erik Palmenin aukio 1, 00560 Helsinki, Finland
[12]National Physical laboratory, Air Quality and Aerosol Metrology Group, Teddington, Middlesex, TW11 0LW, UK
[13]Atmo Grand-Est, 5 rue de Madrid, 67300 Schiltigheim, France
[14]Neftel Research Expertise, Switzerland
[15]Tiger Optics, USA
[16]Office of Waste, Water, Energy and Air AWEL; Canton Zürich, Switzerland
‡ Deceased

*Correspondence to*: Marsailidh M. Twigg (sail@ceh.ac.uk)

**Abstract.**

Ammonia (NH₃) in the atmosphere affects both the environment and human health. It is therefore increasingly recognised by policy makers as an important air pollutant that needs to be mitigated, though it still remains unregulated in many countries. In order to understand the effectiveness of abatement strategies, routine NH₃ monitoring is required. Current reference protocols first developed in the 1990s, use daily samplers with offline analysis but there have been a number of technologies developed since, which may be applicable for high time resolution routine monitoring of NH₃ at ambient concentrations. The following study is a comprehensive field intercomparison held over an intensively managed grassland in South East Scotland using currently available methods that are reported to be suitable for routine monitoring of ambient NH₃. In total 13 instruments took part in the field study, including commercially available technologies, research prototype instruments and legacy instruments. Assessments of the instruments' precision at low concentrations (< 10 ppb) and at elevated concentrations (maximum reported concentration of 282 ppb) were undertaken. At elevated concentrations all instruments performed well on precision ($r^2$ >0.75). At concentrations below 10 ppb however, precision decreased and instruments fell into two distinct groups with duplicate instruments split across the two groups. It was found that duplicate instruments performed differently as a result of differences in instrument setup, inlet design and operation of the instrument.

New metrological standards were used to evaluate the accuracy in determining absolute concentrations in the field. A calibration-free CRDS Optical Gas Standard (OGS, PTB, DE) served as an instrumental reference standard, and instrument operation was assessed against metrological calibration gases from i) a permeation system (ReGaS1, METAS, CH) and ii) Primary Standard gas Mixtures (PSMs) prepared by gravimetry (NPL, UK). This study suggests that though the OGS gives

good performance with respect to sensitivity and linearity against the reference gas standards, this in itself is not enough for the OGS to be a field reference standard because in field applications a closed path spectrometer has limitations due to losses to surfaces in sampling $NH_3$, which are not currently taken into account by the OGS. Overall, the instruments compared with the metrological standards performed well but not every instrument could be compared to the reference gas standards due to incompatible inlet designs and limitations in the gas flow rates of the standards.

This work provides evidence that though $NH_3$ instrumentation have greatly progressed in measurement precision, there is still further work required to quantify the accuracy of these systems under field conditions. It is the recommendation of this study that the use of instruments for routine monitoring of $NH_3$ needs to be set out in standard operating protocols for inlet set-up, calibration and routine maintenance, in order for datasets to be comparable.

## 1    Introduction

Excess reactive nitrogen in the environment has been demonstrated to have environmental impacts, as highlighted by the European Nitrogen Assessment (ENA) (Sutton et al., 2011). The ENA identified five key threats of excess reactive nitrogen to Europe; **W**ater quality, **A**ir **Q**uality (AQ), **G**reenhouse gas (GHG) balance, **E**cosystem and biodiversity, and **S**oil quality (WAGES). Atmospheric ammonia ($NH_3$) plays a direct role in four of the five WAGES and is indirectly implicated in the greenhouse gas (GHG) balance as it influences the radiative balance through secondary aerosol formation. Ammonia is the highest concentration basic gas in the atmosphere forming secondary inorganic particulate matter of 2.5 μm or less in aerodynamic diameter ($PM_{2.5}$) following reaction with acidic gases. $PM_{2.5}$ has AQ impacts on human health, visibility and climate (Sutton et al., 2020). Vieno et al. (2016) have shown that reductions in $NH_3$ emissions in the United Kingdom (UK) would result in the reduction in $PM_{2.5}$, findings that were mirrored in the global studies of Gu et al. (2021) and Pozzer et al. (2017). Globally and across Europe, agriculture is the primary source of $NH_3$ emissions (>80 %) (Backes et al., 2016). It is predicted that current $NH_3$ emissions will increase under most future scenarios due to 1) a rise in global temperatures and 2) predicted growth in global consumption of animal products. Fowler et al. (2015) estimate that global annual emissions of $NH_3$ will increase from 65 Tg N yr$^{-1}$ in 2008 to 135 Tg N yr$^{-1}$ in 2100, based on an assumed increase in global warming of 5˚C in 2100 and the continued increase in the global consumption of animal products. There are however large uncertainties in $NH_3$ emission inventories, with up to an order of magnitude in some sectors (Kuenen et al., 2014). It is therefore essential to accurately measure ambient $NH_3$ concentrations to better quantify concentration and concentration changes and hence evaluate impacts of $NH_3$.

To understand the complexities of $NH_3$ in the atmosphere and provide evidence of the effectiveness of mitigation strategies, accurate, traceable routine $NH_3$ monitoring is required. One of the major challenges is to achieve accurate and precise $NH_3$ measurements at the source (typically >1 ppm), close to emission sources (typically >100 ppb) and ambient background concentrations (<0.1 to 10 ppb). Concentrations of $NH_3$ vary greatly across spatial and temporal scales, as this molecule is deposited rapidly and is also reactive in the atmosphere. Until recently, achieving quantitative artefact-free measurements of Long Term Monitoring at High Temporal Resolution (LTMHTR) required a high attention to detail and operation of instrumentation. This tended to only be economically feasible in the research domain; hence monitoring strategies of ambient $NH_3$ vary between countries. The United States Environmental Protection Agency (US EPA, 1997) and the European Monitoring and Evaluation Program (EMEP, 1996), current reference method is by sampling a known volume of air through acid coated denuders (typically citric acid) for 2-12 hours with offline analysis. The disadvantages of the US EPA/ EMEP denuder methods are that they are labour intensive and susceptible to handling and storage artefacts, as well as not providing the high temporal resolution information that state-of-the-art methods can provide. Individual European countries have taken different approaches, sometimes combining a few high temporal resolution monitoring of $NH_3$ sites (1 s to 1 hour), alongside

passive monitoring networks that sample at a lower frequency (weekly to monthly). The passive sampler networks tend to follow the recently published European standard diffusion sampler methodology, EN 17346: Ambient air - Standard method for the determination of the concentration of ammonia using diffusive samplers (CEN, 2020). In the Netherlands, LTMHTR has been carried out since 1992, initially using continuous flow annular wet rotating denuders (WRD) with selective ion membrane / conductivity analysis in an instrument called the Ammonia MOnitoR (AMOR, ECN, NL) until 2015, and then using differential optical absorption spectroscopy (DOAS, RIVM, NL) (Volten et al., 2012, Berkhout et al. 2017 ). In the UK, there are two LTMHTR (hourly) $NH_3$ measurements at rural background sites using WRDs with online ion chromatography analysis as implemented in the commercial Monitor for AeRosols and Gases in Ambient air (MARGA, Metrohm, NL) (Twigg et al., 2015). Wet chemistry LTMHTR instruments (AMOR and MARGA) require specialist operators and are labour intensive, however calibration and quality assurance are accurate and simple as they use liquid calibrations. The disadvantage to the wet chemistry approach is that there is the potential that at elevated concentrations not all $NH_3$ is captured by the WRD and for the selective ion membrane / conductivity analysis method it is not ion specific and therefore it is possible that there could be interference from other gas phase compounds.

There have been major advances in spectroscopic approaches to $NH_3$ measurement over the last 20 years. Previously mid-infrared (MIR) lead salt diodes required cryogenic cooling and frequently were multimodal but these have been replaced by stable, more powerful and monochromatic thermoelectrically cooled lasers. The development of reliable IR light sources, initially near-infrared (NIR) diode lasers and later mid-infrared quantum cascade lasers, resulted in an increasing number of spectroscopic instruments on the market. These include cavity ring down systems (CRDS, (Martin et al., 2016; Kamp et al., 2019)), Optical-feedback cavity-enhanced absorption spectrometers (OF-CEAS, (Leen et al., 2013; Leifer et al., 2017)), quantum cascade laser absorption spectrometers (QCLAS, (Whitehead et al., 2008; Ellis et al., 2010; Zöll et al., 2016)), open path Fourier Transform InfraRed systems (FTIR, (D. L. Bjorneberg et al., 2009; Suarez-Bertoa et al., 2017)) and photoacoustic methods (Pogány et al., 2009; von Bobrutzki et al., 2010; Liu et al., 2019). Recently, CRDS instruments have been introduced for routine ambient $NH_3$ monitoring in France, as well as the French national metrology institute has been involved in the calibration of the instruments (Macé et al., 2022). There are also other types of instruments e.g. utilising the ultraviolet (UV) spectrum for spectroscopy, and the aforementioned DOAS systems in the Dutch network. Chemical ionisation spectrometers (CIMS) including the Proton Transfer Reaction Mass Spectrometer (PTRMS, Ionicon) have been shown to be applicable for the measurement of $NH_3$ (Norman et al., 2009; von Bobrutzki et al., 2010; Pfeifer et al., 2019). There is no record in the literature of CIMS, however, being used for routine $NH_3$ monitoring, presumably due to their high acquisition cost.

Since the most recent $NH_3$ intercomparison studies (Schwab et al., 2007; Norman et al., 2009; von Bobrutzki et al., 2010), there are more LTMHTR instruments on the market, advertised to be applicable for routine $NH_3$ measurements. The instruments have become more affordable and now no longer, in theory, require specialist operators, resulting in reduced labour costs and some claim to provide quantitative measurements down into the parts per trillion (ppt) range. However, their capabilities under field conditions have still to be evaluated against established methods, as no standard protocols for setup, operations in the field and routine calibrations of these instruments exist. Traceable $NH_3$ gas standards are now available, but they have not been tested in field systems for undertaking routine in-field quality assurance and quality control.

This study reports a field intercomparison within a European Joint Research Project (EMRP), Metrology for $NH_3$ in ambient air (MetNH3, (Pogány et al., 2016)). MetNH3 aimed to improve comparability and reliability of ambient air $NH_3$ measurements by achieving metrological traceability for $NH_3$ measurements in the amount fraction range 0.5-500 ppb from primary certified reference material (CRM) and instrumental standards to the field level. In this study 13 instruments, including commercially available technologies, research prototype instruments and legacy instruments were deployed and exposed to concentrations

from background (<10 ppb) to elevated (>200 ppb). The instruments included: an online ion chromatography system (MARGA, Metrohm-Applikon,NL), two wet chemistry continuous flow analysis systems (AiRRmonia, Mechatronics, NL), a photoacoustic spectrometer (NH$_3$ monitor, LSE, NL), two mini Differential Optical Absorption Spectrometers (miniDOAS; NTB Interstate University of Applied Sciences Buchs, now part of "Eastern Switzerland University of Applied Sciences, CH and RIVM, NL"), as well as seven spectrometers using cavity enhanced techniques: a Quantum Cascade Laser Absorption Spectrometer (QCLAS, Aerodyne, Inc. US), a Picarro G2103 Analyzer (Picarro US), an Economical NH$_3$ Analyser (Los Gatos Research, US), a Tiger-i 2000 (Tiger Optics, US) and a ProCeas® gas analyser (AP2E, FR). In this study we evaluate the precision of these instruments by comparing their data to the ensemble median and studying the between instrument variability, including those operated on a common manifold, as recommended by von Bobrutzki et al. (2010). The importance of set-up and time response is also considered through the use of duplicate instruments with different inlet designs. Metrological methods developed under the MetNH3, are also evaluated under field conditions as standards for determining the accuracy of the instrumentation deployed, as previous studies of the metrological applications have focused on laboratory settings (Pogány et al., 2016, 2021). We discuss recommendations for future LTMHTR ambient NH$_3$ measurements, considering instrument capabilities and sampling set-ups to achieve high precision for use in routine monitoring of NH$_3$ and where further developments are still required in determining the accuracy of ambient NH$_3$ measurements.

## 2    Methods

### 2.1    Field site description

Instruments were deployed at an intensively managed grassland in South East Scotland, which lies approximately 12 km south of Edinburgh, between 22$^{nd}$ August - 2$^{nd}$ September 2016. The grass is dominated by *Lolium perenne* (perennial ryegrass) over an area of approximately 5 hectares, which is split into two fields. The instrumentation was positioned along the boundary between the two fields (Figure 1), which are typically used for intensive grazing. For the campaign, the South field with the dominant wind direction was being grown for silage in order that a uniform surface was available for the study. On the 23$^{rd}$ August both fields were fertilised with approximately 35 kg N ha$^{-1}$ of urea (pellets) to generate larger concentrations. This field site was previously used in an NH$_3$ intercomparison in 2008 (von Bobrutzki et al., 2010) where an application of 35 kg N ha$^{-1}$ urea resulted in NH$_3$ concentrations of up to 120 ppb at the site.

### 2.2    Instrumentation

During the campaign, instrumentation was housed in either the tow van or the mobile laboratory, the exceptions were the open path miniDOAS instruments that were positioned on the scaffolding and the AiRRmonia#1, which is designed to be operated outside to minimise the inlet used. All participants were given the opportunity to sample from a common high flow inlet, where applicable. The instruments housed in the mobile laboratory shared a high flow inlet with a Pyrex manifold, with the exception of the MARGA. This manifold set-up used a 1/2" (ID) polyethylene (PE) tubing with a length of 3.5 m (sampling point to manifold) with an airflow of 50.08 l min$^{-1}$, when all instruments were operational. The residence time from the sampling point to manifold exit was calculated to be ~1.62 s . All instruments were configured to sample at a height of approximately 1.7 m. Table 1 presents a summary of all instrumentation employed including sampling position, reporting temporal resolution and manufacturer/user reported limit of detection. Table 2 summarises, where applicable, instrument inlet characteristics including length, flow rate, residence time, air velocity and Reynolds number. The table also states if the instrument has a filter inline for sampling.

a)

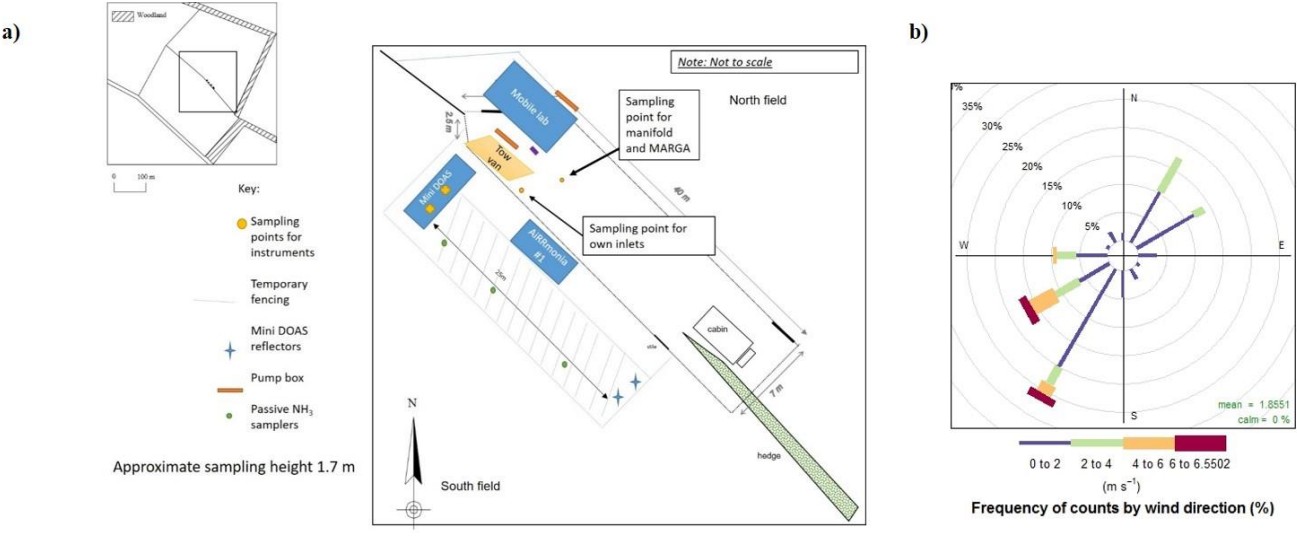

b)

c)

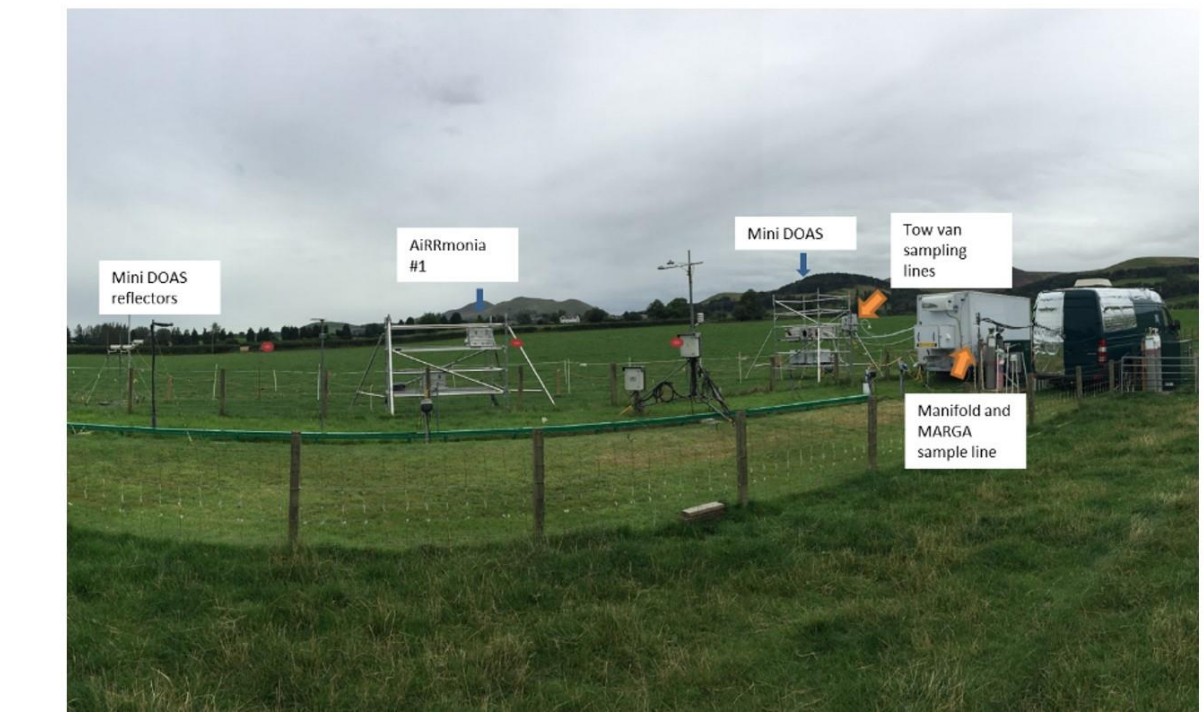

Figure 1. a) Layout of field site b) wind rose of wind direction and wind speed for the period 23rd August to 29th August ( Generated using OpenAir package in R (Carslaw and Ropkins, 2012)  and c) photo of setup instruments (*photo credit: M. Coyle, UKCEH*)

**Table 1.** Summary of instrumentation that participated in the campaign. Measurement height was set to approximately 1.7 m for all instruments. N/A= not applicable. - = Where information is unknown. LOD= limit of detection. Instruments that used the common manifold are shaded in grey. (Note: Reporting time by the instrument was selected by the operator)

| Location | Name used in this study | Manufacturer Name | Manufacturer | Availability | Flowrate (SLPM) | Reporting time (s) | Accuracy [%] | Manufacturer reported Precision (ppb) | Reporting range |
|---|---|---|---|---|---|---|---|---|---|
| Scaffold 1 | miniDOAS #1 | miniDOAS | NTB, Buchs, CH | Adapted | N/A | 60 | unavailable | 0.26[4] | Unknown* |
| | miniDOAS #2 | miniDOAS | RIVM, NL | Commercially available | N/A | 60 | 2 | 0.36 | 0.36 - 430 ppb (Currently validated, higher is possible) |
| Scaffold 2 | AiRRmonia[1] #1 | AiRRmonia[1] | Mechatronics BV, Netherlands | Discontinued | 1.0 | 60 | 3 | 0.05 | 0.04 to 500 ppb |
| Tow van | QCLAS | Mini-TILDAS Ammonia Monitor | Aerodyne Inc, USA | Commercially available | 13± | 1 | - | 0.045 at 300 sec | 0-1 ppm |
| | AP2E | ProCeas® gas analyser[1] | AP2E, France | Commercially available | 1.0 | 60 | - | LOD: 10 | 0 - 10 ppm |
| | AiRRmonia#2 | AiRRmonia[1] | Mechatronics BV, Netherlands | Discontinued | 1.0 | 350 | 3 | 0.05 | 0.04 to 500 ppb |
| | Picarro #1 | Picarro G2103 Analyzer[1] | Picarro Inc. USA | Commercially available | 0.8 | 60 | ±5 % of reading + 0.5 ppb | 0.5 at 1 s 0.17 at 10 s 0.03 at 300 s | Guaranteed range: 0-500 ppb Operational range: 0-10 ppm |
| Mobile lab | LGR #1 | Economical NH₃ Analyser (EAA 30r)[1] | Los Gatos Research (LGR) Inc, USA | Commercially available | 0.25 | 1 | - | <1.5 at 1 s <0.6 at 10 s <0.2 at 100 s | Measurement Range: 0.5 – 10000 ppb Operational Range: 0-200 ppm |
| | LGR #2 | Economical NH₃ Analyser (EAA 30r)[1] | Los Gatos Research (LGR) Inc, USA | Commercially available | 2.30 | 5 | - | <1.5 ppb at 1 s <0.6 ppb at 10 s <0.2 ppb at 100 s | Measurement Range: 0.5 – 10000 ppb Operational Range: 0-200 ppm |
| | Picarro #2 | Picarro G2103 Analyzer[1] | Picarro Inc. USA | Commercially available | 1.35 | 60 / 300 (OGS) | ±5 of reading + 0.5 ppb | 0.5 at 1 s 0.17 at 10 s 0.03 at 300 s | Guaranteed range: 0-500 ppb Operational range: 0-10 ppm |
| | Tiger Optics | Tiger-i 2000[1] | Tiger Optics, USA | Superseded by T-I Max | 0.48 | 1 | ±4 of reading or ½ LOD | 0.83 | 0-40 ppm |
| | LSE | NH₃ monitor[1] | LSE monitors, NL | Commercially available | 0.10 | 60 | - | ±4 % of reading or 2 ppb | 0-15 ppm |
| | MARGA | MARGA[1] | Metrohm-Applikon, NL | Superseded by 2060 MARGA | 16.7 | 3600 | 6 | 5 % | Range is not reported by manufacturer but limit of detection is 0.05 µg m⁻³ on a loop. |
| Posts | ALPHA[2] | UKCEH ALPHA® | UK Centre for Ecology & Hydrology, UK | Available | N/A | N/A | 15 % | 3 % | 0.04 to 137 ppb |
| Mobile lab | Optical Gas Standard (OGS)[3] | Picarro G2103 Analyzer[1] (Picarro #2) | Hardware : Picarro Inc. USA  Software: PTB, DE | Research prototype | 1.35 | 300 | ±3% @k=2 of reading + 0.5 ppb | 0.5 at 1 s 0.17 at 10 s 0.03 at 300 s | Guaranteed range: 0-500 ppb Operational range: 0-10 ppm |

[1]Based on manufacturers specifications. [2]Reference method used to assess the homogeneity between miniDOAS and reflectors,

[3]Metrological developed algorithm which uses Picarro #2 (refer to Section 2.3.1).[4]Taken from Sintermann et al. (2016), *Estimated to be 500 ppb when the pathlength is 100 m.±Flowrate upstream of the inlet critical orifice, flowrate after orifice is 11.16 l min⁻¹ with only 90% of the flow going to the inlet.

**Table 2.** Summary of the characteristics of the inlets. Instruments sampling through the common manifold are shaded in grey and data presented is the length between the manifold and the instruments. The characteristics of the manifold inlet and manifold dimensions are presented separately. Residence time is the time taken from sampling point to entry point of instrument based on flowrate and volume of line. ‡ - Line heated. (PFA –Perfluoroalkoxy, PTFE – Polytetrafluoroethylene)

| | AiRRmonia #1 | QCLAS[‡] | AP2E | AiRRmonia #2 | Picarro #1 | LGR#1 | LGR #2[‡] | Picarro #2 | Tiger Optics | LSE | MARGA | manifold inlet | manifold |
|---|---|---|---|---|---|---|---|---|---|---|---|---|---|
| Inlet material | PTFE | Quartz /PFA | PTFE | PTFE | PTFE | PTFE | PTFE | PTFE | PTFE | PTFE | PE | PTFE | Pyrex |
| Length (m) | 0.05 | 3.00 | 4.69 | 6.40 | 4.88 | 2.00 | 1.45 | 2.15 | 2.64 | 1.12 | 8.46 | 3.50 | 0.30 |
| Inner diameter (mm) | 4.7 | 10 | 2 | 4.7 | 2 | 4.7 | 4.7 | 4.7 | 4.7 | 1.6 | 9.5 | 10 | 70 |
| Flow velocity (m s$^{-1}$) | 0.94 | 2.21 | 5.31 | 0.94 | 4.24 | 0.23 | 2.15 | 1.15 | 0.45 | 0.81 | 3.91 | 10.99 | 0.20 |
| Reynolds number | 294 | 183[*] | 699 | 294 | 560 | 73.5 | 676 | 378 | 141 | 86.3 | 2452 | 6902 | 911 |
| Total surface area/volume (m$^{-1}$) | 8.40 | 4.00 | 20.00 | 68.4 | 20.00 | 8.40 | 8.40 | 8.00 | 8.40 | 24.69 | 4.20 | 4.20 | 0.57 |
| Calculated residence time in inlet (s) | 0.05 | 1.35 | 0.88 | 6.83 | 1.15 | 6.19 | 0.93 | 1.88 | 5.87 | 1.39 | 2.17 | 0.15 | 1.47 |
| Operated with a filter | N | No[1] | Yes | No | Yes | Yes | Yes | Yes | Yes | Yes | No | No | No |
| Heated inlet | No | Yes | No | No | No | No | Yes | No | No | No | No | No | No |

[1] An inertial inlet instead of a filter was used to remove particles from the air stream. It is estimated to be about 90 % efficient, depending on the particle size. Refer to Section 2.2.3 and Roscioli et al., (2016) for further details. [*] After the critical orifice assuming pressure of 100 Torr with at flowrate of 10.44 l m$^{-1}$.

### 2.2.1 Wet chemistry methods

During this campaign, three wet chemistry instruments, which convert gas phase $NH_3$ to aqueous $NH_3$ ($NH_4^+$) for online analysis, participated in the field campaign: a Monitor for Aerosols and Reactive Gases (MARGA, Metrohm NL) and two AiRRmonia (Mechatronics B.V., NL) instruments.

### MARGA

The MARGA (Metrohm, NL) is a method used to measure both the gas phase of several water-soluble species ($NH_3$, HCl, $HNO_3$, HONO and $SO_2$) as well as their aerosol counterparts ($NH_4^+$, $Cl^-$, $NO_3^-$ and $SO_4^{2-}$) and base cations ($Na^+$, $K^+$, $Ca^{2+}$, $Mg^{2+}$) by online ion chromatography. The gas phase species that are water soluble, including $NH_3$, are sampled using a wet rotating annular denuder (WRD), through which air is drawn and the gas diffuses into a continuously exchange liquid film on the surface. Water soluble aerosols do not have sufficient time within the denuder to diffuse into the liquid film and instead are then drawn into a steam jet aerosol collector (SJAC, Khlystov et al. (1995)), where they undergo rapid growth in a steam chamber and are then mechanically separated out by a cyclone. Both the liquid from the WRD and SJAC are continuously drawn by syringes and sequentially analysed by ion chromatography. The cation chromatography was set-up with a 500 µl loop and as a result the detection limit for $NH_3$ has previously been reported to be 0.05 µg m$^{-3}$ (0.72 ppb at 25 °C at STP). An instrument blank was undertaken but it was not subtracted from the reported concentrations. A more detailed description of the instrument can be found in Makkonen et al. (2012). During the campaign the instrument's inlet had a PM$_{2.5}$ cyclone (URG Inc. USA). The inlet sampled at a rate of 16.7 l min$^{-1}$. Due to limited space within the mobile laboratory and the positioning of the MARGA, the positioning of the instrument resulted in a longer inlet with a length of 8.46 m, which is atypical compared to other studies (Makkonen et al., 2012; Twigg et al., 2015; Stieger et al., 2018).

**AiRRmonia**

The AiRRmonia (Mechatronics B.V., NL) is a wet chemistry instrument based on $NH_4^+$ analysis using a selective diffusion membrane / conductivity method (Erisman (2001). Sampling is carried out by drawing air over a Teflon diffusion membrane where gas-phase $NH_3$ diffuses into ultra-pure water, which is in counterflow to the air sample. The sample is then mixed with a sodium hydroxide solution, which forces the liquid $NH_4^+$ back to the gas phase so that diffusion can occur across a second Teflon membrane into ultrapure water. The conductivity of the water and sample are measured to derive a temperature corrected concentration of $NH_4^+$ from which the $NH_3$ gas concentration can be derived. The sample is continuously drawn using syringe pumps providing a constant liquid flow rate. The two AiRRmonias instruments were calibrated together at the start and end of the trial using liquid $NH_4^+$ standards ranging from 0 to 500 ppb. The limit of detection has been reported as 0.08 - 0.1 $\mu g\ m^{-3}$ (equal to 0.114 – 0.142 ppb at STP @ 25°C) and an operational accuracy of 3-10 % (Erisman, 2001; Norman et al., 2009). In this study there were differences in the reporting resolution and inlet set-up between the two AiRRmonias instruments (refer to Table 1 and Table 2 for further details).

**ALPHA® samplers**

During the campaign passive samplers, Adapted Low-cost, Passive High Absorption diffusive samplers (ALPHA®), UK) were placed in triplicate (1.7 m height) at 3 positions along a transect at 3.5 m, 10.5 m and 17.5 m measured from the scaffolding, to investigate the homogeneity between the miniDOAS instruments and the reflectors (refer to Figure 1). The ALPHA sampler is a diffusion badge type device with a citric acid coated filter. The ALPHAs were exposed in triplicate, with a rain shelter, at each position for two periods; Period 1: 22/08/2016 16:35 to 29/08/2016 16:29 and Period 2: 29/08/2016 16:29 to 05/09/2016 17:42. Chemical analysis was performed using an AMmonia Flow Injection Analyser (AMFIA; ECN, NL) which deployed the same analytical principle as the AiRRmonia. An uptake rate of 0.00324 $m^3\ hr^{-1}$ was established by comparison with a local active sampler (UKCEH DELTA®, UK). The preparation, deployment and analysis followed the EN17346 standard methodology (CEN, 2020). Further details of the theory of the passive sampler can be found in Tang et al. (2001) and showed an expanded uncertainty of < 11.6 % for concentrations ranging from 1 to 23 $\mu g\ m^{-3}$ in a recent exposure chamber study (Martin et al., 2019).

### 2.2.2 Cavity ring down spectroscopy (CRDS)

Cavity ring down (CRD) instruments utilise the near infrared region and use an optical cavity to increase the pathlength and thereby to improve sensitivity in measuring the absorption. The laser is periodically turned off to allow the light to decay as it leaks out of the cavity through the mirrors. This happens as the beam is reflected multiple times off the mirrors within the cavity resulting in a large pathlength. When an absorbing gas is added to the cavity the mean lifetime of the beam decreases, and the absorption coefficient can be obtained from the measured ring-down times. The concentration is calculated from the 'ringdown time', which is the time it takes for the light to decay to 1/e of its original intensity. During the campaign there were three instruments that used this analytical technique.

**Picarro G2103 Analyzer (Picarro)**

The Picarro G2103 Analyzer (Picarro, US) uses the CRDS technique. The gas temperature and pressure are kept constant in the cavity, at 45 °C and 140 Torr (corresponding to ~187 hPa), respectively. The analyser uses a tuneable NIR diode laser as a light source, which is scanned over multiple, isolated data points inside the spectral window from 6548.50 to 6549.25 $cm^{-1}$, which includes several $NH_3$, $H_2O$ and $CO_2$ absorption lines. A cross-sensitivity to $H_2O$ and $CO_2$, originating from the overlapping absorption lines of the three molecules, is effectively eliminated by using empirical correction functions as

outlined in Martin et al. (2016). During the campaign, two of these instruments were operated (Picarro #1 and #2), however , this correction was not yet released by the manufacturer at the time of the field study and thus was not yet implemented in the participating instruments. The reported detection limit from the manufacturer for this instrument is 0.09 ppb. In this study, Picarro #1 relied on an external pump with a sampling rate of 0.8 l min$^{-1}$, whereas Picarro #2 utilised an external pump with a

sampling rate of 1.35 l min$^{-1}$ (refer to Table 1 and Table 2). The Picarro #2 instrument was also used as an optical gas standard (OGS) as described in Sect. 2.3.1.

**Tiger-i 2000 (Tiger optics)**

The Tiger-i 2000 (Tiger Optics, US) analyser also uses the CRDS technique. Like the Picarro G2103, it utilises a tunable

continuous wave (CW) NIR diode laser. The instrument is configured to deliver concentration measurements of $NH_3$ in the ppb regime and with regular maintenance prescribed by the manufacturer, the system should not in theory require calibration. The manufacturer states that Tiger-i is able to measure trace $NH_3$ in ambient air without effects from varying humidity levels, or from potentially interfering molecules, due the high specificity of the CRDS technology. During the campaign the instrument was configured to have a detection limit of 10 ppb.

### 2.2.3    Quantum Cascade Laser Absorption Spectroscopy (QCLAS)

**Mini-TILDAS Ammonia Monitor**

The Mini-TILDAS Ammonia Monitor is a Quantum Cascade Laser Absorption Spectrometer (QCLAS) produced by Aerodyne Reasearch Inc. (Billerica, USA) and is provided with an inertial inlet. Due to the instrument being reported already in the

literature (Whitehead et al., 2008; von Bobrutzki et al., 2010), it is referred to as the QCLAS during this study in order to limit confusion. Air was sampled at 13 l min$^{-1}$ through a quartz siloxyl coated inertial inlet (removing particles >300 nm from the air stream) followed by a 3 m Perfluoroalkoxy (PFA) tube, both of which were heated to a temperature of 40˚C, based on the design of Roscioli et al. (2016), though no passivation was used. The QCLAS uses an Astigmatic Multi-pass Absorption Cell (AMAC) with a pathlength of 76 m (volume 0.5 l and 30 Torr) and a continuous wave mid-infrared quantum cascade laser

operated at 966.814 cm$^{-1}$ during this campaign (Roscioli et al., 2016), and a thermoelectrically cooled detector. Substraction of the background spectrum was performed every 30 minutes with dry research grade nitrogen (BOC, Product 293679-L, 99.9995 % $N_2$ min) for 30 s. The manufacturer reported detection limit for this instrument is 0.05 ppb. Although the instrument can be operated at 10 Hz for eddy-covariance flux measurements, here it sampled at 1 Hz to increase sensitivity and reduce data volume.

### 2.2.4    Off-Axis Integrated Cavity Output Spectroscopy (OA-ICOS)

**GLA331-EAA Enhanced-Performance Economical $NH_3$ Analyser (LGR)**

The GLA331-EAA Enhanced-Performance Economical $NH_3$ Analyser (ABB-Los Gatos Research, US) uses the Off-Axis Integrated Cavity Output Spectroscopy (OA-ICOS) technique.  The LGR instrument uses either an internal 2-head or external

3-head diaphragm pump (Table 1) to continuously draw air through a ¼" PTFE inlet tube into the cavity for measurement, pressure controlled to maintain a pressure of 100 Torr. The OA-ICOS cavity is a cylindrical two-mirror design with the gas inlet and outlet at either end, and sensors for gas temperature and pressure are inserted via ports in the middle of the cavity. A fibre-coupled, continuously scanned ~1.7 μm diode laser is directed into the gas inlet side of the cavity, and a wideband IR detector with collimating lens covers the mirror on the gas outlet side of the cavity. Although the cavity mirrors are highly

reflective (>99.99 %), a fraction of the light directed into the cavity will "leak" on each pass, allowing the collection of a resolved, continuously scanned absorption spectrum which forms the basis of the measurement. The laser is pulsed to produce

wavelength scans at several hundred Hz, which are then integrated to provide 1 s real-time data. It is able to achieve a detection limit of 0.3 ppb at 100 s. During the campaign two LGR instruments were used; LGR #1 used its internal pump (0.25 l min$^{-1}$) and LGR #2 used an external pump (2.3 l min$^{-1}$); in addition, the inlet for LGR #2 was heated to stabilise between 40 °C and 70 °C (refer to Table 1 and Table 2 for further details).

### 2.2.5 Optical-feedback cavity enhanced absorption spectroscopy (OF-CEAS)

**ProCeas gas analsyser (AP2E)**

The Optical-Feedback Cavity Enhanced Absorption Spectroscopy (OF-CEAS) uses the principle of absorption spectroscopy. In OF-CEAS, the concentration is based on a scanned wavelength direct measurement of absorption as a function of integrated

10 transmitted laser intensity. For a detailed description of the OA-CEAS refer to Morville et al. (2005). The ProCeas® gas analyser produced by AP2E, Aix-en-Provence, France utilises the OF-CEAS techniques with a high-finesse V-shaped optical cavity made with 3 highly reflective mirros and including a fibered distributed feedback diode laser to operate in the near-infrared at a wavelength of ~ 1.53 μm. It had a reported detection limit 45 ppt (3 sigma, 300 sec). During the campaign, the instrument was operated only with an external pump (refer to Table 1 and Table 2 for further details).

### 2.2.6 Photoacoustic spectroscopy

**NH$_3$-1700 analyser (LSE)**

During this campaign, only one instrument used photoacoustic spectroscopy, which takes advantage of the development of stable quantum cascade lasers in the IR, however instead of measuring the absorption of light it measures an acoustic signal.

The signal is generated as target molecules absorb light of the IR and become excited resulting in a pressure change. The LSE NH$_3$-1700 analyser (LSE) by LSE Monitors, Netherlands uses this method by modulating the laser at an acoustic frequency of 1600 Hz and the resultant pressure modulation is detected by a microphone. By scanning the laser over a specific spectral range, the gas of interest can be determined by the recorded microphone signal. It has a detection limit of 1 ppb.

### 2.2.7 Mini Differential Optical Absorption Spectrometer (miniDOAS)

Differential Optical Absorption Spectroscopy, or DOAS, retrieves the concentration of a trace gas from its characteristic fingerprint in an optical spectrum in the ultraviolet spectral range, refer to Platt et al. (2008) for a thorough discussion of this method.

The two systems taking part in this campaign were miniDOAS #1 developed by the Bern University of Applied Sciences, Switzerland, in collaboration with Neftel Research Expertise and miniDOAS #2 developed by the Dutch National Institute for Public Health and the Environment (RIVM), Netherlands. The systems were of a similar set-up. Each system uses a UV-lamp to generate a light beam. The beam is reflected back to the instrument by a retroreflector placed at a distance of 22 m, creating an optical path of 44 m. The light is collected by a telescope and measured with a low-cost compact spectrograph. A adjustable

mirror corrects for small changes in alignment of the set-up. Measured spectra are averaged over a period of typically 1 minute. Whilst the closed-path instruments described above work at low pressure and reduce line broadening so that they can distinguish different absorption lines for different compounds, the open-path nature of the DOAS necessitates the NH$_3$ concentration to be retrieved from an averaged spectrum along with concentrations of SO$_2$ and NO, which also have optical absorptions in the wavelength range used (205-230 nm), using the DOAS inversion algorithm.

Both systems were designed and built at their respective institutes. They are described in detail elsewhere: in Sintermann et al. (2016) for miniDOAS #1 and in Volten et al. (2012) and Berkhout et al. (2017) for miniDOAS #2. The most important differences between the systems were:

- In miniDOAS #1 uses a deuterium lamp, and miniDOAS #2 a xenon arc lamp. Because a xenon lamp emits much visible light, miniDOAS #2 uses an interference filter to block this part of the spectrum, miniDOAS #1 does not require a filter.
- The spectrograph in miniDOAS #1 is peltier-cooled, the one in miniDOAS #2 is not.
- Although both instruments are housed in temperature-controlled boxes, the temperature of miniDOAS #1 is better stabilised than that of miniDOAS #2.

Calibration of the systems took place in the laboratory (Sintermann et al., 2016 and Berkhout et al. 2017), before deployment at the field site. The lamp reference spectra used were obtained from the 61 spectra with the lowest $NH_3$ concentrations measured during the campaign. The reference spectra are the baseline, the DOAS concentrations are calculated as the difference to this concentration. So they can also be negative. During this campaign, the instruments were placed side-by-side on a scaffolding (see Figure 1). Their optical paths ran at 1.78 m above ground. Because the optical paths are in the free atmosphere, no delay or interference from inlets, filters or surfaces can occur. This means the measurement is not affected by temporal averaging beyond the integration time, but note that the concentration retrieved by a DOAS is an average over the entire optical path, this is to be taken into account when comparing results to instruments that sample air from a single inlet point. Since this campaign, significant improvements have been made to miniDOAS #2, especially in the handling of the spectrograph dark current and in stabilising the optical alignment (Swart et al., 2022).

## 2.3 Metrological developed components

As part of the study, metrological methods developed under the MetNH3 were evaluated under field conditions, which were used to estimate the accuracy of LTMHTR instruments but not used to calibrate any instruments.

### 2.3.1 Optical Gas Standard (OGS)

An optical gas standard (OGS) is an instrumental transfer standard concept that doesn't require initial or repetitive calibration using calibration gases. Instead, an OGS determines absolute concentrations based on first-principles, i.e. a full physical model of the absorption process. Here, the measured absorption in a sufficiently spectrally isolated ro-vibrational transition of a small molecule like $NH_3$ is described by Beer-Lambert's law, an analytical absorption line shape model, and molecular spectral parameters like the absorption line strength. The OGS concept is explained in Nwaboh et al. (2021), Nwaboh et al. (2017) and Qu et al. (2021). Buchholz et al. (2014) rigorously validated the calibration-free property of an OGS for the case of $H_2O$ by cross-comparing the $H_2O$-OGS named SEALDH with PTB's primary gas humidity standard. An OGS thus can serve as a field transfer standard and be used to calibrate and validate other instruments. In this study, the Picarro#2 CRDS instrument operated by PTB was converted into an OGS, by extracting and refitting the raw CRDS absorption spectra. The OGS essentially extracts and re-evaluates the Picarro raw spectra, hence it uses the same hardware but a completely different evaluation and different spectral reference. To this end, it was fully metrologically characterised in the German national metrology institute, Physikalisch-Technische Bundesanstalt (PTB), i.e. the accuracy of the temperature sensors, pressure sensors and of the spectral scale (wavenumber) was verified by comparison to SI-standards. Furthermore, a custom spectral fitting algorithm using accurately measured spectral line parameters (Pogány et al., 2021) was developed and employed by PTB. An expanded uncertainty of 1% could be achieved for the line intensity of the two strongest $NH_3$ lines, which allowed the total uncertainty of the retrieved $NH_3$ concentration to be decreased down to 3% (k=2, 95 % confidence interval). Further important contributors

to this uncertainty are spectral line broadening coefficients or the choice of the fitted spectral model. Due to this full physical model the need for empirical, calibration-based instrument corrections, e.g. to compensate spectral interferences (Martin et al., 2016) was eliminated. As a result, traceable and absolute $NH_3$ concentrations were obtained. .

### 2.3.2 Permeation calibration system

Ammonia calibration in the field is difficult due to the adsorptive nature of $NH_3$ resulting in losses to inlet and surfaces of both the calibrator, tubing and instruments, associated with long stabilisation times to achieve equilibrium, and uncertainty of absolute concentrations (Vaittinen et al., 2014, 2018). A metrological traceable source was developed under laboratory conditions in the framework of the EMRP MetNH3 project. The campaign was a means to determine the applicability of the system in the field, to determine the accuracy of measurement instrumentation under field conditions, and thus to allow for comparability of the results. The traceable source was a dynamic calibration system known as ReGaS (Reactive Gas Standard, Pascale et al. (2017)), developed and constructed by the Federal Institute of Metrology (METAS), Switzerland. For this campaign only the ReGaS1 was applied in the field. The ReGaS1 reference gas generator was developed to dynamically generate SI traceable reference gas mixtures with very low levels of uncertainty ($< 3$ %) in the 0.5 - 500 nmol mol$^{-1}$ range (0.5-500 ppb). It employs as the $NH_3$ source a permeation device in a temperature-controlled oven and two dynamic dilution steps with mass flow controllers to obtain the required amount fractions using zero grade synthetic air (SA) (158283-L-C, BOC). Additionally, a commercially available gas purification cartridge (Microtorr, model MC 400-203V SAES Getters, Pure Gas Inc.) was used for additional synthetic air purification. According to the product specifications, the outflow of purified SA should contain less than 100 pmol/mol $H_2O$ and less than 100 pmol/mol $CO_2$. The content of acids, bases, organics and refractory compounds in the outflow should not exceed 10 pmol/mol. The Microtorr purification system is based on inorganic sorbent materials and operates at normal ambient temperature (no heating or cooling required). The connectors of the cartridge are made of stainless steel. ReGaS1 is transportable to allow for in-situ calibration of $NH_3$ instrumentation. A SilcoNert2000 coating has been applied to all interior surfaces of ReGaS1 in contact with $NH_3$ in order to reduce adsorption effects and thus, stabilisation times. During the calibration, the ReGas1 was connected to a Teflon 6 port manifold using ¼" PFA which was connected to a 3-way valve and T-piece that had been coated in SilcoNert2000.

The instruments that were evaluated against the ReGaS (i.e. LSE, Picarro #2, LGR #1, LGR #2 and Tiger Optics) were transferred from the Pyrex manifold to the Teflon manifold for this purpose. Due to the maximum flowrate of the ReGaS1 (5 l min$^{-1}$) the LGR #2 did not use its external pump but was reliant on the internal pump of the instrument, so had a flow rate of 0.25 l min$^{-1}$ which equates to a residence time of 6.83 s for the inlet. The system was set for the following concentrations in sequence for the duration of 31 minutes each: 0 ppb, 9.98 ppb, 24.39 ppb, 39.71 ppb, 2.95 ppb and 1.02 ppb. Unfortunately, the data of following instrument was excluded from the analysis; the LGR #1 concentrations remained low even at elevated concentrations indicative of a fault and the Tiger Optics reported 0 ppb as it could not detect concentrations below the 10 ppb detection limit. As a result in this study only information from the OGS, LSE and LGR#2 are evaluated against the ReGaS.

### 2.3.3 Gas cylinders

Stable traceable Primary Standard Gas Mixtures (PSMs) of $NH_3$ were developed in order to improve the current state-of-the-art metrological traceability and validation of $NH_3$ instrumentation by the UK's national metrology institute, the National Physical Laboratory (NPL). The PSM employed in this work was prepared gravimetrically using the method outlined in guide (ISO (International Organisation for Standardisation), 2001) from pure ammonia (Air Products, VLSI, 99.999 % purity) and nitrogen (Air Products, BIP +, 99.99995 % purity). Full details of the preparation of the cylinders can be found in Martin et al.

(2016). During this study PSM cylinder number 1825R2, which contained 99.78 ppm $NH_3$ in $N_2$ was used to calibrate the miniDOAS#2 instrument.

### 2.4 Data analysis

For each instrument the data quality assurance (QA) procedures, where applicable, are outlined in the Method Section for each instrument. For the AiRRmonia, MARGA, QCLAS and the miniDOAS instruments, the zeros and/or calibration standards used are described in Sections 2.2.1, 2.2.3 and 2.2.7, respectively. They were applied to the data sets prior to undertaking the data analysis presented in Section 3. No other LTMHTR instrument had any zero or calibration applied, as the instrument manufacturers described the methods as 'calibration free' at the time of the study, so were only operated with the manufacture factory calibrations. Data which did not meet the QA was not included in the analysis, further details are found in the Section 3.2. Measurements provided in units of $\mu$g m$^{-3}$ were converted to parts per billion (ppb) using the temperature and pressure measured at the Easter Bush site. To facilitate direct comparisons, data were averaged to 1 hour, unless stated otherwise, to match the reporting time of the slowest instrument. The data analysis assumed that instruments "received" or "saw" the same concentrations in the field. Efforts were made to remove likely periods of inhomogeneity during the data analysis (refer to Sect. 3.5), however instruments which did not share a common inlet will not have received exactly the same concentrations at all time (Table 1). This is specifically an additional consideration for the miniDOAS instruments that measure a line average concentration (22 m) rather than sample at a point. Though instruments were deployed for a longer period, for the purpose of this study only the period of the 23rd to 29th August is studied unless otherwise stated, as not all instruments were operational at the start and end of the campaign.

## 3 Results

### 3.1 Meteorology and background aerosol composition during the campaign

Figure 2 summarises the meteorology (wind speed, wind direction, temperature and relative humidity) for the period studied. The cumulative rainfall during the campaign was atypical for the site, 2.8 mm compared to averages of 98 mm for the month of August (2005 – 2014). Though the site was unusually dry, the average temperature of 14.3˚C was typical (climatological average 14.03 ˚C, 2005-2014) for August in South East Scotland with temperatures ranging from 7.8˚C to 20.6˚C. As expected for this time of year the predominant wind direction was from the SW.

As well as reporting $NH_3$ gases, the MARGA also reported the PM$_{2.5}$ water-soluble inorganic species. Prior to fertilisation on the 23rd August PM$_{2.5}$ was dominated by sea salt (NaCl), but during the interim was dominated by secondary inorganic aerosol, which coincided with a drop in wind speed and a reduction in the relative humidity on the 24th August.

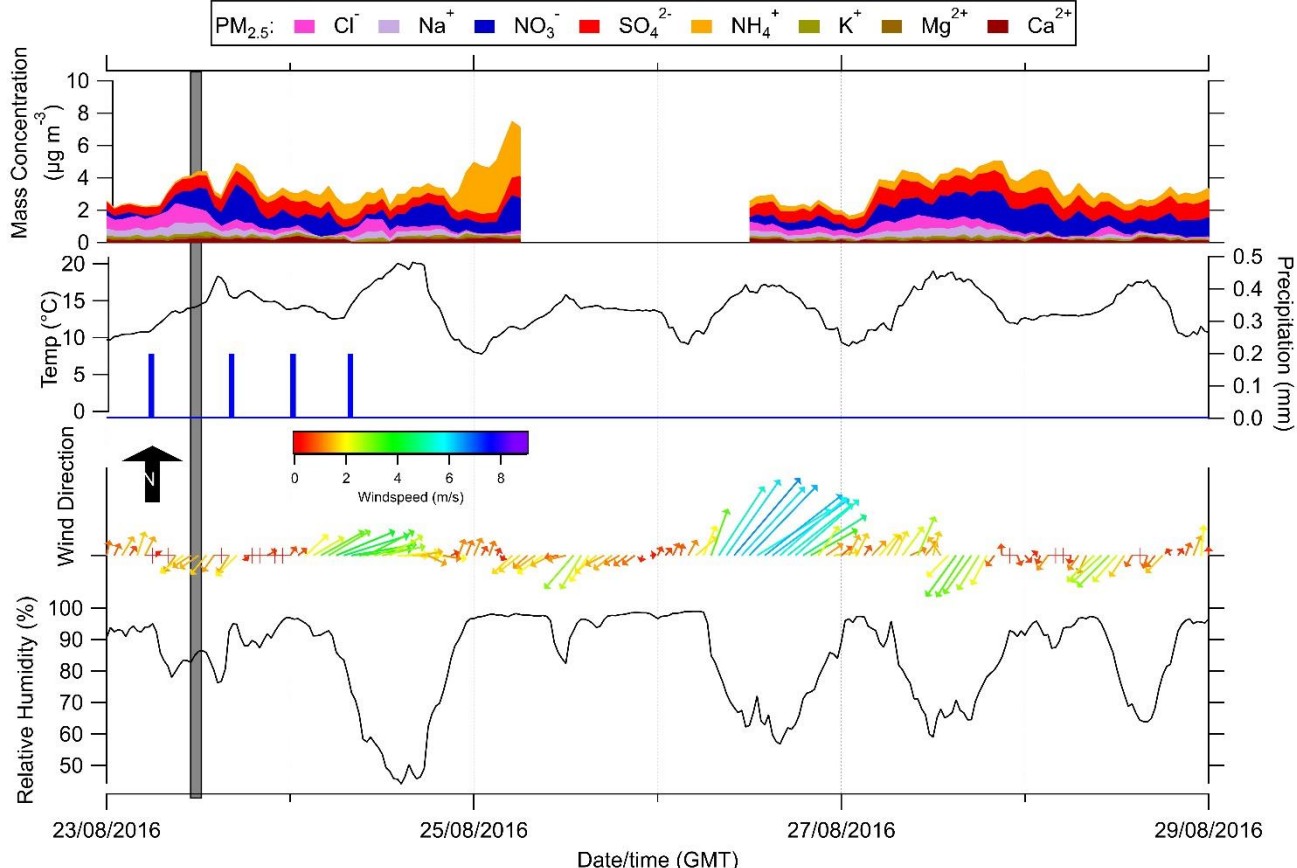

**Figure 2. Summary of the meteorology and the inorganic composition of water soluble PM2.5 at Easter Bush during the intercomparison campaign from the 23 to 29 August 2016. The grey shaded line is the period where urea was applied to fields. Blue bars are precipitation and black line is the temperature.**

### 3.2 Overview of the NH₃ measurements during the campaign

The time series of the measurements by the instruments at their reporting temporal resolution (1s to 1 hour), unfiltered, are summarised in Figure 3 (Table S1). Instruments display similar temporal features for NH₃ concentrations over the duration of the study, though there are differences in their structures due to differences in the reporting and measurement resolution (refer to brackets in legend of Figure 3). The maximum NH₃ concentration observed was on the evening of the 24th August following fertilisation on the 23rd August (Figure 3). It is likely that the emission of NH₃ was suppressed following fertilisation due to intermittent precipitation during the 23rd August (Figure 2) and instead the peak in NH₃ concentrations observed on the following evening due to stable (Figure S2) and dry conditions (Figure 2). The LSE instrument reported the highest concentration with a maximum of 282 ppb (1 minute average). The concentrations reported by all the instruments following fertilisation were large compared to the von Bobrutzki et al. (2010) study which reported maximum concentrations of 120 ppb, though the same amount of urea was applied to the same field. The difference in meteorological conditions during the von Bobrutzki et al. (2010) study is likely to have impacted NH₃ emissions, where the site received a high volume of rain resulting in the formation of a pond in the North field, whereas this study was relatively dry (Figure 2).

Many temporal features can be picked out as the concentrations change throughout the field campaign (Figure 3) however, to have a brief look at instrument response, the response on the 25/08 at 04:00 is discussed for each panel in Figure 3. Panel a) presents the time series for each of those instruments that were using their own inlets during the campaign. It was observed that the QCLAS had a faster decrease in concentration compared with the other instruments using their own inlets, at 04:00 (GMT) on the 25/08, as the wind direction changed to a north easterly direction (Figure 2). The delay in the time response of the other instruments is likely to be due to instrument set up with long inlets and low airflow rates (Table 2). The delay in the MARGA is also likely to be due to both the reporting interval (1 hour average), as well as the atypical inlet length. Instruments on the manifold (Figure 3, Panel b) did not show the delayed response following the change in wind direction. The exception

to this is the LGR#1 (but not LGR#2). LGR#1 was reliant on its internal pump to sub sample from the manifold, and as a result had a lower sample flow rate resulting in a slower response time (Table 1). The Tiger Optics instrument was set up in a configuration with a 10 ppb limit of detection and, following post-campaign data analysis, only the period of 23/08 20:00 to the 26/08 11:00 was valid and is presented. The LGR#1 was reporting 0 ppb $NH_3$ initially and a laser fault was identified by

the operators. The fault was corrected remotely by the manufacturer on the 24/08 at 10:00 (GMT) (Refer to the arrow on Figure 3). Following this there is an apparent improvement in agreement of LGR #1 compared to the other instrumentation on the manifold (refer to the arrow on Figure 3). Therefore, only data after the 24/08 10:00 is used for the LGR#1 for the remainder of this study. Instruments in the campaign situated on scaffolding (Figure 3 Panel c) were either open path or had a very short inlet. The AiRRmonia #1 though reporting at the same temporal resolution as the miniDOAS instruments (1 min averages),

did not capture the same temporal features, demonstrating a slower instrument response time. This is not surprising since it was previously reported that the AiRRmonia had a time response of 14 ±4 mins in von Bobrutzki et al. (2010) study.

As described in Sect. 2.2.1 the ALPHA samplers were deployed along the miniDOAS pathlength, to evaluate the homogeneity of the $NH_3$ during the campaign. Both miniDOAS#1 and miniDOAS#2 compared well to the ALPHA samplers (Figure 4), reporting 11.3 ppb and 10.9 ppb respectively, compared to 10.9 ppb from the ALPHA samplers during period 1. This is within

15 the method error. A summary of the averages from each instrument can be found in Table S2. It is worth noting that during period 1, which is the main focus of this study, though there were large temporal variations in concentrations (Figure 3) the transect of ALPHA samplers reported similar concentrations, suggesting the $NH_3$ concentrations were relatively homogenous spatially. In period 2, however, the miniDOAS#1 appears to report a higher average concentration over the whole period due to a lower data capture of 89 %, for this period.

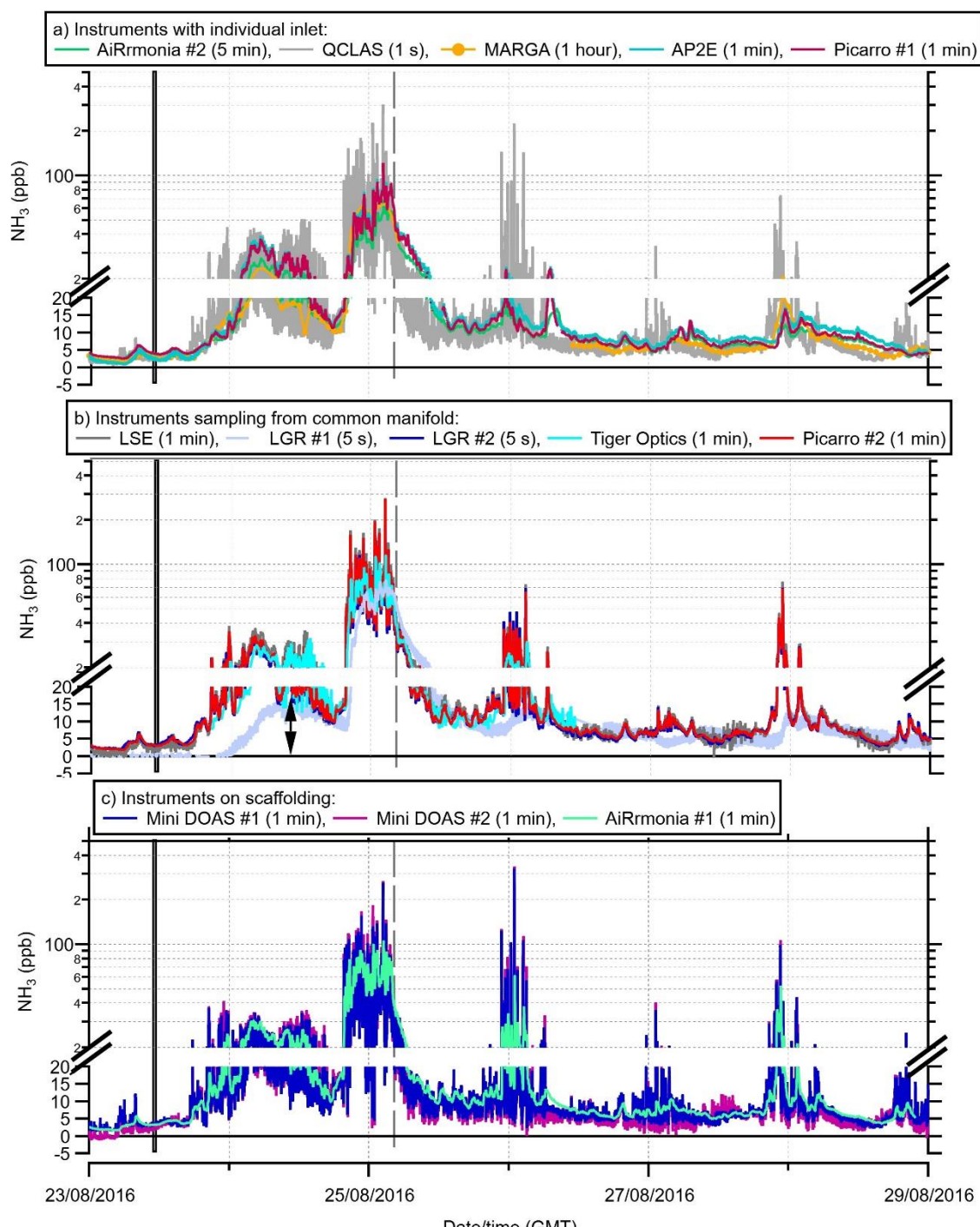

**Figure 3. Summary of the reported concentrations from the instruments divided into categories a) instruments with individual inlet set-up b) instruments subsampling from the manifold and c) instruments on scaffolding. Number in brackets is the reporting time resolution of each instrument. The thick black line is the fertilisation of both fields, the grey dashed line indicates the change in wind direction at 04:00 on the 25/08 and the black arrow indicates the point at which the laser position was changed on the LGR #1.** *Note: The scale changes at 20 ppb to a log scale.* **(Supplementary Figures plotting the data on linear scale can be found in Figure S1).**

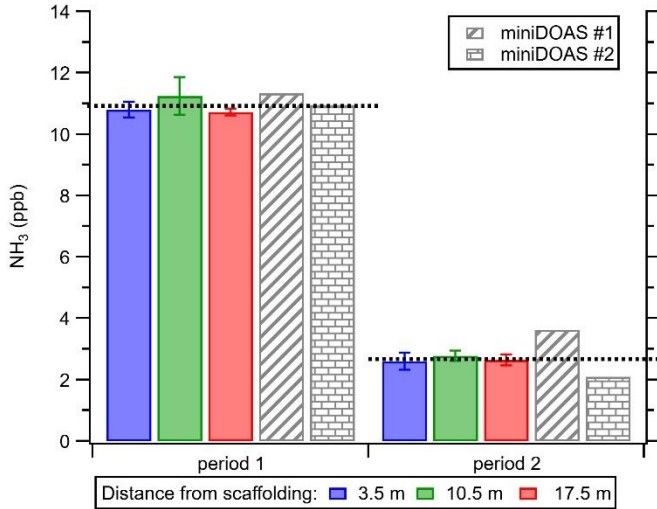

**Figure 4. Average concentrations along the path length of the miniDOAS instruments measured by passive diffusive samplers (ALPHAs) in triplicate at increasing distance from scaffolding. Error bars are ± $\sigma_A$ of the replicates at each position. (Period 1: 22/08/2016 16:35 to 29/08/2016 16:29 and Period 2: 29/08/2016 16:29 to 05/09/2016 17:42). Black dashed lines are the overall average concentration measured by the ALPHA samplers for the period. Data capture for period 1 is summarised in Table S2. Data capture for period 2: miniDOAS#1 = 89 % and miniDOAS #2 = 98 %.**

Across all instruments, though the temporal pattern was comparable, there are large variations in the reported magnitude of concentrations measured, even when data is averaged to an hour (Figure 5). For example, on the morning of the 25/08 02:00 (GMT) when $NH_3$ was elevated the AiRRmonia #2 reports the lowest concentration of 57.2 ppb, whereas AiRRmonia #1 reported a concentration 66.8 ppb. The highest concentration, reported at this point, was by the LSE with 88.5 ppb. These extreme values are a function of the averaging time and the response time of the instrument. A faster instrument naturally shows larger extreme values and the $NH_3$ adsorbed to inlet walls has the potential to desorb during subsequent hours. On the longer-term average the instruments that covered the period 23 to 29 August with high data coverage ($\geq$ 98 %) agreed within +/- 15 % of the overall mean (Table S1).

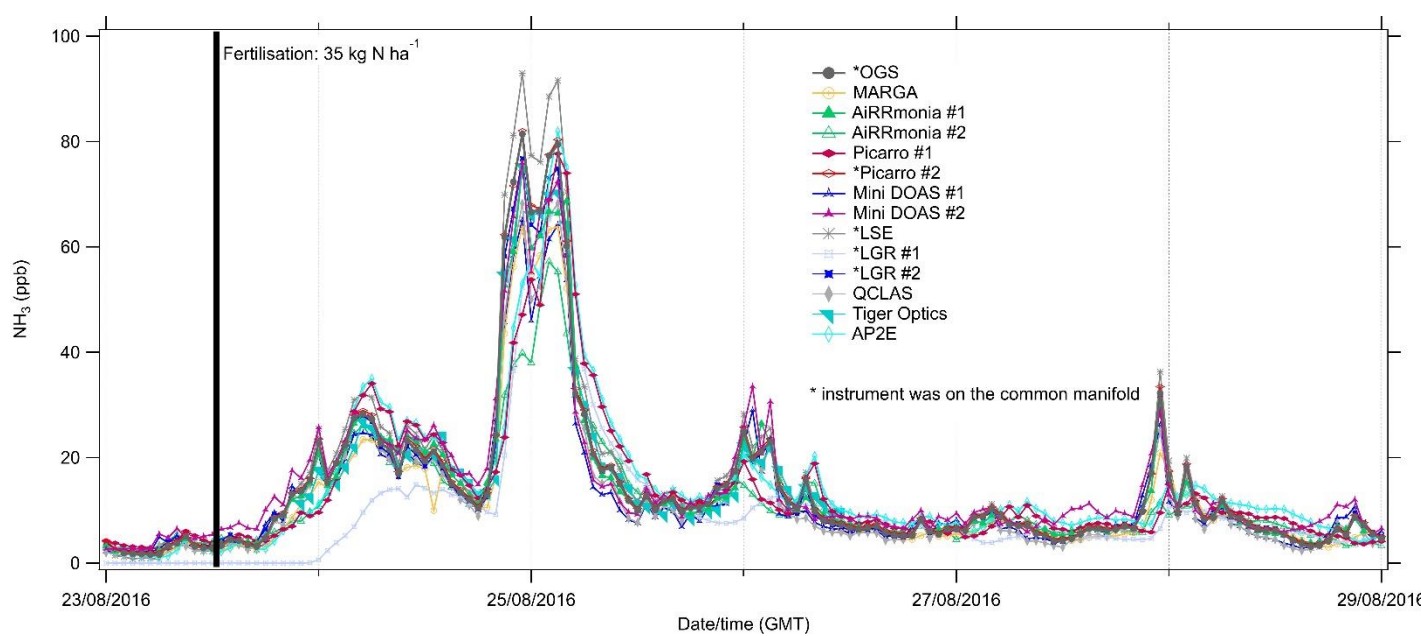

**Figure 5. Time series of hourly averages from the 23/08/2016 to 29/08/2016 of $NH_3$ measurements at Easter Bush. The shaded area is the period of fertilisation using urea.**

### 3.3 Precision across the suite of instrumentation

To assess the precision across the suite instruments during the campaign the coefficient of variance was studied (CV, Equation 1). As a guidance, the US EPA accepts a CV of up to 10 % for PM sensors, and up to 15 % for $NO_2$ monitors (EPA, 2012; Sousan et al., 2016; Crilley et al., 2019), an increase beyond this range suggests a worsening of the reported precision, where currently there is no guide for $NH_3$. For the purposes of this study a CV limit of 20 % was set. The CV (%) is calculated using the following equation:

$$CV = 100 * \frac{\sigma}{\mu}$$

Equation 1

where $\sigma$ = standard deviation and $\mu$ = mean for the measurement of the hourly average reported by the reporting instruments in each period. Figure 6 summaries the CV compared to the hourly average reported concentration by the ensemble median during the campaign. It observed that the ensemble CV varies between 10 and 50 %. On the 23/08 the CV is high (>20 %), which matches a period of low $NH_3$ concentrations (<10 ppb). The CV then suddenly drops (< 20 %) at around midnight on the 24/08 coinciding in an increase in the average concentration (>10 ppb). At 20:00 on the 24/08 there is a spike in the CV as there is a rapid increase in the $NH_3$ concentration but then drops again at 22:00 as the concentration remains elevated. It is postulated that the loss in agreement between instruments during this period is due to the different response times to a rapid change in concentration. The same reduced precision is also observed when the $NH_3$ concentrations decrease on the 25/08, which again is likely due to the differing response times of instrumentation.

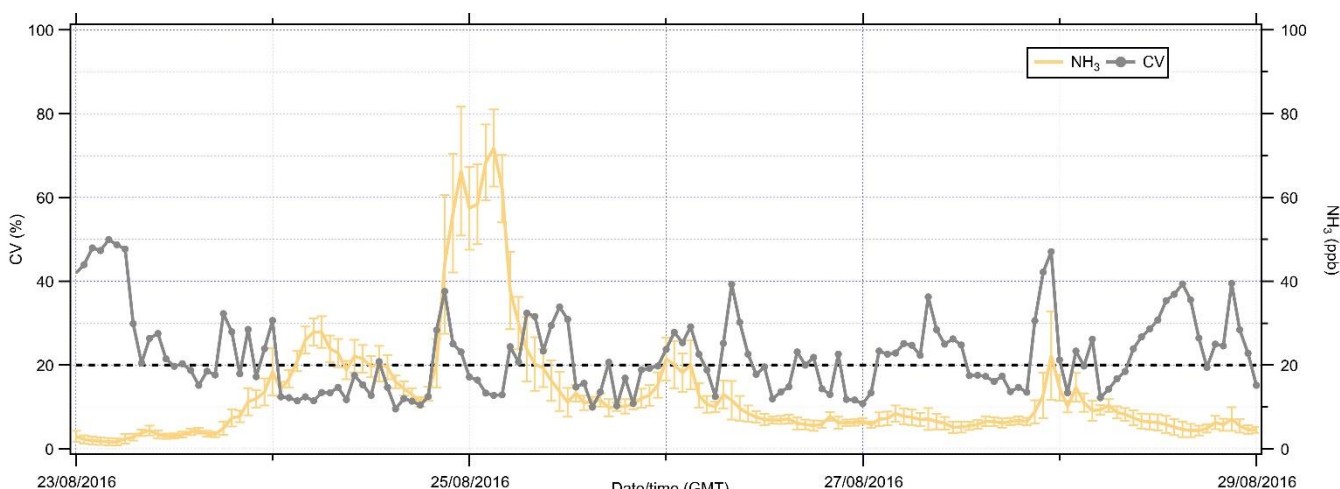

**Figure 6. Time series hourly coefficient of variance (CV) and the average reported $NH_3$ concentration (ppb), where error bars are ± $\sigma_A$. The black dotted line is a CV limit of 20%.**

### 3.4 Effect of inlet set-up on response time

There were a number of inlets used during the campaign and this is hypothesized to have affected the concentrations received by the instruments, which is especially apparent at lower concentrations in the CV of the suite of instruments. To study the inlet design impact on time response the two collocated instruments of the same model, the Picarro and AiRRmonias, were studied as the operational difference was the instrument and inlet set-up. The LGR comparison was excluded due to the poor performance of the LGR#1 (refer to Sect. 3.2 for further details). The time response of each instrument was calculated based on the response of the miniDOAS #1, as it does not have an inlet and therefore assumed to have an immediate response to changes in concentration. It is assumed that any differences in time response is due to adsorption/desorption effects. To determine the response of the instruments, the miniDOAS #1 data was smoothed using the running mean on the measured

concentrations ($c(t)$) by adjust its smoothing factor ($f$) until the delayed smoothed concentration ($c'(t)$) matched the data from the slower instrument in each case, based on the same method as von Bobrutzki et al. (2010);(Eq. 5).

$$c'(t) = fc(t) + (1 - f)c'(t - 1).$$

**Equation 2**

The $e$-folding time ($\tau_{1/e}$) was then calculated by $\tau_{1/e} = 1/f$. Figure 7a compares the results of the AiRRmonia #1 and #2 and Picarro #1 and #2 to those of the miniDOAS #1, under elevated concentrations. It is clear that the AiRRmonia #2 has a slower response compared to the AiRRmonia #1, with a 95 % response time from AiRRmonia #1 of 18.4 mins compared to the AiRRmonia #2 with a response time of 372 mins, demonstrating that the presence of an inlet with a low flowrate (1 l min$^{-1}$) leads to a loss of the NH$_3$ temporal features. This is not, however, the only controlling factor for the response of an instrument, as the Picarro #1 inlet is calculated to have a residence time for air of 1.3 s compared to Picarro #2 that has a residence time of 3.6 s (including the manifold inlet and manifold), but it still appears that the Picarro #2 performs better. It is postulated that as the surface area/ volume ratio for the Picarro #1 is two times the surface area/volume ratio of Picarro #2 (Table 2), resulting in more molecules interacting with the inlet walls leading to the observed a smoothed feature. It was discounted that turbulent flow was a controlling factor in the response time, as it would be expected that wall interactions would increase under a turbulent regime leading to greater losses (Table 2).

In contrast, under ambient conditions, the response time of the instruments is reduced (Figure 7b). The AiRRmonia #1 $e$-folding time increased from 6.15 mins to 32.8 mins and similarly the Picarro #2 change from 4.48 mins to 49.5 mins. This is to be expected as the losses of NH$_3$ due to adsorption/ desorption effects of both the inlet and instrument are more apparent as any loses make a greater contribution to the absolute concentrations when at low concentrations.

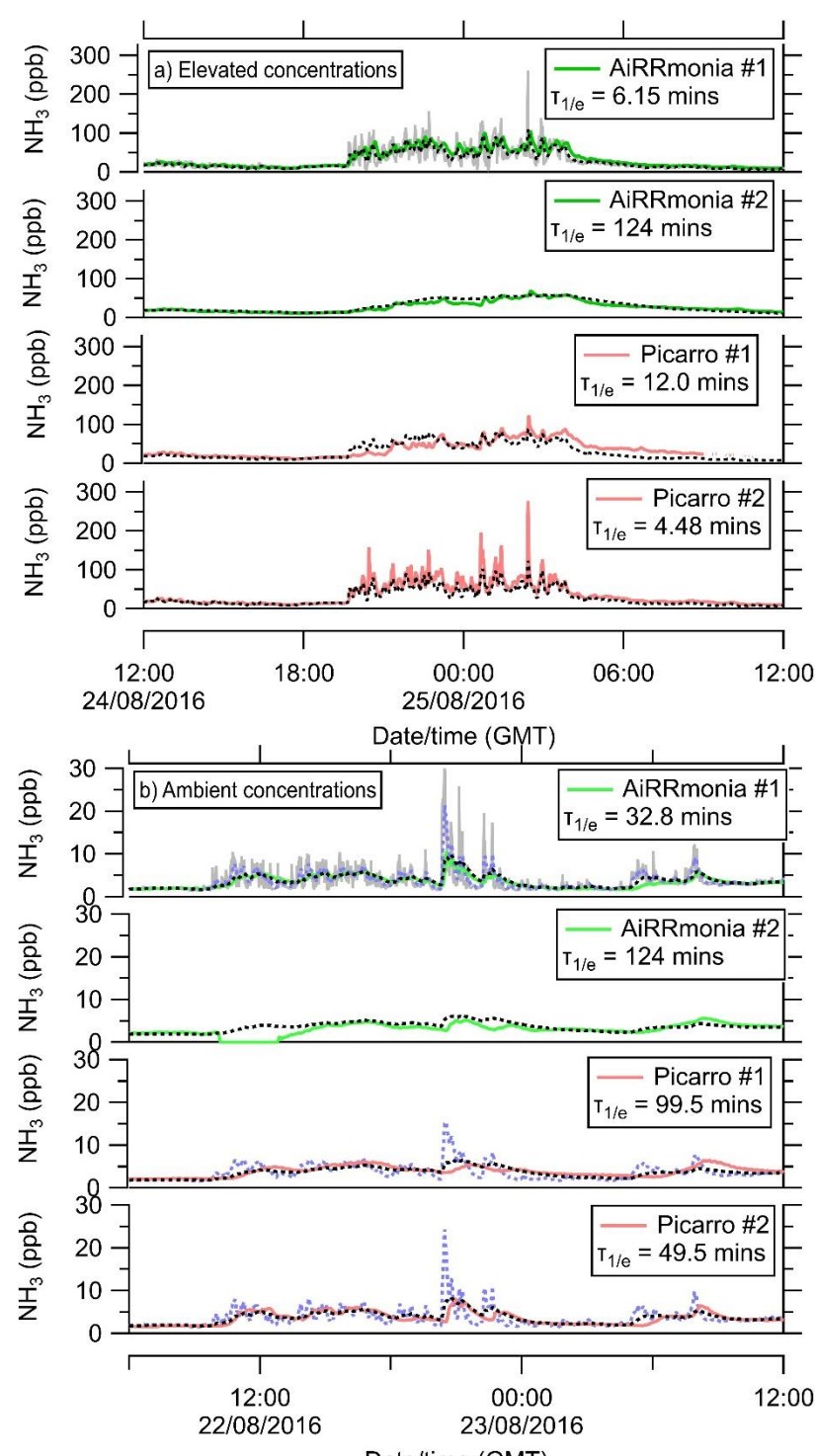

**Figure 7**. **Smoothed time series of miniDOAS #1 (black dotted line) calculated from the 1-minute miniDOAS #1 signal (grey line) until fitting by eye the time series is similar to the reporting data of individual instruments. a) Elevated concentrations following fertilisation (fertiliser applied from 11:00 23/08/2016) b) ambient concentrations. The blue line on panel b) represents the smoothed**

5 **time series using the time response derived from elevated concentrations from panel (1) to visualise the significant additional smoothing encountered under ambient conditions.**

## 3.5 Performance of instrumentation at ambient conditions (<10 ppb)

Though there is evidence of agreement across the suite of instruments at high concentrations, in order to understand the varying

10 performance across the instruments the hourly ensemble median (excludes the Tiger Optics and LGR#1 due operational issues, refer to Sect. 3.2 for details) was split into $NH_3 < 10$ ppb or $NH_3 \geq 10$ ppb, so a direct comparison could be made to the von Bobrutzki et al. (2010) study and to assess performances for lower concentrations without the results being skewed by

individual large concentration points. The intercomparison is presented for the full dataset, however, in Figure S3 in the Supplementary Material.

In the absence of a 'perfect' reference instrument, Figure 8 presents the summary of the instrument comparison to the ensemble median for $NH_3$ <10 ppb It is acknowledged that the ensemble median could be biased if the majority of instruments are biased. The Tiger Optics data was excluded from this analysis as the instrument used during the comparison had a limit of detection of ~10 ppb. For all the data (both the open and closed circles), the majority of instruments have a spread of points around the one-to-one line. Instruments which reported an $R^2 < 0.6$ compared to the ensemble median were the miniDOAS#2, LGR#1, Picarro#1, AP2E and AiRRmonia#2, whereas instruments that reported $R^2 > 0.9$ were the LSE, AiRRmonia#1, Picarro#2 and LGR#2, three of which sampled from the common manifold. To investigate if the differences were due to periods of inhomogeneity in $NH_3$ concentrations at different sampling locations, caused by low windspeed and atmospheric stability conditions, the data was filtered to exclude data when wind speed was < 0.8 m s$^{-1}$ and atmospheric stability was filtered for -0.1 < z-d/L > 0.1 (Figure 8). There was an improvement in the performance of most instruments with reported $R^2$ ranging from 0.71 to 0.98, with the exception of the LGR#1 and the miniDOAS#2, reporting the lowest $R^2$ values with 0.27 and 0.55 respectively, suggesting that these instruments randomly deviated from the ensemble median. It is assumed that the difference between the miniDOAS instruments is due to the stability of instrumentation in regulating temperature, however it is beyond the scope here to interrogate each instruments temperature dependence. It is noted, that the slopes and intercepts changed when applying a meteorological filter. In general instruments with faster response found their slopes reduced, whereas the reverse was observed for the instruments with the slower time response. With the exception of the AP$_2$E, slopes after filtering were closer to unity and with the additional exception of Picarro #2 the intercepts decreased. For the remainder of this discussion however we will only discuss the filtered data.

AMost instruments (Figure 8) had a slope less than 1 with the exceptions of the AP2E, Picarro #1 and the LSE. The largest slope reported was from the AP2E (1.47) and it had the largest negative offset of -1.39 ppb. The y-axis offset is a result of uncertainties in the linear fit, and contamination/losses of $NH_3$ in inlet or the instrument. Interpretation of the intercept is here limited in order to hypothesise regarding the relationship between predicted $NH_3$ (from the ensemble median) and the concentration response of the instruments. Contamination, inlet losses, limits of detection and non-linear instrument response are the major issues which will lead to linear slopes with significant offsets. Negative intercepts are often indicative of losses of $NH_3$ either to the inlet or the instrument, however the large slope and high scatter ($r^2=0.76$) would also be contributing to the offset value. The instrument with the smallest offset is the QCLAS, which had an offset of 0.05 ppb but had a slope of 0.82 compared to the ensemble median. The largest positive offsets are seen in the Picarro #1 (with an offset of 1.05 ppb), miniDOAS #1 (0.74 ppb), LGR #1 (2.11 ppb), LGR #2 (0.65 ppb) and the AiRRmonia #2 (0.75 ppb). Working with the assumption that within the uncertainty of the regression, the positive offsets are real, the positive offsets in this case could be attributed to contamination in the inlet or in the case of the CRDs on the inline filters. For the LGR #2, another possible explanation is that heating the sample line may have resulted in a positive offset due to the volatilisation of $NH_4NO_3$. The (large) positive offset found for the miniDOAS #1 cannot be due to contamination since it is an open path instrument. The two miniDOAS systems reported different offsets at below 10 ppb, as the systems use different approaches to derive the concentrations. The differences between the two instruments can include variation in the spectral fits leading to biases for $NH_3$ or another interfering gas (e.g. $SO_2$, NO), uncertainties in the spectral lines used, or technical issues including alignment, dark current or imperfections in the spectral response of the spectrograph. Identifying the source of the differences between the miniDOAS systems is challenging. A similar positive offset was observed in Berkhout et al. (2017) who compared miniDOAS instrument to an AMOR wet chemistry analyser. It is suggested that the miniDOAS #2 was sensitive to ambient temperature as the spectrometer was not temperature controlled compared to the miniDOAS #1.

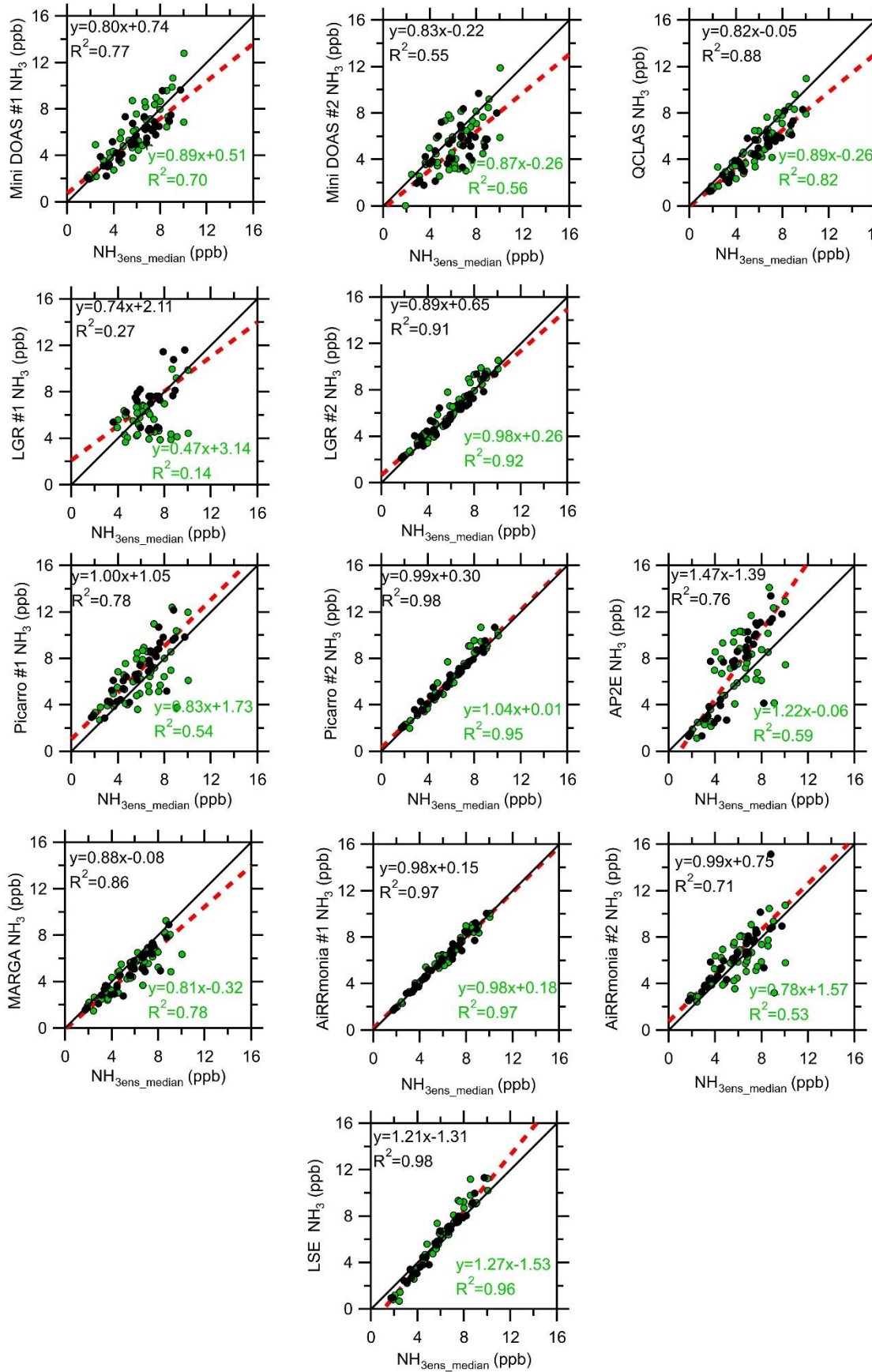

**Figure 8. Intercomparison of hourly instrument averages from 22/08/2016 to 29/08/2016 to the ensemble median (excluding LGR #1 and Tiger Optics) when the median <10 ppb NH₃. Green circles are the data removed after applying a met filter (<0.8 m s⁻¹ and |(z-d)/L| > 0.1). The green and black legends are the correlations of the unfiltered data and the filtered data, respectively. The solid black line is the 1:1 line and red dashed line is the fit.**

### 3.6    Performance of instrumentation at elevated ambient NH$_3$ concentrations ($\geq$ 10 ppb)

Under elevated concentrations of NH$_3$ $\geq$ 10 ppb, filtered for wind speed and atmospheric stability, all instruments demonstrated improved agreement with the ensemble median (Figure 9). The AP2E, Picarro #1, AiRRmonia #2 and the LGR #1 all report an R$^2$ $\leq$ 0.81, whereas all other instruments report a correlation of R$^2$ > 0.95. The instruments reporting a lower R$^2$, with the exception of LGR #1, sampled from the same location but used their own inlets. The same instruments also reported large positive offsets of 4.3 ppb, 2.67 ppb and 2.4 ppb for AiRRmonia #2, AP2E and Picarro #1 respectively. For concentrations $\geq$10 ppb the instruments with a slope greater than 1 are the miniDOAS #2, LGR #2, Tiger Optics, Picarro #2, AiRRmonia #1 and the LSE and are the instruments which have an R$^2$>0.96. The only exceptions to this are the miniDOAS #1, QCLAS and the MARGA that consistently reported a slope less than 1 but report an R$^2$ of 0.97, 0.99 and 0.98 respectively. It is likely that the MARGA would have losses due to the length of inlet used (Table 2). In addition, the capture efficiency of the MARGA of the WRD was limited, at high concentrations of NH$_3$. When the solution becomes more alkaline 'breakthrough' can occur, where NH$_3$ is not captured by the WRD but continued through to the SJAC where the NH$_3$ would be reported as NH$_4^+$ aerosol. To confirm the breakthrough, the ion balance of the PM$_{2.5}$ reported was investigated. It was apparent that at elevated NH$_3$ concentrations there was an excess of NH$_4^+$ aerosol over neutralising anions, which can be attributed to be the breakthrough of NH$_3$ gas from the WRD to the SJAC (Figure S4). This therefore highlights that in the configuration presented, the MARGA is limited in its range of concentration measurements. The work here comes to a similar conclusion with regards to the slope for the QCLAS, as von Bobrutzki et al. (2010) who also reported a slope less than 1 when compared to the ensemble median of the partaking instruments. The two studies however differ in that there it is not a clear split on the performance of the wet chemistry instruments. In von Bobrutzki et al. (2010) found all the wet chemistry instruments had a slope >1, whereas in this study at > 10 ppb the AiRRmonia#1 had a slope >1, whereas the reverse is observed for MARGA and AiRRmonia #2. This potential highlights how performance varies with set-up, but could also reflect further progress in the development of the spectroscopic methods since the 2010 study.

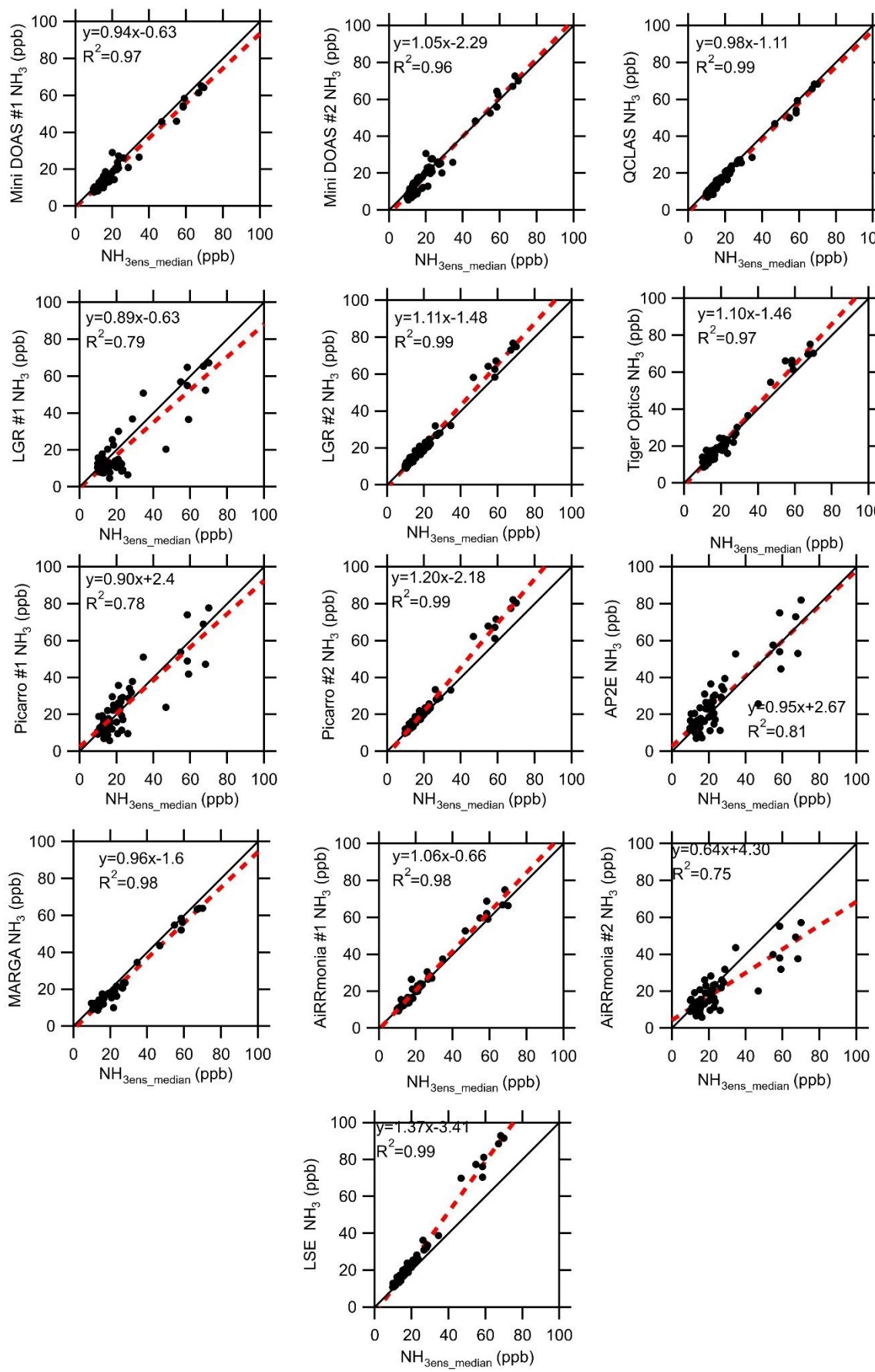

**Figure 9. Intercomparison of instruments (hourly) averages from 22/08/2016 to 29/08/2016 to the ensemble median (excluding LGR #1 and Tiger Optics) when the median is equal or greater to 10 ppb NH₃. Data were filtered for low wind speed and stable/unstable conditions that could have led to inhomogeneity at the site. The solid black line is the 1:1 line and red dashed line is the fit.**

### 3.7 Variability between individual instruments

To investigate the relationship between individual instruments least squares regressions were carried for i) the whole range and ii) when values were <10 ppb of the ensemble median (Tiger Optics was excluded for the <10 ppb comparison). The instruments were then clustered according to Euclidean distances based on their correlation coefficients. It is immediately clear (Figure 10) that using this approach all instruments compared well, when the whole period is studied. However, if the analysis is limited to below 10 ppb a different relationship emerges. The LGR #1 is the worst performing instrument with an average $R^2$= 0.44 when studying concentrations below 10 ppb, whereas the LGR #2, which is the same make and model compares well with other instruments. Even though remote troubleshooting from the manufacturer has been performed on LGR#1 (see section 3.2), this may be linked to a remaining misconfiguration of the instrument, preventing low $NH_3$ concentrations from being quantified with acceptable performance. The miniDOAS instruments compare well when studying the whole time series ($R^2$ = 0.99) and are even clustered together, however their relationship changes when examining concentrations below 10 ppb with an $R^2$= 0.88. At concentrations below 10 ppb, the instruments operating with their own inlets, with the exceptions of the QCLAS and AiRRmonia #1, correlated well with each other but not with the instruments on the manifold or the miniDOAS instrumentation. AiRRmonia #1 instead was grouped with the LSE and Picarro #2 on the manifold even though their locations were different. The QLCAS was grouped with the LGR #2 and the miniDOAS #1, even though its sampling point was the same as the instruments with their own inlets, suggesting the sampling point was not a factor.

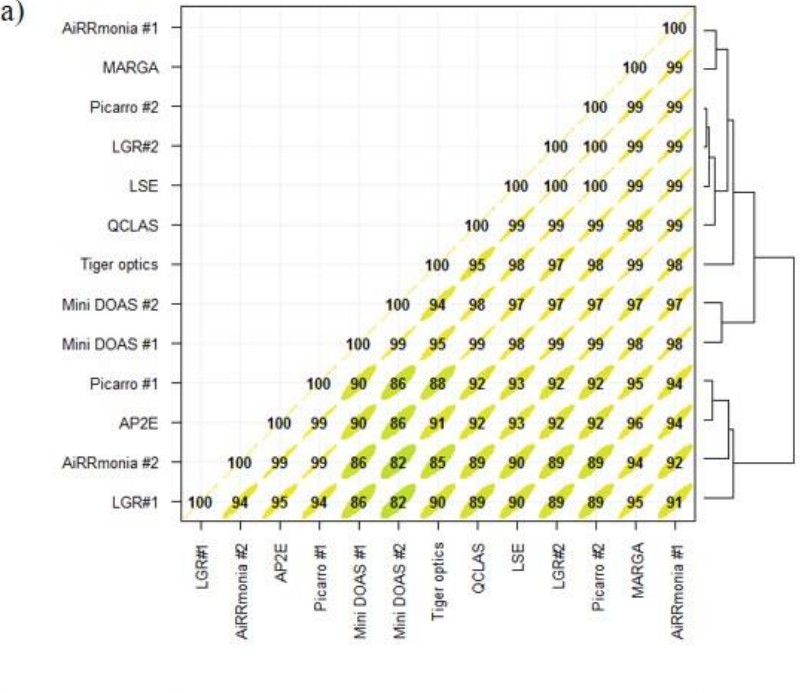

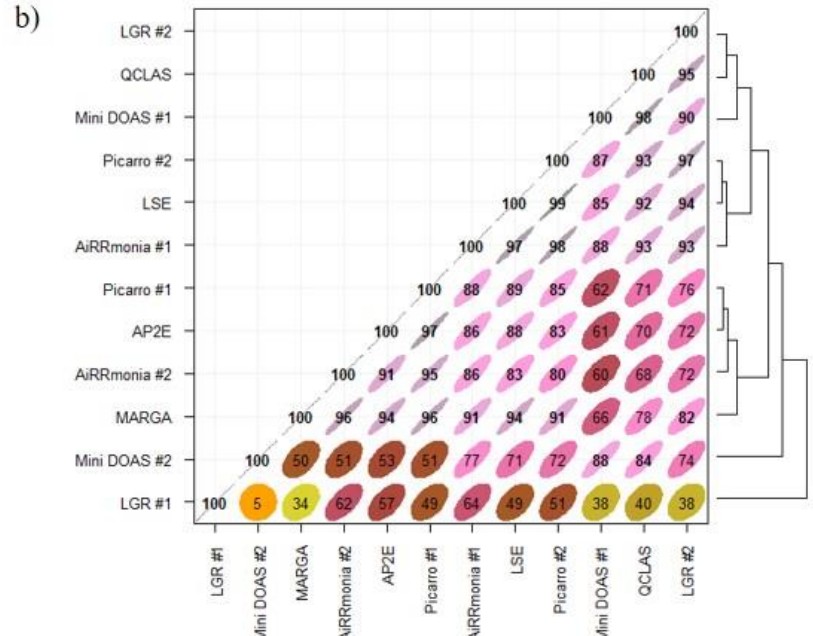

**Figure 10. Least squares regression correlation coefficients between instruments clustered into a matrix based on their Euclidean distances (black lines on RHS of the figure) for the a) whole range and b) when NH₃ <10 ppb of the ensemble median for the period of 23/08/16 00:00 and 29/08/16 01:00 based on their hourly averages. Graph generated using OpenAir package (Carslaw and Ropkins, 2012). Note the Tiger Optics is excluded from panel b). The colour scale relates to the magnitude of the correlation coefficient.**

The second approach used to assess the variability between instrumentation was to look at the normalised difference (ND) calculated between instrumentation using the equation (Pinto et al., 2014):

$$ND = \frac{X_i - X_j}{X_i + X_j}$$

where $X_i$ is the concentration of one instrument and the $X_j$ is the concentration measured by another instrument. The ND is then used to calculate coefficients of divergence (CD) to investigate the similarity between instruments as (Wongphatarakul et al., 1998):

$$CD_{ij} = \sqrt{\frac{1}{P}\sum_{i=1}^{p}\left(\frac{X_i - X_j}{X_i + X_j}\right)^2}$$

where $P$ is the number of points. For CD = 0 the two instruments are identical and a CD of 1 indicates the instruments are completely different. The reason this additional technique was chosen to compare instruments is that the statistical technique
provides greater weighting to low concentrations, where the main deviations occur between instruments, as observed when comparing the ensemble median to concentrations below 10 ppb (Sect. 3.5), and it also describes the systematic differences whilst even a correlation coefficient of 1 still allows for an offset and slopes other than unity. Table 3 summarises the CD values between instruments, with the comparison of the LGR #2 and the Picarro #2 having the smallest CD (0.04). It is clear that there is not much difference between the LGR #2 and Picarro #2 when looking at the ND (Figure 11a), though there may
be a positive bias of the LGR #2 to the Picarro #2 at lower concentrations. The two instruments which operated on the same manifold agreed well. There are a number of possible explanations for the positive bias at lower concentrations. It is known that both spectrometers have a potential for water interferences, as previously reported by Martin et al. (2016) for the Picarro. In this study the Martin et al. (2016) correction had not been applied to the Picarro #2. An alternative explanation is that the air sampled by Picarro #2 had a longer residence time between the manifold and the instrument (Table 2) resulting in greater
losses of NH$_3$ to the inlet, which is more evident at lower concentrations. Another hypothesis could be that the use of a heated inlet by the LGR #2 could have led to the potential of volatilisation of ammonium nitrate (NH$_4$NO$_3$ $\leftrightarrow$ NH$_3$ + HNO$_3$) generating an NH$_3$ interference. Compared to the miniDOAS #1, which does not have an inlet, there was no obvious difference in the ND for the LGR #2 and the Picarro #2 to provide a further explanation of the above hypothesises (Figure 11b and c).

The comparison of the miniDOAS #2 and the AP2E resulted in the largest reported CDs. When the ND is displayed (Figure 11d), it is apparent that the data is scattered, especially at the lower concentrations. It is especially noticeable that there was a divergence of the miniDOAS #2 when the instruments in this study are grouped based on their CD using a hierarchal clustering approach, where the Euclidean distance was calculated based on CD and presented in a dendrogram (Figure 12). Even though the miniDOAS #1 and miniDOAS #2 are the same analytical method they are separated into the two distinct groups. This is
hypothesized to be most likely the result of the different approaches in spectral algorithm and calibration procedures between the two miniDOAS instruments, see previous discussion in Sect. 3.5. Instead, the miniDOAS #1 clustered with the QCLAS. Even though the CD between the miniDOAS #1 and the QCLAS was low (Table 3), there appears to have been an obvious positive bias at lower concentrations in the QCLAS measurements when looking at the ND between the two instruments (Figure 11e). This positive bias was not observed for the Picarro #2 or LGR #2 in the ND when compared to the miniDOAS
#1, but was observed when both instruments were compared to the QCLAS, suggesting the bias lay with the QCLAS. The positive bias was investigated to see if it was related to drift in the instrument with time, background NH$_4^+$ aerosol or the influence of relative humidity; however, none of the parameters assessed could explain this bias at lower concentrations. One additional potential factor is the fit of the absorption spectrum at lower concentrations where the influence of optical fringes becomes greater. Even when the QCLAS is compared to the ensemble median either at <10 ppb or >10 ppb, it also had a slope
less than 1. This is not the first time the QCLAS is reported to underestimate compared to other instruments. Whitehead et al. (2008) reported in an earlier version of the instrument (using a pulsed rather than a continuous quantum cascade laser) that the

QCLAS reports lower concentrations but has a good $R^2$, compared to a wet chemistry method that sampled with WRD and analysed with selective ion membrane / conductivity analysis.

The large CDs of LGR #1 are likely due to drift of the instrument, which has been reported previously. In Misselbrook et al. (2016) data from two LGR instruments measuring $NH_3$ were rejected after there was significant drift in the reported values when doing periodic calibration checks. It cannot, however, be stated that this issue is only evident in the off-axis approach, as the AP2E and the Picarro #1 also performed poorly but instead highlights that all measurement techniques should be compared to either a calibration standard or another instrument at regular intervals. Overall, there is no clear message on the clustering of instrumentation based on their CD or using the correlation coefficient, as the LGR, Picarro and AiRRmonia instruments separate into the two distinct groups. von Bobrutzki et al. (2010) suggested that there should only be one sampling point for future intercomparisons, but it is clear that although most instruments that sampled from the manifold were clustered together, it is not the controlling factor of the CD clustering. The AiRRmonia #1, which was on the scaffolding at another location in the field, is also grouped with the manifold instruments. It is most likely that the clustering is also due to the time responses as a result of the instruments and inlet setup (refer to Table S3, Sect. 3.4). For example, the AiRRmonia#1, LGR#2 and Picarro#2 have similar time responses and are clustered together, whereas the AiRRmonia#2 and Picarro#1 have much slower responses and are clustered together.

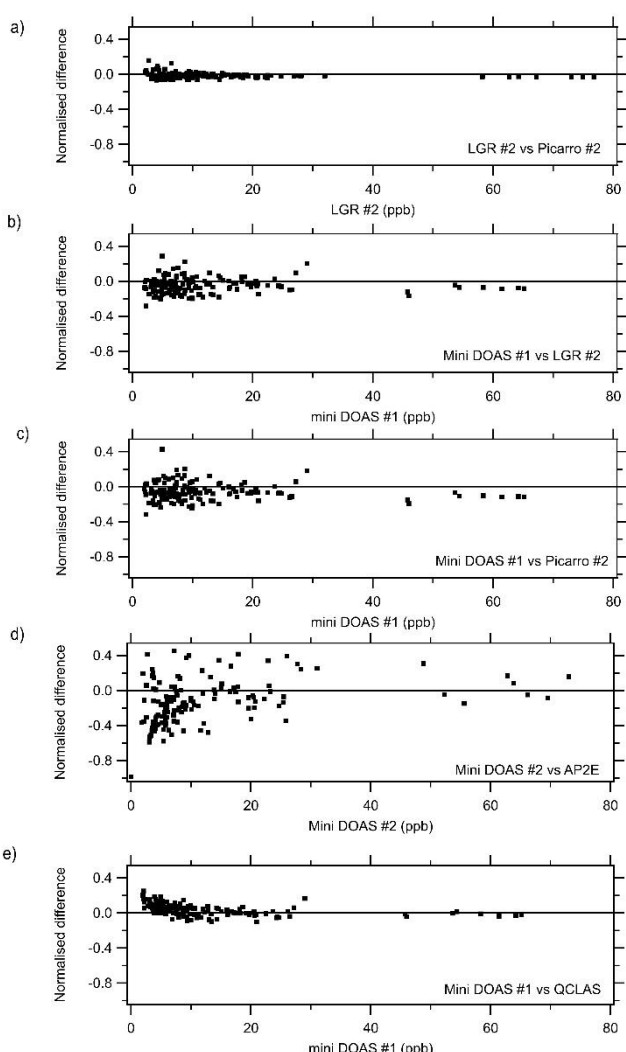

**Figure 11. Normalised difference (Equation ) between selected instruments for the period of the 23/08/16 00:00 and 29/08/16 01:00 based on their hourly averages.**

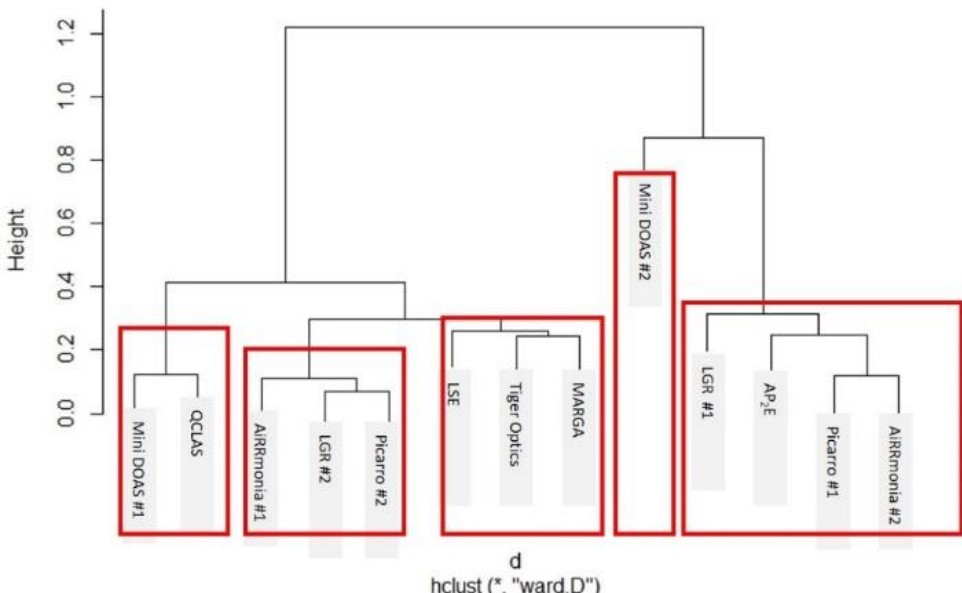

**Figure 12. Euclidean distances between instruments based on their coefficients of divergence for the period of the 23/08/16 00:00 to 29/08/16 01:00 based on their hourly averages. Height relates to the order at which the clusters occurred. The red boxes indicate the instruments that are clustered together.**

**Table 3. Summary of the coefficient of divergence (CD) of instruments based on their hourly averages for the period of 23/08/16 00:00 and 29/08/16 01:00. Comparisons with a CD ≤ 0.1 are highlighted in bold and italics.**

| | miniDOAS #1 | miniDOAS #2 | QCL | LGR #1 | LGR #2 | Tiger Optics | Picarro #1 | Picarro #2 | AP2E | MARGA | AiRRmonia #1 | AiRRmonia #2 | LSE |
|---|---|---|---|---|---|---|---|---|---|---|---|---|---|
| **miniDOAS #1** | | 0.25 | ***0.08*** | 0.23 | ***0.10*** | 0.14 | 0.21 | 0.12 | 0.24 | 0.16 | ***0.10*** | 0.20 | 0.16 |
| **miniDOAS #2** | | | 0.26 | 0.25 | 0.28 | 0.16 | 0.33 | 0.28 | 0.35 | 0.31 | 0.27 | 0.32 | 0.31 |
| **QCL** | | | | 0.22 | 0.11 | 0.13 | 0.23 | 0.13 | 0.24 | 0.14 | 0.11 | 0.21 | 0.15 |
| **LGR #1** | | | | | 0.21 | 0.18 | 0.15 | 0.21 | 0.19 | 0.16 | 0.19 | 0.12 | 0.23 |
| **LGR #2** | | | | | | ***0.09*** | 0.17 | ***0.04*** | 0.20 | 0.13 | ***0.06*** | 0.17 | 0.11 |
| **Tiger Optics** | | | | | | | 0.14 | ***0.08*** | 0.13 | ***0.10*** | ***0.07*** | 0.15 | 0.11 |
| **Picarro #1** | | | | | | | | 0.17 | 0.11 | 0.17 | 0.16 | ***0.07*** | 0.21 |
| **Picarro #2** | | | | | | | | | 0.18 | 0.13 | ***0.06*** | 0.17 | ***0.10*** |
| **AP2E** | | | | | | | | | | 0.18 | 0.18 | 0.14 | 0.18 |
| **MARGA** | | | | | | | | | | | 0.12 | 0.15 | 0.15 |
| **AiRRmonia #1** | | | | | | | | | | | | 0.16 | 0.11 |
| **AiRRmonia #2** | | | | | | | | | | | | | 0.21 |
| **LSE** | | | | | | | | | | | | | |

## 3.8 Bias compared to the Optical Gas Standard

An estimate of the bias of each instrument was calculated compared to the OGS (i.e. the alternative, first principles, offline evaluation of the Picarro#2 concentration using raw spectra) as the reference (refer to Figure 5 for hourly time series), where

*m* is the slope of the orthogonal regression when the intercept is forced through zero, as it is assumed that there is no artefact in the reference measurement (von Bobrutzki et al., 2010):

$$Bias = (m - 1) * 100.$$

Table 4 summaries the bias compared to the OGS, which ranged from -20 % to +23 % for the whole period (Figures can be found in Figs S2 and S3). The worst performing instruments, based on this metric, with a positive bias are the AP2E with +23 % and the Picarro #1 with +21 %, while those with a negative bias were the miniDOAS #2 and the QCLAS with -20 % and -15 % respectively. In contrast, unsurprisingly the manufacturer based evaluation of the Picarro #2 has a relatively small bias of 5 % since the OGS uses this instrument's spectra. The smallest reported biases, however, of ±1 % for the whole period are the LGR instruments, followed by the Tiger Optics and Airrmonia #1 with + 2 %. It is noted for the LGR #1 the correlation coefficient was weaker with an $R^2 = 0.79$ compared to the LGR #2 which had an $R^2 = 1.00$ (Figure S5).

The data was then filtered to only include periods where the ensemble median <10ppb. The bias previously reported for instruments compared to the OGS increased or remained the same with the exception of the miniDOAS instruments and the AP2E, where there was an apparent improvement in the bias (Table 3). It was apparent all the instruments sampling from the manifold have quite a low bias. LGR #2 and LSE, as well as miniDOAS #1 and AiRRmonia #1 had the lowest bias compared to the OGS. Though the Picarro #2 had a larger bias of 7 %, likely due to different spectral data and different data evaluation, it had a high correlation of $R^2 = 1.00$ (Figure S6), which again was to be expected as the same instrument was used to derive the OGS values. Below 10 ppb the largest positive biases are with the AP2E, Picarro #1 and AiRRmonia #2, where there are large negative biases for the miniDOAS #2, QCLAS and the MARGA. The bias of the miniDOAS and the QCLAS is most likely due to the OGS using spectral data from the Picarro #2, which has already been shown to be greatly influenced at below 10 ppb by the inlet set-up, resulting in a smoothed temporal pattern (refer to Sect. 3.4), whereas the miniDOAS and QCLAS retained the temporal features of $NH_3$, even at lower concentrations. To investigate the accuracy of the OGS in the field it was checked alongside the LSE and LGR #2 using standards produced by the ReGaS1 calibration system.

**Table 4. Bias calculated from orthogonal regressions (Figures S4 and S5) of hourly averages from instruments for the period of the 22/08/2016 to 29/08/2016 compared the OGS method. Data was filtered for wind speed and atmospheric stability. Bold data are biases which are >5 %.**

| | Full range of reported concentrations | <10 ppb NH₃ |
|---|---|---|
| Instrument | Bias (%) | Bias (%) |
| miniDOAS #1 | **-10** | -3 |
| miniDOAS #2 | **-20** | **-17** |
| QCLAS | **-15** | **-17** |
| LGR #1 (manifold) | 1 | **6** |
| LGR #2 (manifold) | -1 | 4 |
| AP2E | **23** | **20** |
| Picarro #1 | **21** | **22** |
| Picarro #2 (manifold)* | 5 | **7** |
| Tiger Optics (manifold) | 2 | N/A |
| MARGA | **-10** | **-12** |
| AiRRmonia #1 (minimal inlet) | 2 | 3 |
| AiRRmonia #2 (long inlet) | **9** | **14** |
| LSE (manifold) | 4 | -3 |

Note: The spectra from the Picarro #2 are used to produce the OGS

### 3.9 Ammonia calibration system

s previously stated the LGR #2, LSE and OGS were compared to the ReGaS1 calibration system. For 0 ppb it was found that the instruments reported the following average concentration during the of LSE: -0.77 ppb, LGR #2: 0.16 ppb and the OGS: 0.14 ppb (refer to Figure 13). The LGR #2 performed poorly compared to the other instruments. However, it is noted that the instrument was operated on a lower flowrate compared to that used during the field campaign (Table 1) resulting in a slower time response. It is evident in Figure 13 a that the LGR #2 was still stabilising and had not reached equilibrium. LGR #1 was part of this calibration; however, it developed a fault therefore no results are reported here.

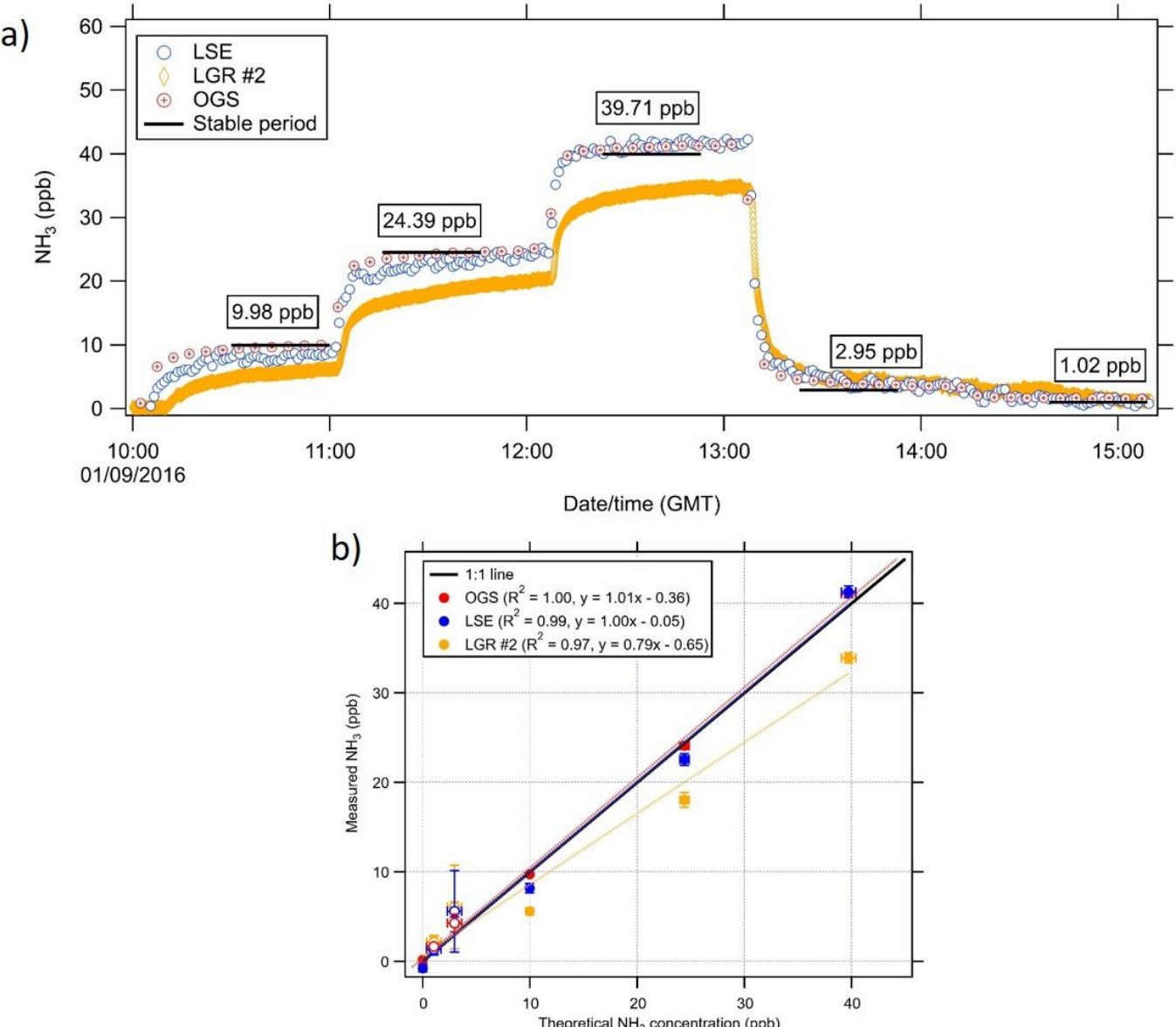

**Figure 13. Comparison of the LSE, LGR #2 and the OGS with the ReGaS1 calibration system. a) Time series of the comparison with the black horizontal bar is the period where the calibration gas is known to have stabilised in the ReGaS1 instrument. Boxes give the average reported ReGaS1 concentration during the stabilization period. b) Correlation plot of the comparison of the period. Open circles are concentrations following a reduction in concentration. The error bars for the theoretical concentration are the relative uncertainty and the error bars for the measured concentration is the $\sigma_A$ of the reported concentration. The calculated uncertainty for the OGS can be found in Table S4 and Figure S7.**

The OGS agrees closely to the expected concentrations, except at the two lowest concentrations, which were measured at the end of the experiment after a reduction from a higher concentration value and hence might be affected by longer response time (hysteresis) of the instruments (Figure 13b, Table S4). The OGS and ReGas1 values, however, are metrologically compatible (refer to Table S4 and Figure S7 for further details).

### 3.10 Calibration of miniDOAS with the Gas standard

On 22/08/2016, the miniDOAS #2 was compared to a PSM. A flow cell of 75 mm in length was installed in the optical path. From 13:30 to 14:52 UTC, $NH_3$ was flushed through this cell, from the PSM cylinder (#1825R2), which contained 99.78 ppm $NH_3$ in $N_2$. Taking into account the pressure and temperature in the flow cell, and the ratio between the open air path and the cell length, the extra concentration that the miniDOAS was expected to measure was 163 ppb. The results of the experiment are shown in Figure 14 below.

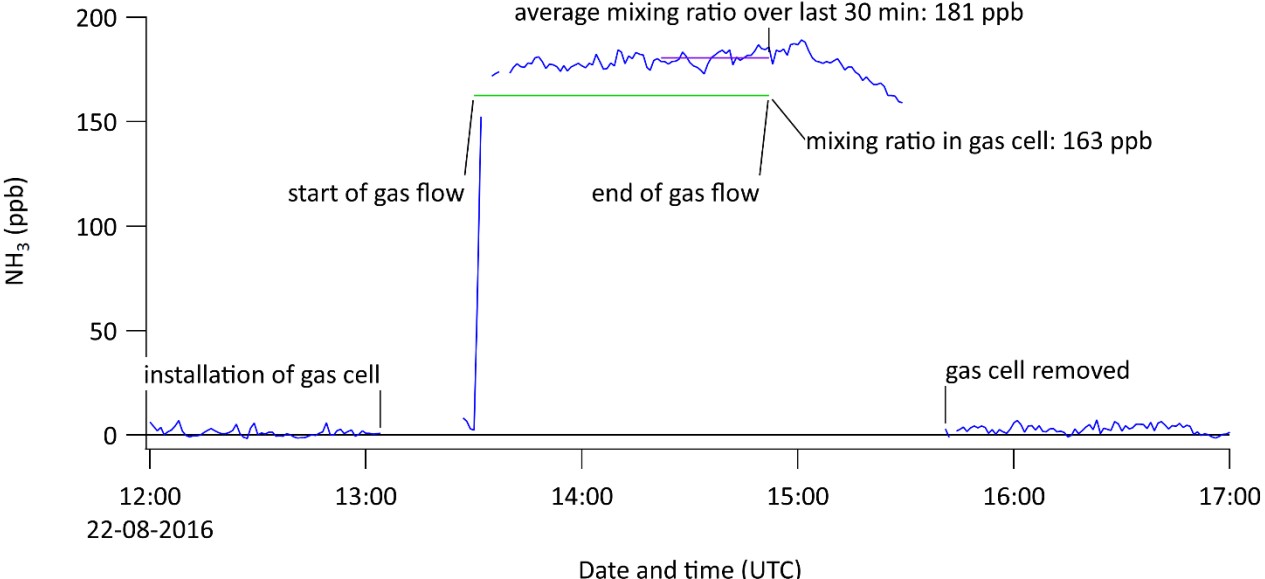

**Figure 14. Concentration measurements with the miniDOAS #2 on 22/08/2016, before, during and after the gas standard comparison experiment.**

The $NH_3$ concentrations measured in the open air on this day were low, 1.37 ppb averaged over the hour before the experiment, 3.31 ppb averaged over the hour after the experiment. After the start of the gas flow from the PSM cylinder, which had an expanded uncertainty of <2 %, the concentration as measured by the miniDOAS rose sharply at first, and then rose much more slowly as an equilibrium was established. Even after the gas flow was stopped, the measured concentration still rose somewhat, indicating that a steady-state had not yet been reached. After 15:00, the $NH_3$ diffused out of the cell and the open tubing, and the measured concentration decreased. For the comparison, we take the average of the concentrations measured by the miniDOAS over the last 30 min the gas flowed. This was 181 ppb, 11.1 % more than the nominal concentration in the flow cell. The experiment showed in principle the gas cell approach can be used for span checking the miniDOAS, however further research into making this type of span checking affordable, routine and at concentrations relevant to ambient concentrations is needed before this approach can be routinely applied in the field.

## 4 Discussion

### 4.1 Accuracy and precision of the measurements

In this study we assessed the precision by comparing the inter-variability between instruments and the variability against the ensemble median ($R^2$). In a previous study by von Bobrutzki et al. (2010), the main factors identified for affecting the precision of the measurements were a) inlet design, b) the condition of inline filters (where applicable) and c) the quality of gas phase calibration standards. In this study, it has further been shown that the precision across the suite of instruments is also dependent on the ambient concentration measured and instrument response time to rapidly changing concentrations. The majority of the

instruments, with the exception of the QCLAS (specifically designed for fast response) and miniDOAS (open-path), have a fairly slow response to variations in ambient concentration (Table S3), some because of their internal measurement principle, others because their inlet and filter systems dampen concentration peaks (von Bobrutzki et al., 2010). The fast response instruments (QCLAS and miniDOAS) therefore had more structure in their temporal patterns compared to the ensemble

median (Figure 3). As a result, more scatter is observed in the correlation plots for these instruments (Figure 8 and Figure 9), resulting in a misconception that these instruments had a poor precision, however when these fast response instruments were compared to each other (QCLAS vs miniDOAS #1, Figure 15a) at differing averaging times of 1 min (Figure 15b), 10 min (Figure 15c) and 720 min (Figure 15d), the precision improved correspondingly: (a) $r^2 = 0.85$, slope:0.87, intercept: 2.69 ppb; (b) $r^2 = 0.94$, slope:0.97, intercept: 0.72 ppb and (c) $r^2 = 1.0$, slope: 1.09, intercept: 1.17 ppb. The larger scatter therefore in

comparing the QCLAS and miniDOAS instruments to the ensemble median (Figure 8 and Figure 9) should not be taken as a sign that the faster response instruments have reduced precision but as evidence of the difference of precision due to differences in instrument time response (refer to Sect. 3.4). It is likely the observed finer scale structure reflects the heterogeneity of the air concentrations of $NH_3$ across the field that both instruments would detect. Although it is, beyond the scope of this study to carry out a full site emissions modelling exercise (e.g. with Lagrangian modelling) the data from this study could in future be

used to explore concentration heterogeneity at these fine scales. Also, evidence of the precision of the MARGA has to be treated with caution as the inlet set-up in the study was atypical, with a long-length (8.46 m) *c.f.* more typical 1.29 to 4 m inlet setups (Makkonen et al., 2012; Rumsey et al., 2014; Twigg et al., 2015). It is therefore likely that the time response reported here (Table S3) is not a true reflection of the time response of the MARGA instrument, instead a set-up without an inlet would have to be undertaken to quantify the time response. More generally, differences between the performances of near identical

instruments and spectroscopic methods (e.g. Picarro#1 and #2; LGR#1 and #2, miniDOAS #1 and #2) shows that performance is not purely linked to the measurement approach or instrument but greatly influenced e.g. by inlet set-up, operation (e.g. flowrate), and the status of the instruments, which likely includes the status of the filters where applicable.

An assessment of the accuracy in this study was determined by the comparison to a CRDS based OGS (Picarro#2 with modified

algorithm) and for some checks through the in-field gas calibration standards. For the comparison to the OGS (Table 4, Figures S4 and S5) some instruments show very little bias (LGR#1, LGR#2, Picarro#2, Tiger Optics, AiRRmonia#1 and LSE), all of which, with the exception of AiRRmonia#1 were attached to the manifold also used by the OGS. The remaining instruments were not attached to the same manifold and had either the slowest (Picarro#1, AiRRmonia#2, AP2E and MARGA) or fastest time responses (QCLAS, miniDOAS#1, miniDOAS#) as set out in Table S3. Therefore, no conclusion can be made on the

accuracy of the reported concentrations of these instruments. The OGS comparison is likely to be limited by i) instruments not sampling at the same point, with miniDOAS measuring a line average and ii) the OGS concentrations being limited by the set-up of the Picarro #2 instrument. The OGS, however, is a promising methodology, as the OGS and the ReGaS1 values were comparable but further research is required, especially regarding gas sampling issues, prior to the system being used as a reference methodology for routine monitoring.

Both the spectroscopic methods and wet chemistry methods have some cross sensitivities that would affect the accuracy of the reported concentrations. The reported concentrations are likely to have been impacted by ammonium aerosol deposition to surfaces (inlets or filters), which have the potential to generate an artefact through the volatilisation into $NH_3$ gas. In the von Bobrutzki et al. (2010) study, it was found that one (photoacoustic) instrument overestimated $NH_3$ concentrations compared

to other instrumentation prior to the filter being replaced. Stieger et al. (2018) also observed when comparing the MARGA to a Picarro instrument, that the Picarro reported up to 3 µg m$^{-3}$ more $NH_3$ compared to the MARGA when it was reporting low concentrations ($< 5$ µg m$^{-3}$). This was attributed by Stieger et al. (2018) to be artefacts of the volatilisation of ammonium nitrate from the filter, whereas the reverse was observed under higher $NH_3$ concentrations. It was hypothesised, by the authors,

potentially negative artefacts could occur at higher concentrations due to the formation of aerosol on the filter. Unfortunately, during this study, filters used by the instruments were not replaced. Therefore, the reported positive intercepts discussed above (Figure 8 and Figure 9) cannot be conclusively attributed to contaminated filters. It is however noted that the average concentration reported when compared to the ALPHAs (Table S1) that the instruments with filters tended to report higher concentrations compared to filter free methods (Table S1), supporting the suggestion that filters introduces an artefact. There is recent evidence that frequent filter changes are starting to be considered by network operators to limit artefacts in measurements. For example, He et al. (2020) reported changing filters a frequency of between 2 weeks to monthly, dependent on atmospheric conditions for a CRD instrument.

The Picarro instruments operated during the campaign are known to have suffered a spectral interference by $H_2O$. As Martin et al. (2016) found this could be corrected for by a water correction algorithm for the Picarro. However, this interference is known to be rather minor (<4 %). In case of the OGS the data evaluation algorithm has included spectral water suppression approach and thus has no need for additional empirical water corrections. The Picarro instruments in this study did not have the water correction applied and therefore the results are likely to change with humidity, as a result it is likely to have affected the accuracy of the reported concentrations. During the campaign, also the LGR#1 displayed issues in precision and accuracy (refer to Sect. 3.2). Misselbrook et al. (2016) has previously reported issues in accuracy of an LGR instrument. Misselbrook et al. (2016) found that there was significant drift in the recorded values during calibration checks. This issue is not only limited to the LGR but has also previously been observed in the laboratory too on the Picarro (Twigg, 2022). Unfortunately, an assessment of the drift of instrumentation studied using the ReGaS was not possible during this study. It is recommended that such assessment is undertaken in future studies. However, it provides evidence that regular calibration span checks are required to be carried out to determine the accuracy and precision of instrumentation, especially instrumentation considered to be plug and play instruments which are thought to be stable in time. If it had not been for the comparison with other instruments, the poor performance of LGR#1 may have taken longer to identify if operating in isolation.

Manufacturers of some instruments used state the instruments are stable and do not require recalibration, however they do recommend routine span checks. But no frequency is provided by the manufacturer (LGR, Picarro, AP2E and Tiger Optics). The exception is the LSE instrument where a calibration is recommended to be undertaken twice a year and at the same time the filter replaced. At the time of this study there was no routine maintenance protocol from the other manufacturers on the frequency of filter changes. Tiger Optics recommend that their inline filter is replaced when it begins to show discolouration, and Picarro only when the filter becomes blocked. As filters are a known source of uncertainty of the absolute $NH_3$ concentration, it is of concern that manufactures do not provide a recommended maintenance schedule for both filters and span checks.

## 4.2    Inlet design

Consistent with previous $NH_3$ measurement studies, our results have demonstrated that inlet design is important. Whitehead et al. (2008) demonstrated that polyethylene (PE) or Teflon (PTFE) had the best response time compared to stainless steel or silcosteel, whereas PE was found to be best by Dias (1988). Vaittinen et al. (2014) studied the absorption of $NH_3$ under a range of humidities for stainless steel, stainless steel with Dursan, SilcoNert 2000 and halocarbon wax coatings, as well as Teflon (PTFE) and polyvinylidene difluoride (PVDF). It was found that PVDF and PTFE were the least absorbing materials. In this study, all operators used either PFA/PTFE or PE for their inlet. It has become evident that though inlet material is important (where applicable), consideration of the surface to air volume ratio and residence time are also important controlling factors. For example, the air sampled by Picarro #1 had a shorter residence time in the inlet compared to the Picarro #2 but had a larger ratio of surface to volume (Table 2), which is likely to have led to greater interaction of $NH_3$ molecules with the surface wall. It therefore would be the recommendation to operate, where an inlet is required, to minimise the wall interactions by

minimising the length of inlet used, residence time and surface to volume ratio of the inlet. A previous study by Norman et al. (2009) demonstrated the importance of condensation on inlet lines and that care needed to be taken toensure that condensation did not occur in the inlet. They recommended an optimal design might therefore include thermal insulation and if possible, keeping inlets heated a few degrees above the ambient temperature, particularly also any sections that run within air conditioned measurement cabins. Ellis et al. (2010) also evaluated the use of a heated inlet and found that heating the inlet line led to an improvement in the time response of a QCLAS. During this study, only the LGR#2 and the QCLAS used heated inlets. Caution however is required when heating an inlet, as if the temperature is too high, this will lead to the dissociation of $NH_4NO_3$ leading to an artefact. There is no evidence to suggest for the QCLAS that this led to $NH_3$ artefacts from ammonium nitrate, as the QCLAS had a very small positive intercept (0.05 ppb) for concentrations <10 ppb when compared to the ensemble median (Figure 8). However, the inertial inlet of the QCLAS is designed to remove much of the ammonium nitrate from the air stream. The LGR#2, however, had a positive intercept of 0.65 ppb (Figure 8), though this cannot be concluded to be the result of heating as the instrument used filters too. Not considered in this study is the contamination of the inlet, which is likely to occur over time and has been discussed previously in the literature, though there still no recommendations for frequency of either cleaning or replacing inlets. Moravek et al. (2019) for example, demonstrated that for the QCLAS time response degrades with age (based over a 5 month study) due to contamination of the inlet, but even after cleaning the response time did not always return. As a result some network instruments have already started to frequently replace their inlets. Twigg et al. (2015) replace their inlet at quarterly intervals for the MARGA currently operated in the UK to try to minimise contamination. Though not studied here, it would therefore also be recommended to consider the frequency that an inlet is either cleaned or replaced to account for potential loss of precision.

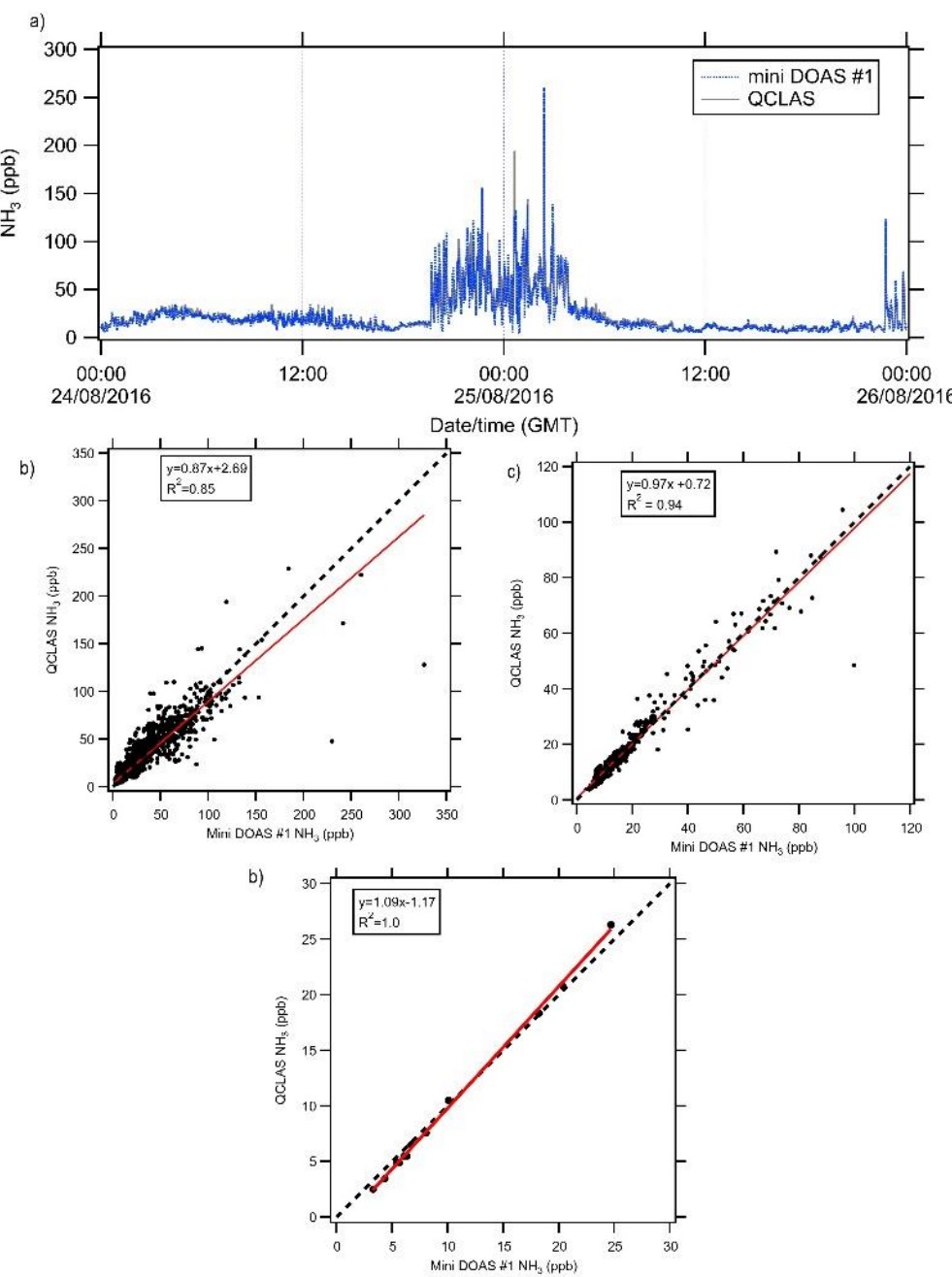

**Figure 15. a) Time series of mini-DOAS #1 vs QCLAS (1 min averages) for the period 24/08/2016 to 26/08/2016 b) correlation of 1 minute averages c) regression plot for 10 min averages d) 12 hour averages for the same period.**

### 4.3    Progress towards standard operating procedure for routine NH₃ monitoring

This study highlights that currently there is no standard operating procedure for NH₃ instrumentation in monitoring networks and it is at the discretion of the user to determine the monitoring network design. There is evidence to suggest that this approach will lead to variations in reported concentrations, as seen in both the comparison to the ensemble mean (Figure 8) and between instrument variability (Figure 10 and Table 3). The inferences from artefacts and alterations in instrument performance is an ongoing area of concern for NH₃ instruments used in long term monitoring. It highlights the need for further development of protocols to ensure the precision and accuracy of instrumentation. This is likely to be achieved through regular zero and span checks, as well as a regular servicing program, which is not yet available for any of the instrumentation presented to the authors knowledge. Work is required to determine if span checks and calibrations should be undertaken using humidified air, as the evidence from Martin et al. (2016) would suggest that reported NH₃ concentrations from spectrometry methods are likely to suffer interferences from water. However, preparing humid gas samples with accurately characterised NH₃ concentrations in

the ambient concentration range is challenging and work is required to develop standard methodologies to produce a humidified gas standard such as using a scrubber or heated catalysts. Pollack et al. (2019) provides a valuable study in evaluating these approaches. It would be advisable that a standard is also used on a frequent basis to determine the contamination of the set-up, as previously demonstrated by Ellis et al. (2010) and Pollack et al. (2019), who observed that inlet

contamination can be identified via an increase in the calculated time response. In addition, a standard inlet design needs to be agreed (where applicable).Evidence from the Picarro and AiRRmonia set-ups in this study (Figure 7), would suggest that inlet design can lead to losses of information of the temporal pattern of $NH_3$. Consideration is also required to determine if passivation of the inlet is valuable to routine air quality monitoring, as there is evidence that it can effectively reduce the interactions of $NH_3$ with the inlet walls (Roscioli et al., 2016). Open path techniques, such as DOAS, will benefit from the

availability of zero-air facilities, where instruments can check their zero level on ammonia free air. Work on such a facility is ongoing.

This study did not include the methodologies that are the current $NH_3$ reference methods used by the US EPA and EMEP. There is literature evidence, however of the MARGA being compared to reference methodologies. Makkonen et al. (2014) compared the MARGA to the EMEP filter pack method at a background station, Hyytiälä in Finland. It was found that

MARGA compared well to the filter pack method at low concentrations ($< 0.8$ µg m$^{-3}$). Stieger et al. (2018) also, found that the MARGA compared well to acid coated denuders ($NH_3$ mini-denuder, Midefex and Radiello®) with $r^2$ from 0.82 to 0.98. However the MARGA reported higher concentrations compared to these denuder methodologies with slopes ranging from 1.30 to 1.53. This is in contradiction to Rumsey et al. (2014), who found that though precision was within acceptable limits, the accuracy of the MARGA was variable with concentrations being consistently underestimated compared to the US EPA

reference denuder methodology, which has a sampling frequency of 12 hours. This loss was attributed by Rumsey et al. (2014) to be due to the consumption of $NH_3$ by bacteria. The studies for the MARGA give mixed conclusions that is likely to be due to variations in set-ups between studies and the reference methodology used. To the authors knowledge there are no further comparisons of the US EPA and EMEP reference methodologies for the other instrumentation presented in this study. It would be therefore advisable for any future study to compare the instrumentation presented here to the US EPA and EMEP reference

methodologies, using a similar approach outlined in the European guidance to demonstrate equivalence for ambient air monitoring (GDE, 2010) in order to quantify the uncertainty in the different measurement techniques. This study did not include all instruments currently used in routine monitoring of $NH_3$ across the globe. In India, for example, the Central Pollution Control Board (CPCB) monitors $NH_3$ concentrations by the indirect measurement of $NH_3$ through conversion by a molybdenum convertor, coupled to an NO chemiluminescence analyser (Pawar et al., 2021). In future, any other instruments

identified to be used in routine monitoring of $NH_3$ should be added to the suite of instruments to take part in any follow up study looking at the uncertainties compared to reference methodologies.

With the available instruments showing significant variability within the $< 10$ ppb range, it is clear that the accurate assessment of the exceedance of Critical Levels (CL) of $NH_3$ concentrations for sensitive ecosystems with these automated measurement methods remains a challenge. Critical levels are there for the protection of vegetation from damage from $NH_3$, currently these

set at annual averages of 1 µg m$^{-3}$ for lichens and bryophytes, and 3 µg m$^{-3}$ for higher plants by the International Cooperative Programme (ICP) Vegetation of the United Nations Economic Commissions for Europe (UNECE, 2007). Therefore to achieve a quantitative annual measurements with high temporal resolution instruments, great care in set-up and operational quality assurance, and data quality control would be required to achieve the CV $< 20$ %, set in this study. The UNECE (2007) retained a monthly critical level of 23 µg m$^{-3}$, as a provisional value for the prevention of ecological damage during intermittent periods

such as fertiliser/manure spreading seasons. The instrumentation in the study have been shown to be cable to achieve an acceptable CV at these more elevated concentrations, however care would be need to be taken to minimise base-line drift and instrument contamination. As with the annual averages, a similar care in set-up, operational quality assurance and data quality control would be required to ensure a traceable and acceptable level of data quality for policy evidence purposes.

## 5    Conclusions

To date this study is the most comprehensive comparison of $NH_3$ instruments which are, or have the potential to be, used in routine monitoring of $NH_3$ from background concentrations (<1 ppb) to agricultural emission sources (>100 ppb). Due to the interaction of $NH_3$ with inlets and other surfaces, comparison of instruments is complicated due to some instrument response times reaching or exceeding the 1 hour averaging time and the difficulty of sampling at the same location (due to size of instruments and the need for longer sampling lines). Overall, the instruments studied performed well at elevated $NH_3$ concentrations, though there is evidence that MARGA has a limited range in the configuration presented. At concentrations below 10 ppb, performance in precision, however differed, with instrumentation splitting into two distinct groups based on instrument set-up. At low concentrations, even seemingly identical instruments performed differently, highlighting the impact of the setup, inlet design and operation (external pump, inlet length, maintenance, filter aging); here inlet and filter-less instruments have an intrinsic advantage. It should be noted that real-time instruments are currently evolving and some of the instruments included in this study have been further developed since and partly in response to the study.

In general, the level of agreement between instruments participating in this intercomparison was encouraging. However, given the variation in performances at low and high concentrations, there is still a lot of work to do to achieve equivalence of measurements. Ideally we should be aiming for a coefficient of variance of 10%, if there is to be confidence in measurements from different places being compared – for example measurements being used to evaluate critical levels, long term trends or for integrated concentrations for exposure. Therefore if different instruments are to operate in an air quality network, equivalence work is required to determine the uncertainties across the techniques to ensure comparability against a reference method.

Overall, the simple requirement for both science and policy is that ambient $NH_3$ concentrations are measured to a known accuracy and precision, particularly for long-term measurements (weeks, months, years). Therefore, long-term $NH_3$ measurements need to be fit-for-purpose taking into account the time response required and the range of concentrations to be observed. Networks with multiple measurement sites need to be comparable and this will be only achieved by prescribed set-ups and traceable quality assurance and quality control protocols which are developed to achieve data quality with operational economy. Without further support of the other instrumentation present in this study it would have been incredibly difficult for a data user to verify the accuracy and precision of the reported $NH_3$ concentrations. Therefore, to understand instrument performance it is strongly recommended for any short- or long-term deployment as part of routine QA/QC, regular calibrations as well as zero and span checks should be undertaken at a frequency determined by the operational need in the location of the measurements (i.e. high concentration and high PM concentrations will likely necessitate more frequent maintenance and checks). This will enable routine reporting of monitoring station and inlet/instrument system specific precision and accuracy. Further long-term monitoring research is required to develop and test standard operating protocols for instrument set-up, in-situ calibrations and maintenance routines such that an international set of standards can be agreed.

*Data Availability:*
The one hour averaged data will be available all instruments at the UKCEH Environmental Information Data Centre and raw data files are available data from authors on request.

*Supplement Link:*

*Author contributions:*

MT, CFB and EN designed the experiment. All co-authors were involved in various aspects of data collection. MT conducted the data analysis and interpretation of data, with input from all co-authors. MT prepared the manuscript with contributions from all co-authors.

*Competing interests*

Authors LJ and DM are employed by Enviro Technology Services who are the distributors of the LGR and LSE in the UK.

Authors AN, JS and CH have contributed to the development of the miniDOAS #1.

Authors AB and MH are employed by RIVM who are the manufacturers of miniDOAS #2.

Author BK is employed by LSE monitors who are the manufacturers of LSE.

Authors RO and CN are employed by TigerOptics who are the manufacturers of the Tiger Optics instruments.

Authors DL, CP and BN are employed by METAS who developed and constructed ReGaS1.

Authors AP and NL are employed by PTB who developed the OGS.

**Acknowledgements**

The measurements were funded through the following projects which supported the measurements during the intercomparison study:

- Metrology for NH3 in ambient air (MetNH3) project as part of the European Metrology Research Programme (EMRP) of the European Union.
- The Academy of Finland as part of the Centre of Excellence program (project no 1118615)
- The Swiss Federal Office for the Environment (Contract 06.9115.P2I P264-1000)
- For IMT Nord Europe, the participation to this work has been supported by the French Ministry of Environment (Grant number: : 2200995403),  as a part of the activities of National Reference Laboratory for air quality monitoring (LCSQA)

The field site infrastructure at Easter Bush is supported by the NERC UK Status, Change and Projections of the Environment (UKSCAPE) National Capability Programme (NE/R016429/1). The data analysis and preparation of this manuscript was supported by the NERC Integrated Research Observations System for clean air (OSCA, NE/T001798/1). We dedicate this paper to Dr Rens Zijlmans, who sadly passed away before the publication of this paper.

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
