# Peer review of "In-situ measurements of NH3: instrument performance and applicability"

_Atmospheric Measurement Techniques, 2022_

## Author Comment (AC1)

Comment on amt-2022-107 Anonymous Referee #1 comment on "In-situ measurements of NH3: instrument performance and applicability" by Marsailidh M. Twigg et al., Atmos. Meas. Tech. Discuss., https://doi.org/10.5194/amt-2022-107-RC1, 2022

The authors thank the referee for the careful and constructive consideration of this manuscript. The answer is structured as follows: the comments from referee #1 are marked in black and the authors' response and changes to the manuscript are written in blue.

**Summary:** Ammonia (NH3) is not regularly monitored in networks. It is also an unregulated pollutant in many countries. Identifying a standard among the available technology for routine gas-phase ammonia monitoring is vital to our understanding of this pollutant and how future regulation policy can be shaped. This paper describes a comprehensive intercomparison of 13 instruments for sampling gas-phase NH3. The study is done in-situ at an agricultural field site in Scotland. The instruments represent a variety of currently available technologies for measuring gas-phase NH3. The authors perform a comprehensive analysis of the available data and show that variability in the ensemble is within 20%. This alone is an interesting finding from an intercomparison of 13 independent NH3 instruments of varying techniques and time responses. While there are still nuances of instrument setup, maintenance, and operations to be determined, the observations reported in this work are a step forward towards developing standardized practices for NH3 monitoring. The paper presents findings from a new field study and the topic is highly relevant towards addressing current air quality and climate concerns. Thus, it fits the scope of AMT. I recommend this paper for publication in AMT following minor revisions.

**General Comments**

In general, the flow of the paper could be improved by moving around some of the sections and refocusing the key points in the introduction to match the outcomes.

We already changed the structure a couple of times when compiling the original manuscript and, because different sections rely on each other, do not believe that re-ordering will help. Instead, we have now outlined in the introduction the flow of the paper and made clearer that the metrological methods were not used to correct any of the data used in the study but evaluate the applicability of the metrological methods in the field, as this is where the confusion may have arisen due to the mentioning these later in the paper.

For example, the introduction of this paper implies that the authors will provide recommendations on how to achieve high quality future routine monitoring of NH3. However, the conclusions do not provide specific recommendations for what the optimal inlet setup and operating/maintenance procedures could be for a monitoring site.

We have now gone through the reviewer comments and have expanded the text with regards to inlet design. We do not however feel it is appropriate with the evidence from this specific study to provide firm recommendations for frequency of span and zero checks. Please refer to Sections 4.1 to 4.3 for the revised text. We have adjusted the text in the conclusions as follows (Conclusions paragraph 2:

*"Networks with multiple measurement sites need to be comparable and this will be only achieved by prescribed set-ups and traceable quality assurance and quality control protocols which are developed to achieve data quality*

*with operational economy.,—Without further support of the other instrumentation present in this study it would have been incredibly difficult for a data user to verify the accuracy and precision of the reported NH₃ concentrations. Therefore, to understand instrument performance it is strongly recommended for any short- or long-term deployment as part of routine QAQC, ,—regular calibrations as well as zero and span checks should be undertaken at a frequency determined by the operational need in the location of the measurements (i.e. high concentration and high PM concentrations will likely necessitate more frequent maintenance and checks). This will enable routine reporting of monitoring station and inlet/instrument system specific,—precision and accuracy. Further long-term monitoring research is required to develop and test standard operating protocols for instrument set-up, in-situ calibrations and maintenance routines such that an international set of standards can be agreed."*

In contrast, a key conclusion of this work that is not identified earlier in the paper is that the variability in the ensemble of 13 instruments is within 20%. Such a tight cluster of NH3measurements from 13 independent instruments actually seems pretty good, especially considering that gas-phase NH3 can be challenging to measure. It would help to contextualized this finding better in the text in terms of what this could mean for monitoring networks comprised of a few types of NH3 instrument techniques.

The average variability being within 20% is encouraging. However, given the variation in performances at low and high concentrations, and the variation in peak concentrations, it implies there is still a lot of work to do. The performance which should be achieved is better than 10% and higher if there is to be confidence in measurements from different places being compared – for example for critical levels or for integrated concentrations for exposure.

If different instruments were operated in an air quality network, equivalence work would be required to determine the uncertainties across the techniques to ensure comparability. We have added the following text to the conclusions to reflect this discussion.

*"In general, the level of agreement between instruments participating in this intercomparison was encouraging. However given the variation in performances at low and high concentrations, there is still a lot of work to do to achieve equivalence of measurements. Ideally we should be aiming for a coefficient of variance of 10%, if there is to be confidence in measurements from different places being compared – for example measurements being used to evaluate critical levels, long term trends or for integrated concentrations for exposure. Therefore if different instruments are to operate in an air quality network, equivalence work is required to determine the uncertainties across the techniques to ensure comparability against a reference method."*

There are some very detailed instrument descriptions in the methods section and some that are rather vague. It would be helpful to include a similar level of detail for all instruments, especially in cases where references are not available.

Where possible we have revised the text for the instrument descriptions. Some instruments used during this study are considered 'plug and play' or 'proprietary', with limited information available from the manufacturer on their principles of operation. For others, where we have detailed

information, some have already been published in the literature (to which we can defer), others less so (so more detail needed here). Please refer to Section 2.2 for further details.

Please also clarify the purpose and design of the experiments using identical instruments with different inlets earlier in the paper. This is an important factor in the final recommendation, but there is little information leading up to these results to prepare the reader to understand these findings.

In the introduction we have now added a line to highlight the presence of duplicate instrument with the following text:

*"The importance of set-up and time response is also considered through the use of duplicate instruments with different inlet designs."*

Please include more context in the methods section about how the calibration sources were used in these experiments and what instruments they were used with. Were these calibrations only applied in the field, or were instruments calibrated individually in a laboratory before/after the field experiment? Please clarify in the text. It could also help the flow of the paper to move up "Section 3.9 Ammonia calibration system" to between Sections 3.2 and 3.3.

Thank you for the suggestion. Upon review, we feel it would be inappropriate to move the presentation of the results of the performance of calibration systems. The purpose of this section was to identify if any of the calibration systems available from the project were ready for use in the field, as well as using them to evaluate measurement system performance. The results of ReGaS calibration was not applied to the LSE or Picarro data in any of the analysis, as at the time of the study the Picarro manufacturer claims that the method is a 'calibration free' method. We have reviewed the text and updated the text in the data analysis section 2.4 to make clear which instruments had calibrations and zeros applied:

*"For the AiRRmonia, MARGA, QCLAS and the miniDOAS instruments, the zeros and/or calibration standards used are described in Sections 2.2.1, 2.2.3 and 2.2.7 respectively. They were applied to the data set prior to undertaking the analysis undertaken in Section 3. No other LTMHTR instrument had any zero or calibration applied, as the instrument manufacturers described the methods as 'calibration free' at the time of the study, so were only operated with the manufacture factory calibrations."*

In addition, we have highlighted the purpose of the standards in this study with a revision of the final paragraph of the introduction:

*"Metrological components developed under the MetNH3 project, are also evaluated under field conditions as standards for determining the accuracy of the instrumentation deployed, as previous studies of the metrological applications have focused on laboratory studies (Pogány et al., 2016, 2021)."*

Zeros were routinely performed for the QCLAS, but what about the other instruments? Were span calibrations also performed? Are zeros and span calibrations stable over time?

Please refer to previous comments with regards to the zeros and span calibrations of instruments. In order for the results to be representative of real-world deployment, instrument providers deployed as per their normal practice, as summarised in methods section. In addition, the comparison to the ReGaS system was only carried out for some instruments and then not using the same set-up as

operated during the intercomparison itself, reflecting limitations in the flowrate the ReGAS can provide. An attempt was made at the start and at the end of the campaign to determine the drift of the instruments evaluated, however due to issues in maintaining the stability of the ReGAS system at the start of the campaign, this comparison was considered invalid and not presented. The ReGAS was further developed during the campaign and there was confidence that study presented from the end of the campaign was valid. To highlight that this key assessment is missing from the study the following has been added to the text (Section 4.1).

*"Unfortunately, an assessment of the drift of instrumentation studied using the ReGaS was not possible during this study. It is recommended that such assessment is undertaken in future studies"*

Were corrections associated with these zeros and spans applied to the data prior to analysis in Section 3? Please clarify in the text.

Under section 2.4 Data Analysis. The following text has been added.

*"For the AiRRmonia, MARGA, QCLAS and the miniDOAS instruments, the zeros and/or calibration standards used are described in Sections 2.2.1, 2.2.3 and 2.2.7 respectively. They were applied to the data sets prior to undertaking the data analysis undertaken in presented in Section 3. No other LTMHTR instrument had any zero or calibration applied, as the instrument manufacturers described the methods as 'calibration free' at the time of the study, so were only operated with the manufacture factory calibrations. "*

Instruments were in the field from 22 Aug to 2 Sept. Is there a reason that only 23 Aug and 29 Aug is used for the intercomparison study and not the full period? There are inconsistences in the dates in several places throughout. For example, Figures 8 and 9 and Table 4 suggests a start date of 22 Aug instead of 23 Aug. There are also inconsistencies in the dates in some of the figure captions and table headers in the SI.

Though most instruments were in present the field between the 22[nd] August to the 2[nd] September, not all instruments were monitoring ambient air, as other tests were being undertaken. All instruments were operational between the 23[th] August to 29[th] August and therefore this period is the focus of our study. We have removed the periods out with that period. The following text has been reviewed and revised to make clearer the operational periods. In Section 2.4 Data Analysis the following text has been added:

*"Though instruments were deployed for a longer period, for the purpose of this study only the period of the 23[rd] to the 29[th] August is studied unless otherwise stated, as not all instruments were operational at the start and end of the campaign."*

In addition Section 3.1 was updated to reduce the period of meteorology discussed to cover only the period of the 23[rd] to 29[th] August, as well as reducing the period show in Figure 2 (see below). The text describing period now states:

*"As well as reporting $NH_3$ gases, the MARGA also reported the $PM^{2.5}$ water-soluble inorganic species. Prior to fertilisation on the 23[rd] August $PM_{2.5}$ was dominated by sea salt (NaCl), but during the interim was dominated by secondary inorganic aerosol, which coincided with a drop in wind speed and a reduction in the relative humidity on the 24[th] August."*

[Figure]

1.

2. **Figure 1. Summary of the meteorology and the inorganic composition of water soluble PM₂.₅ at Easter Bush during the intercomparison campaign from the 22 August 2016 to 02 September 2016. The grey shaded line is the period where urea was applied to fields. Blue bars are precipitation and black line is the temperature.**

A major concern for Section 3 is that the bulk of the analysis relies on comparing each instrument to an ensemble median rather than a true, validated reference (e.g., Figures 8 and 9). While this might be the best approach for this case study, it should be specified in the text that the median is not an independent variable (i.e., it is by definition driven by the range of observed values from the different instruments). If the ensemble median is not independent, then the correlation analysis might be better described using 2-sided linear least squares fits (aka. orthogonal distance regression). See additional specific comments below.

We agree that ideally instruments should be analysed against a validated reference but unfortunately we were unable to operate either the US EPA or the EMEP methodology during this study. It is however noted that both the US EPA and EMEP methodologies are for lower temporal resolution and there is still no validated reference method for hourly resolution. We have already stated that ensemble median is likely to be biased if the instruments are biased in Section 3.5 but have now updated the text to state the following:

*"In the absence of a 'perfect' reference instrument, Figure 8 presents the summary of the instrument comparison to the ensemble median for NH3 <10 ppb. It is acknowledged that the ensemble median could be biased if the majority of instruments are biased."*

We have calculated simple linear regressions and not orthogonal regressions because we believed an orthogonal regression is not fully appropriate either. Whilst the ensemble median is not independent, it should hopefully be a closer approximation of the true concentration as it averages out random

uncertainty effects on the various instruments. Strictly speaking the x-axis values neither fulfil the requirements of a independent, equal measurement nor that of an independent reference variable. In addition, this was the approach previously used by von Bobrutzki et al. (2009).

Since, we are not interested in the p-values associated with the regression and are instead primarily using the lack of fit (via the $R^2$) and the fit coefficients for understanding the performance of the instruments we do not believe it is actually so important. By contrast, we did opt for orthogonal regression for the comparison of individual instruments, e.g. Section 3.8.

Specific Comments:

Title: Could the title be more specific to the experiments and findings? Maybe something like "Exploring best practices for gas-phase NH3 monitoring: An in-situ comparison of 13 instruments in Southeast Scotland".

Many thanks to the reviewer for the proposed title, we however would propose that we keep the original title as this study does not provide the evidence that is required for the best practices for $NH_3$ measurements. This study instead explores the current deployment practices by instrument operators and the effect this has on performance. We do not think that the findings are specific to the site where the comparison was performed.

Abstract: The Abstract could be streamlined a bit by relocating some of the context to the Introduction. It could also be added that NH3 is an unregulated pollutant in many countries. Listing each instrument make and model could be solely addressed in the methods section.

We have now reviewed and edited the abstract and highlighted as requested that $NH_3$ remains unregulated in many countries. Below is the revised abstract:

*"Ammonia ($NH_3$) in the atmosphere affects both the environment and human health. It is therefore increasingly recognised by policy makers as an important air pollutant that needs to be mitigated, though it still remains unregulated in many countries. In order to understand the effectiveness of abatement strategies, routine $NH_3$ monitoring is required. Current reference protocols first developed in the 1990s, use daily samplers with offline analysis but there have been a number of technologies developed since, which may be applicable for high time resolution routine monitoring of $NH_3$ at ambient concentrations. The following study is a comprehensive field intercomparison held over an intensively managed grassland in South East Scotland using currently available methods that are reported to be suitable for routine monitoring of ambient $NH_3$. In total 13 instruments took part in the field study, including commercially available technologies, research prototype instruments and legacy instruments. Assessments of the instruments' precision at low concentrations (< 10 ppb) and at elevated concentrations (maximum reported concentration of 282 ppb) were undertaken. At elevated concentrations all instruments performed well on precision ($r^2$ >0.75). At concentrations below 10 ppb however, precision decreased and instruments fell into two distinct groups with duplicate instruments split across the two groups. It was found that duplicate instruments performed differently as a result of differences in instrument setup, inlet design and operation of the instrument.*

*New metrological standards were used to evaluate the accuracy in determining absolute concentrations in the field. A calibration-free CRDS Optical Gas Standard (OGS, PTB, DE) served as an instrumental reference standard, and instrument operation was assessed against metrological*

*calibration gases from i) a permeation system (ReGaS1, METAS, CH) and ii) Primary Standard gas Mixtures (PSMs) prepared by gravimetry (NPL, UK). This study suggests that though the OGS gives good performance with respect to sensitivity and linearity against the reference gas standards, this in itself is not enough for the OGS to be a field reference standard because in field applications a closed path spectrometer has limitations due to losses to surfaces in sampling $NH_3$, which are not currently taken into account by the OGS. Overall, the instruments compared with the metrological standards performed well but not every instrument could be compared to the reference gas standards due to incompatible inlet designs and limitations in the gas flow rates of the standards.*

*This work provides evidence that though $NH_3$ instrumentation have greatly progressed in measurement precision, there is still further work required to quantify the accuracy of these systems under field conditions. It is the recommendation of this study that the use of instruments for routine monitoring of $NH_3$ needs to be set out in standard operating protocols for inlet set-up, calibration and routine maintenance, in order for datasets to be comparable."*

P3, L15. Flip phrase to read as "not all NH3 is captured"

The text has been updated as suggested.

P3, L35. There are many similarities between this work and a prior report by von Bobrutzki et al., 2010, which is referenced several times throughout this work. From the text it seems that the main advantages of this intercomparison are the addition of newer instruments/technology and the evaluation of traceable gas standards. Are there other advancements in this work in terms of the experiment objectives, experiment design, and application of lessons learned from von Bobrutzki? For example, this work uses pairs of identical instruments outfitted with different inlets to characterize artefacts due to the setup itself. This is a unique feature of this experiment that ought to be clearly outlined as a focus of this paper in the intro. The experiments related to this comparison and any additional setup should be clearly described in the methods section.

We thank the reviewer for highlighting this gap. In the introduction we have revised the last paragraph of the text to outline some of the unique features of this study.

*"This study reports a field intercomparison within a European Joint Research Project (EMRP), Metrology for $NH_3$ in ambient air (MetNH3,(Pogány et al., 2016)). MetNH3 aimed to improve comparability and reliability of ambient air NH3 measurements by achieving metrological traceability for $NH_3$ measurements in the amount fraction range 0.5-500 ppb from primary certified reference material (CRM) and instrumental standards to the field level. In this study 13 instruments, including commercially available technologies, research prototype instruments and legacy instruments were deployed and exposed concentrations from background (<10 ppb) to elevated (>200 ppb). The instruments included: an online ion chromatography system (MARGA, Metrohm-Applikon,NL), two wet chemistry continuous flow analysis systems (AiRRmonia, Mechatronics, NL), a photoacoustic spectrometer (NH3 monitor, LSE, NL), two mini Differential Optical Absorption Spectrometers (miniDOAS; NTB Interstate University of Applied Sciences Buchs, now part of "Eastern Switzerland University of Applied Sciences, CH and RIVM, NL"), as well as seven spectrometers using cavity enhanced techniques: a Quantum Cascade Laser Absorption Spectrometer (QCLAS, Aerodyne, Inc. US), Picarro G2103 Analyzer (Picarro US), an Economical $NH_3$ Analyser (Los Gatos Research, US), Tiger-i 2000 (Tiger Optics, US) and LaserCEM® gas analyser (AP2E, FR). In this study we evaluate the precision*

*of these instruments by comparing their data to the ensemble median and studying the between instrument variability, including those operated on a common manifold, as recommended by von Bobrutzki et al. (2010). The importance of set-up on precision is also considered through the use of duplicate instruments with different inlet design. Metrological methods developed under the MetNH3, are also evaluated under field conditions as standards for determining the accuracy of the instrumentation deployed, as previous studies of the metrological applications have focused on laboratory studies (Pogány et al., 2016, 2021). Overall we discuss recommendations for future LTMHTR ambient NH3 measurements, considering instrument capabilities and sampling set-ups to achieve high precision for use in routine monitoring of $NH_3$ and where further developments are still required in determining the accuracy of ambient $NH_3$ measurements."*

In addition, we have also revised the text in Section 2.3.2 to outline how instrument set-ups were modified in order that an evaluation could be undertaken against the ReGAS. We have moved information from Section 3.9 and added additional information into the method Section 2.3.2 in order to clearly describe the set-up. Section 2.3.2 reads as follows:

*"The instruments that were evaluated against the ReGaS (i.e. the LSE, Picarro #2, LGR#1, LGR #2 and the Tiger Optics) were transferred from the Pyrex manifold to the Teflon manifold for this purpose. Due to the maximum flowrate of the ReGaS1 (5 l min$^{-1}$) the LGR #2 did not use its external pump but was reliant on the internal pump of the instrument, so had a flow rate of 0.25 l min$^{-1}$ which equates to a residence time of 11.0 s for the inlet, which is slower than LGR #1. The system was set for the following concentrations in sequence for the duration of 31 minutes each: 0 ppb, 9.98 ppb, 24.39 ppb, 39.71 ppb, 2.95 ppb and 1.02 ppb. Unfortunately, the data of the following instrument was excluded from the analysis; the LGR #1 concentrations remained low even at elevated concentrations indicative of a fault and the Tiger Optics reported 0 ppb as it could not detect concentrations below the 10 ppb detection limit. As a result in this study only information from the OGS, LSE and LGR#2 are evaluated against the ReGAS."*

In Section 3.9 we have now revised the text to also mention the average concentration reported from the blank at the start of the comparison. The text reads as follows:

*"As previously stated the LGR #2, LSE and OGS were compared to the ReGaS1 calibration system. For 0 ppb it was found that the instruments reported the following average concentration during the of LSE: -0.77 ppb, LGR #2: 0.16 ppb and the OGS: 0.14 ppb (refer to Figure 13). The LGR #2 performed poorly compared to the other instruments. H however it is noted that the instrument was operated on a lower flowrate compared to that used during the field campaign (Table 1) resulting in a slower time response. It is evident in Figure 13a that the LGR #2 was still stabilising and had not reached equilibrium. LGR #1 was part of this calibration; however, it developed a fault therefore no results are reported here."*

Figure 1. This is a nice photo of the field site. It gives a lot of perspective for the experiments. It would help to add a wind rose or an arrow to show the predominant wind direction during the study period.

We have added a wind rose for the period of the 23$^{nd}$ to the 29$^{th}$ August to give perspective of the main wind directions, whilst the comparison was undertaken. Please refer to the annotated manuscript to see the updated figure.

Table 1 is missing accuracy, precision, and range information for the miniDOAS #1 – add symbols like n/a or (-) unknown.

Table 1 has now been updated for miniDOAS#1.

The LGR #1 and LGR #2 response times look to be flipped. The LGR#2 with the higher flow rate should have a faster response time of 1 s.

The reporting time in Table 1 is the interval for reporting concentration by the instrument and not the response time. We have now added the following text to the caption to make clear why there are differences between the same instruments

"(Note: Reporting time by the instrument was selected by the operator)".

There are 13 instruments compared in this study, yet there are 15 rows in the table. Should the OGS and the ALPHA sensors be separated from this table or distinguished in some way. Maybe this is simply fixed with a footnote to clarify the usage of the OGS and ALPHA samplers. It would also be helpful to further explain the dependency of the Picarro#2 and the OGS in a footnote in the table. Further, the acronym OGS has not been defined yet in the manuscript.

The OGS has moved to the bottom of Table 1 to make a clear definition from the 13 instruments to ensure that it is distinct from the instrumentation being assessed, please refer to the table for further details. The following footnotes have been added to the end of Table 1 to describe the ALPHAs and OGS presence:

"[2] Reference method used to assess the homogeneity between mini DOAS and reflectors, [3] Metrological developed algorithm which uses Picarro #2 (refer to Section 2.3.1)"

Table 2. Add another row to specify if the inlet components are heated or not. For instruments that do have heated inlets, at what temperature are they maintained?

This has been added to the Table 2.

For consistency, change "N" to "No" for the AiRRmonia#1 filter .

Apologies for this error. This has been corrected.

Is "diameter" meant to be the inner diameter of the tubing or the nominal outer diameter? The i.d. of the tubing will be the most relevant for your residence time calculations, so that could be the better parameter to include in this table. In either case, please clarify.

This is the inner diameter. The text in Table 1 has now been added.

Is there a reason that the manifold inlet and manifold itself are made from different

materials?

The manifolds available to the authors were limited due to the number of instruments and high air flow required.

Is there any research on how NH3 sticks to uncoated Pyrex surfaces?

We are not aware of any research on how $NH_3$ sticks to uncoated pyrex surfaces. As stated above the manifold used was due to the availability of authors at the time. Ideally in future work a manifold made with a material to minimise the losses to manifold walls would be recommended.

I find it interesting that all of the instruments on the common inlet use a filter. Was that

planned?

This is an interesting observation. No this was not by design, but partly reflects that all instruments that use filters have a similar need for inlet lines and flow rates. We have now updated Section 2.2 regarding the set-up to highlight that it was not by design the instruments on the manifold.

*"All participants were given the opportunity to sample from a common high flow inlet, where applicable"*.

Some additional explanation of the parameters for the unique inlet system associated with the QCLAS are likely needed in Table 2 and Section 2.2.3. This instrument setup is unique in that it uses a heated inertial inlet with a critical orifice to separate particles from the airstream. It also requires a rather large capacity scroll pump to create a sample flow rate of 13 l min-1. It is also important to distinguish the size of the critical orifice (~1 mm) located inside the inertial inlet compared to the size of the tubing (typically 3/8" o.d., ¼"i.d.).

To highlight the point that though no filter is used an inertial inlet is used instead to remove particles in the table a footnote has been added for the QCLAS when it states no for a filter. The additional footnote is as follows:

*"[1]An inertial inlet is instead of a filter was used to remove particles from the air stream. It is estimated to be about 90% efficient, depending on particle size. Refer to section 2.2.3 and Roscioli et al. (2016) for further details."*

P8, L9: How often are the passive ALPHA samplers collected and analyzed? Please include this information in the description in the methods section.

The following text has been added on the deployment and analysis of the ALPHAs. *". The ALPHAs were exposed in triplicate, with a shelter, at each position for two periods; Period 1: 22/08/2016 16:35*

*to 29/08/2016 16:29 and Period 2: 29/08/2016 16:29 to 05/09/2016 17:42. Chemical analysis was undertaken using flow injection conductivity analysis using an AMFIA (ECN, NL)".*

P8, L34: It isn't clear here why the correction was not included. I think this is explained later in Section 2.3.1. It would help to add a reference here to the explanation in the later section.

The paper of Martin et al. (2016) was published only a month before the field campaign and it was therefore not feasible to upgrade the instruments before the field campaign. We have modified the text as follows:

*"..however, this correction was not yet released by the manufacturer at the time of the field study and thus was not yet implemented in the participating instruments."*

P8, L37. I suggest moving the last sentence of this paragraph to Section 2.3.1 where the OGS is described. This information gets lost here and is better served in the other section.

We have moved as suggested the last sentence to Section 2.3.1.

P9, L16: Background subtraction from routinely measured zeros seems to be another unique feature of this instrument's operations. Were any other instruments routinely zeroed throughout the measurement campaign?

It is correct this is a unique feature of the QCLAS to do background subtractions. None of the remaining instruments (which are mostly considered "plug and play"), were set up with user-managed background subtraction. It is only an option with the Aerodyne QCLAS.

It seems like this would be a fairly important step for all instruments to accurately report $NH_3$ mixing ratios. It is also interesting that N2 was used for the zeros instead of zero air. While using $N_2$ should not impact a zero calibration, a prior study showed that span calibrations on top of $N_2$ compared to zero air produced a spectroscopic artefact up to 10% (Pollack et al., 2019;

https://amt.copernicus.org/articles/12/3717/2019/).

The objective of the background subtraction with the aerodyne instrument was to remove the background spectral features and any optical interferences such as fringes. We agree that $N_2$ is sufficient for the zero calibration and no span calibrations were performed. The note that it can result in a spectroscopic artefact when used for span gas dilution is not applicable here, but it is interesting and the magnitude would likely depend no the exact setup of the QCL, the spectral line used etc. However, the reference does not seem to address this issue?

Section 2.3.1: What wavelength does the OGS operate at? What is the uncertainty of the absorption cross section (or line strength) at this wavelength? How does this impact the overall uncertainty of the OGS calibration system?

We have added the following with regards to the uncertainty with regards to the OGS:

*"An expanded uncertainty of 1 % could be achieved for the line intensity of the two strongest $NH_3$ lines, which allowed the total uncertainty of the retrieved $NH_3$ concentration to be decreased down to 3% (k=2, 95 % confidence interval). Further important contributors to this uncertainty are spectral line broadening coefficients or the choice of the fitted spectral model."*

In addition, we have added a sentence to Section 2.3.1 to emphasize that the OGS uses the same hardware (I.e. same spectral range, same absorption lines) as the Picarro G2103 instrument. The exact spectral range is 6548.50 to 6549.25 $cm^{-1}$, corresponting to 1.527 μm.

*"The OGS essentially extracts and re-evaluates the Picarro raw spectra, hence it uses the same hardware but a completely different evaluation and different spectral reference."*

P11, L36: "associated with long stabilization times"

Many thanks for the suggested correction.

Section 2.3.2: Please specify the carrier gas used. N2 versus air can have different effects on $NH_3$ permeation devices and spectroscopic artefacts.

Apologies for this omission. The $NH_3$ permeation system used a zero grade synthetic air. The details are now outlined in the section as follows:

*"It employs as the $NH_3$ source a permeation device in a temperature-controlled oven and two dynamic dilution steps with mass flow controllers to obtain the required amount fractions using zero grade synthetic air (SA) (158283-L-C, BOC). Additionally, a commercially available gas purification cartridge (Microtorr, model MC 400-203V SAES Getters, Pure Gas Inc.) was used for additional synthetic air purification. According to the product specifications, the outflow of purified SA should contain less than 100 pmol/mol $H_2O$ and less than 100 pmol/mol $CO_2$. The content of acids, bases, organics and refractory compounds in the outflow should not exceed 10 pmol/mol. The Microtorr purification system is based on inorganic sorbent materials and operates at normal ambient temperature (no heating or cooling required). The connectors of the cartridge are made of stainless steel. "*

Figure 2. Please add something (e.g., shaded bar, horizontal arrow, red dashed lines) to indicate the period of data used for the instrument intercomparison (which I think is 23 Aug to 29 Aug). Is there a reason that the remaining data in the timeseries up to 02 September was not included in the intercomparison?

To avoid confusion the period presented has been reduced to the 23rd August to the 29th August which only encompasses the period that data was used for comparison study.

Figures 3 and 5 make me wonder how the instruments were time aligned prior to correlation analysis. Were inlet delays account for prior to averaging to 1 hour? Please clarify in the text.

In Figures 3 and 5 no alignment was carried out to account for the time response of the instrument. The text has been revised in Section 2.4 on data analysis to make clearer the handling of the data.

*"To facilitate direct comparisons, data were averaged to 1 hour, unless stated otherwise, to match the reporting time of the slowest instrument."*

Figure S2. This is an interesting figure. It seems as if there was less atmospheric stability during the intercomparison period, yet much larger stability during the period immediately after the study period (29 Aug thru 2 Sept). How do the instruments compare during the more stable period? Are there any observable differences? This would be an interesting comparison to add to this study that would be within the scope of this study. Also, there is a typo in the caption; the date range should be from 22/08/2016 to 03/09/2016.

To stop confusion we have modified the Figure S2 to only show the period from the 23rd August to 29th August.

Figure 4 could be moved to the SI if you feel this paper is too lengthy.

Figure 4 is important for understanding how a passive sampler performs compared to a routine instrument, as well as information on the homogeneity of the path of the miniDOAS instruments for the whole period. We will keep it within the main text.

Section 3.4: Can you comment on the effects of a heated vs. unheated inlet? See Ellis et al., 2010 for reference. Some additional comments about the utility of a heated inertial inlet for filterless separation of particles could be included here.

We cannot comment on the basis of the experiment as performed as there are other factors at play such as residence time, flow within inlet we would not be able to disentangle the effect of the heated inlet here as only the LGR #2 and the QCLAS were heated. Strong heating is likely only really a viable option downstream of the removal of the particle phase as in the QCLAS setup as it could otherwise result in evaporation of volatile aerosol components such as ammonium nitrate.

Section 3.3: Does the ensemble median include LGR#1? If yes, how does the ensemble median and related statics change if this measurement is excluded?

The ensemble median presented does not include LGR #1 due to the poor performance. This is stated in line 7 on p 19 of the revised text. We have updated the figures 8 and figure 9 to make clear that both the LGR #1 and Tiger Optics were excluded from the ensemble median calculations. Revised text is below:

*"Figure 2. Intercomparison of hourly instrument averages from 22/08/2016 to 29/08/2016 to the ensemble median (excluding LGR #1 and Tiger Optics) when the median <10 ppb NH₃. Green circles are the data removed after applying a met filter (<0.8 m s⁻¹ and |(z-d)/L| > 0.1). The green and black legends are the correlations of the unfiltered data and the filtered data, respectively. The solid black line is the 1:1 line."*

*"Figure 3. Intercomparison of instruments (hourly) averages from 22/08/2016 to 29/08/2016 to the ensemble median (excluding LGR #1 and Tiger Optics) when the median is equal or greater to 10 ppb NH₃. Data were filtered for low wind speed and stable/unstable conditions that could have led to inhomogeneity at the site."*

Figure 6. Change CV limit to 20% in caption to be consistent with figure and discussion

text. Apologies for this error, the text has now been corrected.

P18, L13: Is the response time truly different under ambient conditions? Without the same level of fine structure in Figure 7b as in Figure 7a, it is difficult to accept the levels of smoothing applied to the DOAS under ambient conditions to match the profiles of the AiRRmonia and Picarro instruments. Can you include another trace in 7b to show the DOAS signal with the same level of smoothing as in 7a? This would help highlight whether additional smoothing of the DOAS is indeed needed to match the features of the AiRRmonia and Picarro instruments under ambient conditions.

We have now updated Figure 7 to include an additional line in Figure 7b which is the smoothing applied in 7a) to demonstrate to the reader the loss in structure and reduced concentrations detected at ambient concentrations.

[Figure]

Figure 4. Smoothed time series of miniDOAS #1 (black dotted line) calculated from the 1-minute miniDOAS #1 signal (grey line) until fitting by eye the time series is similar to the reporting data of individual instruments. a) Elevated concentrations following fertilisation (fertiliser applied from 11:00 23/08/2016) b) ambient concentrations. The blue line on panel b) represents the smoothed time series using the time-response derived from elevated concentrations from panel (a) to visualise the significant additional smoothing encountered under ambient concentrations.

Figure 8. The met filter looks like it could have induced some bias in some of the fits. The met filter was applied to eliminate low wind speed and unstable conditions that could have led to inhomogeneity between the inlets at the field site. But didn't the ALPHA samplers indicate that there was homogeneity during the study period? What do the fits look like without the met filter? At a minimum, you should comment in the text about any differences in fits with and without the met filter applied.

The data presented in Figure 8 includes all data below 10 ppb with the open circles being the data removed after the met filter was applied. The green legend provides information of the comparison for all data, whereas the black legend is the filtered data as outlined in the caption. So the bias is reflected in the difference of the two fits. It is likely that stable conditions that occurred would lead to inhomogeneity of the concentration field and is where the highest concentrations are observed, which the LTMHTR instruments reported. As we have already stated in the text there was an observed improvement in the precision by applying a met filter ($R^2$) however we acknowledge the slope and intercept did alter through the application of the filter on some instruments. For example, for the LGR #2 the slope reduced and the intercept increased due met filter being applied. We have now revised the text to include the following in the manuscript:

*"It is noted, that the slopes and intercepts changed in applying a meteorological filter. In general instruments with faster response found their slopes reduced, whereas the reverse was observed for the instruments with the slower time response. With the exception of the $AP_2E$, slopes after filtering were closer to unity and with the additional exception of Picarro #2 the intercepts decreased. For the remainder of this discussion however we will only discuss the filtered data."*

In addition we have modified Figure 8 by colouring the open circles green, in order that it easier for the reader to relate to the green legend:

[Figure]

**Figure 5. Intercomparison of hourly instrument averages from 22/08/2016 to 29/08/2016 to the ensemble median (excluding LGR #1 and Tiger Optics) when the median <10 ppb NH₃. Green circles are the data removed after applying a met filter (<0.8 m s⁻¹ and |(z-d)/L| > 0.1). The green and black legends are the correlations of the unfiltered data and the filtered data, respectively. The solid black line is the 1:1 line and red dashed line is the fit.**

Section 3.6: What would you consider to be a reasonable deviation from 1 for the slopes? How does this compare to the deviation in the ensemble or with the reported measurements uncertainties in Table 1? For example, if the spread around the ensemble median is 20%, would slopes ranging from 0.80 to 1.20 be considered good?

20% spread around the ensemble median slope (0.8-1.20) would not be considered good from an analytical chemistry or an air quality monitoring perspective, i.e. it would only be within "indicative" agreement if the x-axis had a traceable reference method. Ideally, the ammonia measurement community should be working towards <10% spreads, with world class results <5% for ambient ammonia measurement intercomparisons. The 20% spread means the quantitative concentration values must be regarded with caution, particularly if used together. This said, the absolute value of the discrepancy from the ensemble median must be treated with caution as this chosen reference itself may be biased. It is worth noting again that all instruments measured similar temporal variability.

It would also be interesting to see how the fast instruments compare with an ensemble average of only the fast instruments (like on a 1 min average timescale). Was this something you tested? Can you comment on whether the results would be different?

We did not compare all the instruments at a higher temporal resolution as it is beyond the scope of this work and as demonstrated the inlet is a limiting factor for instruments that have a high reporting resolution. The purpose of this paper is to evaluate the performance for hourly temporal resolution, which an averaging period typically reported within air quality networks.

Data in Figures 8 and 9 are split into NH3 > 10 ppb and NH3 < 10 ppb. This was done to be able to best compare the NH3 < 10 ppb data with the findings in von Bobrutzki. Please include additional discussion of how this work compared with the findings in the prior work.

We thank the reviewer for this suggestion. We have added the following text to Section 3.6

"The work here comes to a similar conclusion with regards to the slope for the QCLAS, as von Bobrutzki et al. (2010) who also reported a slope less than 1 when compared to the ensemble median of the partaking instruments. The two studies however differ in that there it is not a clear split on the performance of the wet chemistry instruments. In von Bobrutzki et al. (2010) found all the wet chemistry instruments had a slope >1, whereas in this study at > 10 ppb the AiRRmonia#1 had a slope >1, whereas the reverse is observed for MARGA and AiRRmonia #2. This potential highlights how performance varies with set-up, but could also reflect further progress in the development of the spectroscopic techniques since the 2010 study."

Does separating the data points by NH3 mixing ratio prior to intercomparison analysis generate any bias? It would be helpful to see the results of the intercomparison fits using all data, which could be a nice figure in general to include in the SI.

We separated the concentrations as higher concentrations are likely to effect the fit of the regression and we wanted to determine the performance of systems for ambient conditions. We have added as suggested the comparison for the whole concentration range to the supplementary material to allow the reader to compare the differences. We have update the main manuscript to state:

"The intercomparison is presented for the full dataset, however, in Figure S4 in the Supplementary Material.".

At the risk of cluttering the figure, it also seems appropriate to also include the actual fits in each plot.

We thank the reviewer for this suggestion. We have now updated both Figures 8 and 9 to include the fit and revised the caption text of both figures.

P24, L1: Are the least squares regressions 1-sided or 2-sided? Please specify. See general comment above.

We have undertaken a linear regression and not an orthogonal regression. Refer to response to general comments for further information.

P24, L6: It seems as if the LGR#1 instrument was having some issues during this experiment. Would it be better to exclude the data from LGR#1 from the paper altogether? Do you have specific reasons for keeping it in this intercomparison? Was the LGR#1 was included in the ensemble median? Please clarify your reasoning in the text.

It is already stated that the LGR was not included in the calculation of the ensemble median, please refer to Section 3.5. Though the LGR #1 did perform well during the experiment, the point being made by the authors is that 1) set-up is vital and 2) quality assurance processes need to be put into place if the plug and play instruments are to be integrated into routine monitoring across the world. If it had not been due to comparing to other instruments, it is unlikely that the operator would have identified an issue. We have reviewed our discussion on this point in Section 4.1 to highlight this risk.

*"Unfortunately, an assessment of the drift of instrumentation studied using the ReGAS was not possible during this study. It is recommended that such an assessment is undertaken in future studies. However, it provides evidence that regular calibration span checks are required to be carried out to determine the accuracy and precision of instrumentation, especially instrumentation considered to be plug and play instruments which are thought to be stable in time. If it had not been for the comparison with other instruments, the poor performance of LGR#1 may have taken longer to identify if operating in isolation."*

Figure 10. What does the color scale correspond to? Please clarify in the caption. Add labels "a" and "b" to plots.

Apologies if this is not clear. We have now added a and b labels to the figure and the following to the caption. *"The colour scale relates to the magnitude of the correlation coefficient."*

P26, L15: What the was the temperature of the LGR#2 inlet? Was it high enough to thermally dissociate enough NH4NO3 to impact the measurement? Gentle heating (<40 degC) might not have a huge impact on the measurement on the sampling timescale (e.g., Fig 4a in Huffman et al., 2009; https://doi.org/10.5194/acp-9-7161-2009).

The LGR #2 inlet was heated to stabilise between 40°C to 70 °C, we have now added this information into the method section 2.24. Volatilisation of the aerosol is minimised where its residence time in the inlet is very short; however, there is the potential for some aerosol that collects on the inlet walls to volatilise and thus result an artefact.

*"During the campaign two LGR instruments were used; LGR #1 used its internal pump (0.25 l min$^{-1}$) and LGR #2 used an external pump (2.3 l min$^{-1}$); in addition, the inlet for LGR #2 was heated to stabilise between 40°C to 70 °C (refer to Table 1 and Table 2 for further details)."*

Figure 11. Since the instruments compared in this figure all have relatively fast response times (1 minute or less), would it be more realistic to compare them on a 1-minute timescale?

As the analysis of the normalised distribution and coefficient of divergence was performed for all instruments, the data had to be averaged to the slowest reporting time which is 1 hour and so it was only appropriate to present hourly averages. This is also the resolution at which air quality monitoring data is often reported. As seen in Figure 15 by reporting higher temporal resolution the data would have more scatter and be no longer comparable to the ND or CD presented. Future studies can focus down on the higher time resolution as needed by a particular application.

Figure 12. Please clarify what height means in the caption. Does it have units? It would be helpful as a quick reference to include labels in the figure about what the different grouping are. You could easily add this by changing the outline colors to match a legend or a description in the caption?

We have now updated caption text to explain the height and the coloured boxes.

*"Figure 6. Euclidean distances between instruments based on their coefficients of divergence for the period of the 23/08/16 00:00 to 29/08/16 01:00 based on their hourly averages. Height relates to the order at which the clusters occurred. The red boxes indicate the instruments that are clustered together."*

P29, L11: Do you mean Figure S4? Apologies for this error. This should have been Figure S4. This has now been corrected in the manuscript.

Table 4: Fix table header to correspond with proper figures in the SI (S4 and S5). Add a footnote to highlight the relationship between Picarro#2 and OGS.

Apologies for the error. The figures have been corrected and the following footnote has now been added to Table 4:

*"*Note: The spectra from the Picarro #2 are used to produce the OGS"*

P34, L17: Based on lessons learned from this study, can you provide a recommendation for how often routine calibrations (zeros and spans) ought to be performed if one or more of these measurement techniques are used at a surface monitoring network site?

Though we have been able to highlight the importance of frequent checks with zeros and spans, we have been unable in this study to demonstrate the frequency at which these should be undertaken. Further work with a longer comparison study would be required to determine the 'drift' which is likely to impact the accuracy and precision of instruments for surface monitoring networks.

Section 4.3: Based on lessons learned in this study, can you provide additional recommendations about inlet setup? It is not surprising that instrument manufacturers do not specify a schedule for calibrating and servicing, as it is largely dependent on how, where, and under what conditions the instrument is utilized. Instead, there could be a list of indicators to watch for that could signal a user to perform routine maintenance. For example, a prior works (Pollack et al., 2019; Ellis et al, 2010) showed that increases in the response time to a step change in NH3 from a calibration source was a good indicator of when the instrument inlet needed to be cleaned.

The drift of the instruments should be monitored (e.g. daily or weekly) by operating a standard. However, this was not done in the study as most instrument systems (instrument plus inlet and pumps etc.) were not configured for this to be a routine activity. In a longer term field intercomparison or network study initiating operations with a high frequency of quality assurance (zero, span, response time) would enable the development of maintenance schedules to maximise measurement quality. We updated Section 4.2 to reflect the reviewer's comments to give further recommendations with regards to inlet design:

*"It therefore would be the recommendation to operate, where an inlet is required, to minimise the wall interactions by minimising the length of inlet used, residence time and surface to volume ratio of the inlet "*

In addition we have updated text in Section 4.3 to include the following:

*"It would be advisable that a standard is also used on a frequent basis to determine the contamination of the set-up, as previously demonstrated by Ellis et al. (2010) and Pollack et al. (2019), who observed that inlet contamination can be identified via an increase in the calculated time response. In addition, a standard inlet design needs to be agreed (where applicable). Evidence from the Picarro and AiRRmonia set-ups in this study (Figure 14) would suggest that this can lead to losses of information of the temporal pattern of NH$_3$. Consideration is also required to determine if passivation of the inlet is valuable to routine air quality monitoring, as there is evidence that it can effectively reduce the interactions of NH$_3$ with the inlet walls (Roscioli et al., 2016). Open path techniques, such as DOAS, will benefit from the availability of zero-air facilities, where instruments can check their zero level on ammonia free air. Work on such a facility is ongoing."*

Figure 15. There are two panel b's in this figure.

Apologies this has now been corrected.

Figure S1. There are inconsistencies in the labels used in the caption (a, i, 2, 3). The time resolutions in brackets are missing for some instruments in the figure. What do you mean by "raw" data in the timeseries? "Raw" typically implies uncalibrated data. Please clarify.

We have now replaced the word raw with reported, added the reporting time series and corrected the labelling of the caption. See below the reviewed text for the caption:

*"Figure S1 Summary of the reported concentrations from the instruments on a linear scale for the whole range for a) instruments with individual inlet set-up b) instruments subsampling from the manifold and c) instruments on scaffolding. Number in brackets is the reporting time resolution of each instrument. The thick black line is the fertilisation of both fields and the black arrow indicates the point at which the laser position was changed on the LGR #1."*

---

## Author Comment (AC2)

Response to Anonymous Referee #2 on "In-situ measurements of NH3: instrument performance and applicability" by Marsailidh M. Twigg et al., Atmos. Meas. Tech. Discuss.,https://doi.org/10.5194/amt-2022-107-RC2, 2022

The authors thank the referee for the careful and constructive consideration of this manuscript. The answer is structured as follows: the comments from referee #2 are marked in black and the authors' response and changes the to the manuscript are written in blue.

**GENERAL COMMENTS**

The manuscript presented by Twigg et al. represents an important study towards improving the quality of in-situ ammonia ($NH_3$) measurements. Significant challenges still exist in obtaining accurate and precise $NH_3$ measurements, specifically in field observations where a large range of ambient conditions impact the performance of instrumentation. So far, there have been only a limited number of studies comparing different $NH_3$ measurement methodologies under field conditions. The authors present a comparison of 13 online instruments, spanning over the most important measurement techniques, and 1 type of passive samplers, making it to my knowledge the broadest $NH_3$ intercomparison study conducted up to date. The authors use different statistical methods to compare the instruments determining their precision and accuracy. The comparison of closed-path systems with the open-path miniDOAS as well as employing some instruments of the same type with different inlet systems, allows the authors to investigate dampening effects caused by inlet tubing surfaces. Finally, the authors also give advice on what future measurement setup should consider such as regular calibrations (which were missing in this study). The setup of the study, analysis of data and presentation of results was performed with great care, considering the major challenges of each instrument. Given the importance and uniqueness of the study, I suggest it to be published in AMT after addressing the comments below. The major parts where the manuscript in my view needs improvement is (1) the determination of the residence time and (2) the description of inlet tubing effects and the derivation of time constants.

We thank the reviewer for taking the time to review this study and providing the constructive feedback of the manuscript.

**SPECIFIC COMMENTS**

P. 1, L. 35: The CEN EN 17346 reference protocol for passive sampling is from the year 2020, therefore, I suggest to write "first developed in the 1990s" or similar.

We have updated the text as suggested.

P. 2, L. 13/14: Other studies, which investigate ammonia emissions on PM2.5 formation on a global scale include Gu et al. (2021) and Pozzer et al. (2017)

We added Gu *et al.* (2021) and Pozzer *et al.* (2017) to the references listed in line 23/24. We have now updated the text to include the following to the manuscript:

*"The same conclusions have been made by (1) and (2) on of mitigating PM$_{2.5}$ across the world.".*

P. 4, L. 19 ff (2.2. Instrumentation): I suggest to give similar details in the instrument description (where possible). Sometimes performance indicators (like detection limits) are given. 2.2.7 is the only

section where the instrument operators are named. Providing wavenumbers for the spectroscopic techniques in 2.2.4 and 2.2.5 would be beneficial.

To maintain consistency the text outlining the operators of the mini DOAS has been modified, as the purpose of the text was to highlight that are differences between the two instruments. As rightly pointed out we do not state anywhere else the other operators. The revised text in Section 2.2.7 states:

"*The two systems taking part in this campaign were miniDOAS #1 developed by the Bern University of Applied Sciences, Switzerland, in collaboration with Neftel Research Expertise and the miniDOAS # 2 developed by the Dutch National Institute for Public Health and the Environment (RIVM), Netherlands.*"

P. 4, L. 23: Do you mean ½" OD? In Table 2, the tubing diameter is given as 9 mm, which is less than ½". In Table 2, I suggest to specify that the value represents the inner diameter.

Apologies there was an error in the table. We have now updated the table to provide the inner diameter and have also updated all the other calculations in this table due to the error of using the outer diameter for some instruments.

P. 4, L. 24 ff & Table 2: When calculating the residence time from the flow rate and tubing dimensions, the pressure drop along the tubing and respective volume flow increase needs to be taken into account. Looking at some of the values in Table 2, is seems a constant volume flow was used for the calculation, is that correct? If so, values would need to be adjusted for the pressure drop, which would lead to faster residence times (assuming that given flow rates are at STP).

Most inlets did not have a substantial pressure drop in the inlet and therefore did not require the pressure to be taken into account for calculation of residence time, however there was an error in the diameter used which has now been corrected in table 2. However, the reviewer is correct to point this out as we neglected to take this effect into consideration for the inlet of the QCLAS which operates with a much reduced pressure over most of the inlet length. Therefore the residence time of the QCLAS has been recalculated and updated in Table 2.

The residence time of the manifold (+inlet) is given as 1.8 s, however, in Table 2 both add up to 1.62 s. Is that correct? In Table 2, it should be specified if flow rate was at STP or different temp/pressure conditions.

Apologies this is an error the text should read 1.62s and not 1.8 s. All measurements were reported in STP.

P. 13, L. 13: I suggest to state the sampling interval (of the 282 ppb) to be clear that this is based on the original sampling interval and not on the 1 hr resolution (as in Table S1).

Thank you for highlighting the fact that this is not clear. We have updated the text as follows to ensure to make sure the reader understands this is the instruments reporting resolution:

*"The LSE instrument reported the highest concentration with a maximum of 282 ppb (1 minute average)."*

P. 17, Figure 6 caption: NH3 mixing ratios is the median value (as reported above)? If so,it should be clarified.

In Figure 6, it is the average concentration (ppb) that is reported as this value is used in the calculation of the CV as stated in the caption.

P. 17, L. 21 ff: The response time due to inlet effects was determined by low pass filtering the respective instrument time series to match the unattenuated miniDOAS time series. While this approach seems sound, the retrieved response times seem very high (e-folding time ranging to more than 2 hrs). Another way to determine the response time is by investigating the exponential decay/rise during calibrations (when zero or span is applied). From Figure 13 a) the OGS (i.e. Picarro #2), seems to have a much lower e-folding time. E.g. the 63% (1-1/e) increase at the step change from 0 to 10 ppb (which would represent ambient conditions) seem to be reached within 1 min as opposed to the 49.5 min with the method used here. The dampening effects of ammonia in inlet systems is better described by a double exponential function with a fast time constant that represents the air exchange in the inlet tubing and a slower time constant that describes the adsorption and desorption effects (Ellis et al., 2010). E.g. Moravek et al. (2019) showed the evolution of the dampening over time using the double exponential function. Even if a single exponential decay/increase is used (-> the low pass filtering method yields a single time constant), the authors should include the time constant values from the calibration measurements, also as comparison with other studies. Also, it should be explained why the time constants of both methods (i.e. low pass filtering and visual fitting to miniDOAS time series) would be so different.

We agree with reviewer #2 that the ideally a double exponential function should be used to describe the time response of instruments, as the response observed is a function of volume and then adsorption/desorption effects. However, we cannot fit two time constants in our methods where we deteriorate a fast-response time-series to match each instrument. This would be an under-constrained mathematical problem.

We could indeed be able to fit a double exponential if we had controlled rise / decay measurements with the setups used for the ambient measurements, under ambient conditions, but unfortunately we do not.

The response observed in Figure 13 is indeed much faster. But here the instruments are used with different inlets, with a much drier calibration gas (which probably accounts for most of the better time response) and, for some instruments, with a modified flow rate (to match the limited output of the ReGAS). Therefore we conclude that the suggested approach by Reviewer #2, although in good in principle, is unfortunately not applicable in this case.

*"The instruments that were evaluated against the ReGaS (i.e. LSE, Picarro #2, LGR #1, LGR #2 and Tiger Optics) were transferred from the Pyrex manifold to the Teflon manifold for this purpose. Due to the maximum flowrate of the ReGaS1 (5 l min-1) the LGR #2 did not use its external pump but was reliant on the internal pump of the instrument, so had a flow rate of 0.25 l min-1 which equates to a*

*residence time of 11.0 s for the inlet, which is slower than LGR #1. The system was set for the following concentrations in sequence for the duration of 31 minutes each: 0 ppb, 9.98 ppb, 24.39 ppb, 39.71 ppb, 2.95 ppb and 1.02 ppb. Unfortunately, the data of following instrument was excluded from the analysis; the LGR #1 concentrations remained low even at elevated concentrations indicative of a fault and the Tiger Optics reported 0 ppb as it could not detect concentrations below the 10 ppb detection limit. As a result in this study only information from the OGS, LSE and LGR#2 are evaluated against the ReGAaS."*

P. 18, L. 10-12: Why was it discounted or what is your underlying hypothesis regarding the influence of a turbulent flow regime? I would have thought the opposite: if laminar flow conditions are increasing the dampening (due to segregation along the tubing cross section), this would explain why the Picarro #1 does not perform as well as the Picarro #2 (a similar statement is expressed by the surface/volume ratio).

It is agreed that a laminar flow regime will smear out high temporal variations concentrations but when reporting an averaged over a seconds to an hour this loss is not considered to be great. In general it is assumed that in using a turbulent flow this will lead to increase in wall interactions leading to further adsorption. We have revised the text as follows to highlight this discussion:

In section 3.4*: "This is not, however, the only controlling factor for the response of an instrument, as the Picarro #1 inlet is calculated to have a residence time for air of 2.9 s compared to Picarro #2 that has a residence time of 4.7 s (including the manifold inlet and manifold), but it still appears that the Picarro #2 performs better. It is postulated that as the surface area/ volume ratio for the Picarro #1 is two times the surface area/volume ratio of Picarro #2 (Table 2), resulting in more molecules interacting with the inlet walls leading to the observed a smoothed feature. It was discounted that turbulent flow was a controlling factor in the response time, as it would be expected that wall interactions would increase under a turbulent regime leading to greater losses (Table 2).*
*."*

P. 21, Figure 8: The y-axis offsets from the linear regressions are used to describe the accuracy (i.e. over- or underestimation) of the respective instruments at low mixing ratios. However, the offset may also result in uncertainties in the linear fit (lowest ensemble median mixing ratios are just under 2 ppb). Was this taken into account?

We agree that the y-axis offset is a result of uncertainties in the linear fit and not necessarily representative of the performance at low concentration. Interpretation of the intercept here is limited in order to understand the relationship between predicted $NH_3$ (from the ensemble median) and the concentration response of the instruments. Contamination, inlet losses, limits of detection and non-linear instrument response are the major issues, which will lead to linear slopes with significant offsets. We have added the text below to note that the discussion is indicative and that in each case reasons for the offset should be investigated, i.e. if an instrument is being deployed to make quantitative measurements.

*"Most instruments (Figure 8) had a slope less than 1 with the exceptions of the $AP_2E$, Picarro #1 and the LSE. The largest slope reported was from the $AP_2E$ (1.47) and it had the largest negative offset of -1.39 ppb. The y-axis offset is a result of uncertainties in the linear fit, and contamination/losses of $NH_3$ in inlet or the instrument. Interpretation of the intercept is here limited in order to hypothesise regarding the relationship between predicted $NH_3$ (from the ensemble median) and the concentration response of the instruments. Contamination, inlet losses, limits of detection and non-linear instrument response are the major issues which will lead to linear slopes with significant offsets. Negative intercepts are often indicative of losses of $NH_3$ either to the inlet or the instrument, however the large slope and high scatter ($r^2$=0.76) would also be contributing to the offset value. The instrument with the smallest offset is the QCLAS, which had an offset of 0.05 ppb but had a slope of 0.82 compared to the ensemble median. The largest positive offsets are seen in the Picarro #1 (with an offset of 1.05 ppb), miniDOAS #1 (0.74 ppb), LGR #1 (2.11 ppb), LGR #2 (0.65 ppb) and the AiRRmonia #2 (0.75 ppb). Working with the assumption that within the uncertainty of the regression, the positive offsets are real, the positive offsets in this case could be attributed to contamination in the inlet or in the case of the CRDs on the inline filters."*

P. 32, L. 24 ff: The authors use the results of the linear regression between instruments to described the precision. While the R2 value is influenced by the instruments' precisions, the slope would rather indicate the accuracy of an instrument in comparison to the ensemble median. I suggest to make clear that the precision only refers to the R2 value and not the slope.

We fully agree but may not have been clear in the text. We have now updated the text to highlight we are referring to the $R^2$ and not the slope.

*"In this study we assessed the precision by comparing the inter-variability between instruments and the variability against the ensemble median ($R^2$)."*

P. 34, L. 25 ff: Next to the tubing material, contamination of the tubing surface over time can influence the time response significantly (e.g. Moravek et al., 2019). Although the experiment was probably not long enough for it to have a major influence, this point may be included in the discussion.

We have now expanded the discussion on the impact of contamination of the inlet on the time response. The text now reads as follows:

*"Not considered in this study is the contamination of the inlet, which is likely to occur over time and has been discussed previously in the literature, though there still no recommendations for frequency of either cleaning or replacing inlets. Moravek et al. (2019), for example, demonstrated that for the QCLAS time response degrades with age (based over a 5 month study) due to contamination of the inlet; but even after cleaning the response time did not always return. As a result, some network instruments have already started to frequently replace their inlets. Twigg et al. (2015) replace their inlet at quarterly intervals for the MARGA currently operated in the UK to try to minimise contamination. Though not studied here, it would therefore also be recommended to consider the frequency that an inlet is either cleaned or replaced to account for potential loss of precision."*

In addition we have expanded the discussion in section 4.1 with regards to artefacts from filters with the following text:

*"It is however noted that the average concentration reported when compared to the ALPHAs (Table S1) that the instruments with filters tended to report higher concentrations compared to filter free methods (Table S1), supporting the suggestion that filters introduces an artefact. There is recent evidence that frequent filter changes are starting to be considered by network operators to limit artefacts in measurements. For example, (3) reported changing filters a frequency of between 2 weeks to monthly, dependent on atmospheric conditions for a CRD instrument."*

P. 34, L. 34 ff: Next to avoiding condensation, inlet line heating was shown to improve the time response in previous studies (e.g. Ellis et al., 2010). This should be mentioned here as well.

Thank you for recommending the Ellis et al. (2010) paper. We have added the following text to manuscript:

*"A previous study by Norman et al. (2009) demonstrated the importance of condensation on inlet lines and that care needed to be taken to ensure that condensation did not occur in the inlet. They recommended an optimal design might therefore include thermal insulation and, if possible, keeping inlets heated a few degrees above the ambient temperature, particularly also any sections that run within air conditioned measurement cabins. Ellis et al. (2010) also evaluated using a heated inlet, they found that heating the inlet line led to an improvement in the time response of a QCLAS. During this study, only the LGR#2 and the QCLAS used heated inlets. Caution however is required when heating an inlet, as if the temperature is too high, this will lead to the dissociation of $NH_4NO_3$ leading to an artefact."*

P. 35, L. 13 ff: Ideally, the humidity in the zero air would match the ambient air humidity levels. One way to produce zero air is by removing $NH_3$ from ambient air through a heating catalyst (without a drying cartridge or similar). Assuming that the water vapor is conserved, the zero air would then have similar humidity levels than the ambient air. Next to its influence on spectroscopy, humidity levels can also affect the adsorption/desorption processes in the inlet line. Therefore, having humidified zero air (at least to some degree) would be a beneficial for all instruments with an inlet line.

Many thanks to the reviewer for their comment. We have now reviewed and updated the progress towards a standard section of the manuscript. Please see below the revision to the text.

*"However, preparing humid gas samples with accurately characterised $NH_3$ concentrations in the ambient concentration range is challenging and work is required to develop standard methodologies to produce a humidified gas standard such as using a scrubber or heated catalysts. Pollack et al. (2019) provides a valuable study in evaluating these approaches. It would be advisable that a standard is also used on a frequent basis to determine the contamination of the set-up, as previously demonstrated by Ellis et al. (2010) and Pollack et al. (2019), who observed that the calculated time response alters due to contamination. In addition, a standard design of inlet needs to be agreed (where applicable), as evidence from the Picarro and AiRRmonia set-ups in this study (Figure 14), would suggest that this can lead to losses of information of the temporal pattern of $NH_3$. Consideration is also required to determine if passivation of the inlet is valuable to routine air quality monitoring, as there is evidence this limits the interactions of the NH3 with the inlet walls, as discussed in Roscioli et al. (2016). Open path techniques, such as DOAS, will benefit from the availability of zero-air facilities, where instruments can check their zero level on ammonia free air. Work on such a facility is ongoing."*

**TECHNICAL COMMENTS**

P. 7, Table 2: Typo in "AiRRmonia #1" & "Operated with a filter" -> "No" Corrected

P. 7, L. 14: "are sampled" instead of "is sampled" Corrected

P. 7, L. 15: do you mean "through which air is drawn and …"? Yes, may thanks for the correction Corrected

P. 7, L. 27: ("Erisman, 2001)" Corrected

P. 22, L. 7: Insert space before "ppb". Corrected

P. 25, Figure 10: "a)" and "b)" missing Corrected

P. 33, L. 13: Move comma to at the end of subclause (after parentheses in L. 14). Corrected

REFERENCES

Ellis, R. a., Murphy, J. G., Pattey, E., van Haarlem, R., O'Brien, J. M., and Herndon, S. C.: Characterizing a Quantum Cascade Tunable Infrared Laser Differential Absorption Spectrometer (QC-TILDAS) for measurements of atmospheric ammonia, Atmos. Meas. Tech., 3, 397–406, https://doi.org/10.5194/amt-3-397-2010, 2010.

Gu, B., Zhang, L., Dingenen, R. Van, Vieno, M., Grinsven, H. J. Van, Zhang, X., Zhang, S., Chen, Y., Wang, S., Ren, C., Rao, S., Holland, M., Winiwarter, W., Chen, D., Xu, J., and Sutton, M. A.: Abating ammonia is more cost-effective than nitrogen oxides for mitigating PM2.5 air pollution, Science (80-. )., 374, 758–762, https://doi.org/10.1126/science.abf8623, 2021.

Moravek, A., Singh, S., Pattey, E., Pelletier, L., and Murphy, J. G.: Measurements and quality control of ammonia eddy covariance fluxes: a new strategy for high-frequency attenuation correction, Atmos. Meas. Tech., 12, 6059–6078, c, 2019.

Pozzer, A., Tsimpidi, A. P., Karydis, V. a., de Meij, A., and Lelieveld, J.: Impact of agricultural emission reductions on fine-particulate matter and public health, Atmos. Chem. Phys., 17, 12813–12826, https://doi.org/10.5194/acp-17-12813-2017, 2017.

Powered by TCPDF (www.tcpdf.org)

---

## Author Response (AR2)

Response to the editor comment on amt-2022-107 on "In-situ measurements of NH3: instrument performance and applicability" by Marsailidh M. Twigg et al., Atmos. Meas. Tech. Discuss., https://doi.org/10.5194/amt-2022-107-RC1, 2022

The authors would like to thank the editor for taking the time to review the responses to the reviewer comments and for providing constructive feedback concerning the title of the manuscript. We agree that the title of the manuscript should be revised. We believe the revised title presented at the end of this document better describes the manuscript and thank the editor for their input. Please find the submission of the supplementary information and manuscript with the revised title.

 *"Inter-comparison of in-situ measurements of ambient NH3: instrument performance and application under field conditions".*

We apologise but in reviewing the final submission of the manuscript, we identified two spelling errors and two formatting errors (additional spaces between words). The manuscript has been corrected for the errors.